



# A framework for likelihood functions of deterministic hydrological models

Lorenz Ammann[1,2], Peter Reichert[1,2], and Fabrizio Fenicia[1]

[1]Swiss Federal Institute of Aquatic Science and Technology (Eawag), Dubendorf, Switzerland
[2]Department of Environmental Systems Science, ETH Zurich, Zurich, Switzerland

**Correspondence:** Lorenz Ammann (lorenz.ammann@eawag.ch)

**Abstract.** The widespread application of deterministic hydrological models in research and practise calls for suitable methods to describe their uncertainty. The errors of those models are often heteroscedastic, non-Gaussian and correlated due to the memory effect of errors in state variables. Still, the residual error models used to describe them are usually highly simplified, often neglecting some of the mentioned characteristics. This is partly because general approaches to account for all of those characteristics are lacking, and partly because the benefits of more complex error models in terms of achieving better predictions are unclear. For example, the joint inference of autocorrelation and hydrological model parameters has been shown to lead to poor predictions. This study presents a framework for likelihood functions for deterministic hydrological models that considers correlated errors and allows for an arbitrary probability distribution of observed streamflow. The choice of this distribution reflects prior knowledge about non-normality of the errors. The framework was used to evaluate increasingly complex error models with data of varying temporal resolution (daily to hourly) in two catchments. We found that (1) the joint inference of hydrological and error model parameters leads to poor predictions when conventional error models with stationary correlation are used, which confirms previous studies, (2) the quality of these predictions worsens with higher temporal resolution of the data, (3) accounting for a non-stationary autocorrelation of the errors, i.e. allowing it to vary between wet and dry periods, largely alleviates the observed problems, and (4) accounting for autocorrelation leads to more realistic model output as shown by signatures such as the Flashiness Index. Overall, this study contributes to a better description of residual errors of deterministic hydrological models.

## 1 Introduction

Deterministic hydrological models are widely applied in research and decision-making processes. The quantification of their associated uncertainties is therefore an important task with high relevance for the scientific learning process, as well as for operational decisions with respect to water management. The total output uncertainty of those models is a combination of (i) propagated input uncertainty (e.g., Sun et al., 2000; Kavetski et al., 2003; Bárdossy and Das, 2008), (ii) model structural errors (e.g. Butts et al., 2004), which can be attributed to aggregation and parameterisation, (iii) parameter uncertainty (e.g. Freer et al., 1996; Wagener et al., 2001), and (iv) observation errors of the output, for example due to errors in rating curves (e.g.



Kuczera and Franks, 2002). The sources (i-iv) usually result in residual errors of streamflow predictions with the following characteristics:

- Non-normality: model residuals are seldom well represented by a normal distribution with constant mean and variance. Instead, residuals are typically heteroscedastic (increasing with streamflow), right-skewed, and charactered by excess
kurtosis (fat tails) (e.g. Schoups and Vrugt, 2010).

- Autocorrelation: several sources of error cause memory effects. For example, errors in internal states of the model (Kavetski et al., 2003) or missed rainfall events, which can have an effect on the residuals several days after the event has occurred (e.g. Beven and Westerberg, 2011)

- Non-stationarity: model residuals can have very different characteristics in time. For example, during wet periods dom-
inated by rainfall, errors are generally less correlated than during dry periods (Yang et al., 2007). Schaefli et al. (2007) find that residuals are less correlated during high flows than during low flows in a glacierised alpine catchment.

- Unequally spaced observations: observations do not always take place at fixed time intervals. Particularly for water quality, volume-proportional sampling strategies are generally preferable to fixed-time strategies (e.g. Schleppi et al., 2006). These strategies generate observations at unequal time intervals. Another cause of unequal observation intervals
is missing data.

- Non-negativity: in typical situations, streamflow measurements are non-negative meaning that streamflow always flows in the same direction

Various studies have investigated error models that consider correlation, heteroscedasticity and non-normality of errors of deterministic hydrological models. A typical approach, which is also applied in this study, is to describe total output uncertainty
in a lumped way (e.g. Schoups and Vrugt, 2010; McInerney et al., 2017). Another group of approaches distinguishes among the different sources of total uncertainty such as input, parametric and output measurement uncertainty (e.g. Kavetski et al., 2006; Renard et al., 2010). The latter approach requires information about input and output uncertainty to overcome an otherwise ill-posed problem, and is not pursued in this work. Current approaches to describe total output uncertainty in a lumped way differ in if, and how, they deal with the various characteristics of residual errors mentioned above. Some of the most common
approaches are the following:

- Residuals are often normalised for weighed least squares error models by parameterising the variance of the normal distribution as a function of the streamflow (Thyer et al., 2009; Evin et al., 2013; Bertuzzo et al., 2013). Another common approach is to apply transformations such as Box-Cox to the observed and modelled streamflow time series and formulate a model for the residuals of the transformed time series (e.g. Bates and Campbell, 2001; Giudice et al., 2013;
McInerney et al., 2017). However, this transformation affects several properties of the residuals simultaneously, including heteroscedasticity, skewness and kurtosis.



- Typically, residual errors are represented as a stationary process. The issue of stationarity has been the subject of recent debate (Milly et al., 2008; Montanari and Koutsoyiannis, 2014). Focusing on streamflow dynamics, an example of representing non-stationarity of residual errors is Yang et al. (2007), who distinguish between wet and dry periods by applying a continuous autoregressive process with different parameters for the wet and the dry periods to the Box-Cox transformed residuals.

- A likelihood function to deal with unequally spaced data was proposed by Duan et al. (1988). A more natural formulation is to adopt a continuous-time formulation of the autoregressive model, such as an Ornstein-Uhlenbeck process (e.g. Kloeden and Platen, 1995; Yang et al., 2007).

- Non-negativity can be addressed by truncating the error pdf so that it does not extend to negative streamflow. However, this approach is seldom followed, and in most applications the truncation occurs "in prediction only" (McInerney et al., 2017).

Residual error models are usually highly simplified, in the sense that they do not account for all the above mentioned characteristics of these errors. In particular, residual error models seldom go beyond using "variance stabilisation" techniques such as Box-Cox. The widespread use of relatively simple error models is due to several reasons. In our opinion, the following are the most important.

First, there is a lack of general approaches that can deal with all the above mentioned characteristics of error models simultaneously. One general error model that can accommodate various characteristics is the likelihood parameterisation proposed by Schoups and Vrugt (2010), which can deal with residual errors that are correlated, heteroscedastic, and non-Gaussian with varying degrees of kurtosis and skewness. They do this by formulating an autoregressive process with a skew exponential power (SEP) rather than a normal distribution. This results in marginal distributions for streamflow at given time points that can be non-normal and skewed, but are not easily accessible analytically. Furthermore, the approach was shown to produce unrealistically large predictive uncertainties caused by the application of the autoregressive process to non-standardised residuals (Evin et al., 2013).

Second, there is limited guidance to the choice of a particular error model for a given application. In the past, the choice has been generally ad-hoc, with limited justification. Only recently, there has been more systematic comparison and testing which has resulted in some general recommendations. For example, McInerney et al. (2017) compared various residual error schemes, including standard and weighted least squares, the Box-Cox transformation (with fixed and calibrated power parameter) and the log-sinh transformation on 23 catchments, and concluded that Box-Cox has on average the best behaviour.

Third, previous experience has shown that more realistic error models, which are more complex, do not always result in better predictions. The additional parameters of some of the more complex error models were found to have undesirable interactions with the parameters of the hydrological model, leading to unrealistic parameter values and poor predictions. For





example, particularly in dry catchments, accounting for autocorrelation produces worse predictions than omitting it (Schoups and Vrugt, 2010; Evin et al., 2013). To circumvent such problems, Evin et al. (2014) recommended inferring autoregressive parameters sequentially, that is, after having estimated all other parameters of the hydrological and of the error model. The joint inference of hydrological and error model parameters remains conceptually preferable, as it recognises potential interactions

between parameters. Understanding the conditions under which this can be achieved remains poorly understood.

Fourth, the potential advantages of more complex error models are under-appreciated by the hydrological community. Most commonly, residual error models are used to plot some "uncertainty bands" around the hydrograph. For such purposes, the use of relatively simplified error models may appear justified. However, there are several applications that go beyond this task, and

for which a simplified error model may lead to poor results. For example, assuming uncorrelated errors may lead to unrealistic extrapolations (Giudice et al., 2013) or unrealistic values of hydrograph signatures, particularly if these are sensitive to noise, such as for example the Flashiness Index (Baker et al., 2004; Fenicia et al., 2018). The ability of correctly representing signatures is not only important for conceptual reasons, but also for practical purposes such as in signature based model calibration.

The goals of this study are the following:

1. Develop a framework for likelihood functions for hydrological models that accounts for the following major characteristics of their errors: non-normality (heteroscedasticity, skewness and excess kurtosis), autocorrelation, non-stationarity in wet and dry periods, unequally spaced observation time points and non-negativity of streamflow.

2. Investigate the ability to infer the various parameters of the error model and the quality of the predictive distributions. In

particular, with case studies in two catchments, we investigate the following questions:

   (a) Can we confirm previous findings about the problems related to joint inference of hydrological and error model parameters?

   (b) What are the causes of the problems encountered in joint inference of hydrological and error model parameters?

   (c) Can we improve the joint inference by introducing non-stationarity by allowing the autoregressive parameter to

change between wet and dry periods?

   (d) Does the consideration of autocorrelation lead to more realistic predictions (e.g. in terms of better representation of hydrograph signatures such as the Flashiness Index)?

   (e) Can parameters controlling the shape of the distribution of the errors be inferred jointly with the hydrological model parameters to account for non-normality?

Note that the developed framework allows for additional flexibility in aspects that are not covered with Questions 2a-2e (e.g. unequally spaced observations, non-negativity). To limit the scope of this paper, we refrain from controlled experiments w.r.t. those aspects. The paper is structured as follows. The theoretical framework for likelihood functions, corresponding to Goal 1,





is presented in Sect. 2.1 and the performance metrics used to evaluate it are described in Sect. 2.3. Section 3 describes the case study setup used to carry out the necessary investigations for Goal 2. The case study is based on two catchments (Sect. 3.1), one hydrological bucket model (Sect. 3.2) and three different time step sizes (daily, 6-hourly and hourly). The results of those investigations are presented in Sect. 4 and discussed in Sect. 5. Section 6 lists the main conclusions and sketches potential

directions for future research.

## 2   Methods

### 2.1   Likelihood framework

Suppose we choose the distribution $D_Q$ to describe the probability of observing streamflow $Q$, given the model output $Q_{\mathrm{det}}$. We believe that this is a natural place to start the derivation of a likelihood function for hydrological models, since many

modellers will have an intuitive idea about the probability distribution of the observations given an output of their model (Fig. 2). Providing explicit control over $D_Q$ facilitates the formulation of the model based on prior knowledge resulting from past experience of hydrologists. Note the major difference to the approach of Schoups and Vrugt (2010) and transformation based approaches (Bates and Campbell, 2001; Giudice et al., 2013; McInerney et al., 2017, e.g.), where $D_Q$ is not easily accessible. Wani et al. (In preparation) present another approach in which the temporal dependence of $D_Q$ is accessed through copulas.

We assume that $D_Q$ is parameterised by $Q_{\mathrm{det}}$ and some error model parameters $\boldsymbol{\psi}$, i.e. $Q(t) \sim D_Q(Q_{\mathrm{det}}(t), \boldsymbol{\psi})$. This implies that the observed streamflow at different time points can be described by different distributions, but these distributions belong to the same parametric family $D_Q$. The distribution $D_Q$ may extend to negative values. In this case, the integrated probability of negative values is assigned to the probability of observing a streamflow of zero. This leads to

$$p_{D_Q(Q_{\mathrm{det}}, \boldsymbol{\psi})}(Q) = \begin{cases} f_{D_Q(Q_{\mathrm{det}}, \boldsymbol{\psi})}(Q) & \text{for } Q > 0 \\ F_{D_Q(Q_{\mathrm{det}}, \boldsymbol{\psi})}(0) & \text{for } Q = 0 \\ 0 & \text{for } Q < 0 \end{cases} \tag{1}$$

where $f_{D_Q}$ and $F_{D_Q}$ be the density and cumulative distribution function of $D_Q$, respectively. $p$ is a probability density for $Q > 0$ and a discrete probability for $Q = 0$. Note that Eq. (1) reflects our prior knowledge that $Q \geq 0$. Transforming $Q$ according to Eq. (3) leads to a (potentially correlated) time series:

$$\eta(t_i) = \eta_{\mathrm{trans}}(Q(t_i), Q_{\mathrm{det}}(t_i), \boldsymbol{\psi}) \tag{2}$$

with

$$\eta_{\mathrm{trans}}(Q, Q_{\mathrm{det}}, \boldsymbol{\psi}) = F_{\mathrm{N(0,1)}}^{-1}\big(F_{D_Q(Q_{\mathrm{det}}, \boldsymbol{\psi})}(Q)\big) \tag{3}$$

where $Q_{\mathrm{det}}$ is the result of the deterministic hydrological model. Note that, if the distributional assumptions about $D_Q$ hold, $\eta$ will be marginally standard normally distributed, except for the truncation at zero, which can lead to lighter tails on the lower end.





To describe autocorrelation between successive streamflow values, we assume that the corresponding time series of $\eta$ are discrete-time results of a continuous-time autoregressive process:

$$\eta(t_{i+1}) \mid \eta(t_i) \sim \mathrm{N}\left(\eta(t_i)\exp\left(-\frac{t_{i+1}-t_i}{\tau(t_{i+1})}\right), \sqrt{1-\exp\left(-2\frac{t_{i+1}-t_i}{\tau(t_{i+1})}\right)}\right) \qquad (4)$$

This so-called Ornstein-Uhlenbeck process (Uhlenbeck and Ornstein, 1930) has a standard normal asymptotic distribution and

5   $\tau(t_{i+1})$ represents the characteristic correlation time that is assumed to be constant over the interval $[t_i, t_{i+1}]$. Consider the conventional AR(1) process:

$$x_t = \phi x_{t-1} + \epsilon_t \qquad (5)$$

where $\epsilon_t$ is white noise. For equidistant time steps, the relation between $\tau$ in Eq. (4) and $\phi$ in Eq. (5) is given by:

$$\tau = -\frac{\Delta t}{\ln(\phi)} \qquad (6)$$

10   where $\Delta t$ is the size of the time step. To apply Eq. (4) to transfer information between time points, we transform the distribution $D_Q$ at time $t_i$ to a standard normal distribution $\eta_i$ according to Eq. (3), advance $\eta_i$ to $\eta_{i+1}$ according to Eq. (4), and transform $\eta_{i+1}$ back to $D_Q$ at time $t_{i+1}$. The formulation of a continuous-time autoregressive process with evaluation at discrete time points allows us to apply it to non-equidistant time series; one advantage of this formulation is that it allows us to easily deal with missing data.

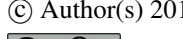



In order to formulate the likelihood of the streamflow $Q$, we derived the following conditional probabilities for $Q(t_{i+1})$ given $Q(t_i)$ (see Appendix A for the full derivation):

if $Q(t_i) > 0$ :

$$p_{i+1}\big(Q(t_{i+1}) \mid Q(t_i), \boldsymbol{\theta}, \boldsymbol{\psi}\big)$$

$$= \begin{cases} f_{D_Q(Q_{\det}(t_{i+1},\boldsymbol{\theta}),\boldsymbol{\psi})}\big(Q(t_{i+1})\big) \\ \quad \cdot \dfrac{f_{\mathrm{N}\left(\eta_{\mathrm{trans}}\left(Q(t_i),Q_{\det}(t_i,\boldsymbol{\theta}),\boldsymbol{\psi}\right)\exp\left(-\frac{t_{i+1}-t_i}{\tau}\right),\sqrt{1-\exp\left(-2\frac{t_{i+1}-t_i}{\tau}\right)}\right)}\left(\eta_{\mathrm{trans}}\left(Q(t_{i+1}),Q_{\det}(t_{i+1},\boldsymbol{\theta}),\boldsymbol{\psi}\right)\right)}{f_{\mathrm{N}(0,1)}\left(\eta_{\mathrm{trans}}\left(Q(t_{i+1}),Q_{\det}(t_{i+1},\boldsymbol{\theta}),\boldsymbol{\psi}\right)\right)} \\ \qquad\qquad \text{for } Q(t_{i+1}) > 0 \\[2ex] F_{\mathrm{N}\left(\eta_{\mathrm{trans}}\left(Q(t_i),Q_{\det}(t_i,\boldsymbol{\theta}),\boldsymbol{\psi}\right)\exp\left(-\frac{t_{i+1}-t_i}{\tau}\right),\sqrt{1-\exp\left(-2\frac{t_{i+1}-t_i}{\tau}\right)}\right)}\left(\eta_{\mathrm{trans}}\left(0,Q_{\det}(t_{i+1},\boldsymbol{\theta}),\boldsymbol{\psi}\right)\right) \\ \qquad\qquad \text{for } Q(t_{i+1}) = 0 \end{cases} \quad (7)$$

if $Q(t_i) = 0$ :

$$p_{i+1}\big(Q(t_{i+1}) \mid Q(t_i), \boldsymbol{\theta}, \boldsymbol{\psi}\big)$$

$$= \begin{cases} f_{D_Q\left(\boldsymbol{\xi}(Q_{\det}(t_{i+1},\boldsymbol{\theta}),\boldsymbol{\psi})\right)}\big(Q(t_{i+1})\big) & \text{for } Q(t_{i+1}) > 0 \\[1.5ex] F_{D_Q\left(\boldsymbol{\xi}(Q_{\det}(t_{i+1},\boldsymbol{\theta}),\boldsymbol{\psi})\right)}(0) & \text{for } Q(t_{i+1}) = 0 \end{cases}$$

Note that $p$ is a probability density (denoted by $f$) if $Q(t_{i+1}) > 0$, and an integrated, discrete probability (denoted by $F$) if
5 $Q(t_{i+1}) = 0$. The full likelihood is then simply the product of the conditional probabilities in Eq. (7):

$$f_{\mathrm{L}}\big(Q(t_0), Q(t_1), \ldots, Q(t_n) \mid \boldsymbol{\theta}, \boldsymbol{\psi}\big) = p_{D_Q(Q_{\det}(t_0,\boldsymbol{\theta}),\boldsymbol{\psi})}\big(Q(t_0)\big) \prod_{i=1}^{n} p_i\big(Q(t_i) \mid Q(t_{i-1}), \boldsymbol{\theta}, \boldsymbol{\psi}\big) \qquad (8)$$

## 2.2 Inference and prediction

When performing inference, the parameters of the hydrological model, $\boldsymbol{\theta}$, are estimated jointly with the parameters of the error model, $\boldsymbol{\psi}$, by evaluating the likelihood function (Eq. 8) according to the following procedure:

10   1. Given a suggested parameter vector $\boldsymbol{\theta}$, evaluate the deterministic hydrological model, $Q_{\det}$, for all time points.

2. Using $\boldsymbol{\psi}$ and $Q_{\det}$, calculate the likelihood in Eq. (8) by substituting the argument $Q$ with the observed streamflow, $Q_{\mathrm{obs}}$.

As the likelihood (Eq. 8) can efficiently be evaluated analytically for a given output of the hydrological model, we do Bayesian inference based on standard MCMC sampling of the posterior. The affine-invariant ensemble sampler by Foreman-Mackey et al. (2013) is used for this purpose.



For prediction, stochastic realisations of model output are obtained by inverting Eq. (3):

$$Q_{\text{trans}}(\eta, Q_{\text{det}}, \boldsymbol{\psi}) = F^{-1}_{D_Q(Q_{\text{det}},\boldsymbol{\psi})}\big(F_{\text{N}(0,1)}(\eta)\big) \tag{9}$$

and applying the following procedure to produce a single stochastic streamflow realisation $\boldsymbol{Q}_j$:

1. Randomly draw a parameter vector $(\boldsymbol{\theta},\boldsymbol{\psi})_j$ from the posterior sample.

2. Using $\boldsymbol{\theta}_j$, evaluate the deterministic hydrological model to obtain, $\boldsymbol{Q}_{\text{det},j}$, for all time points.

3. Using $\boldsymbol{\tau}_j \in \boldsymbol{\psi}_j$ and Eq. (4), produce a stochastic realisation of an OU-process, $\boldsymbol{\eta}_j$, with a standard normal marginal distribution.

4. Use $Q_{\text{det},j}$ and $\boldsymbol{\psi}_j$, determined in Steps 1 and 2, to transform $\boldsymbol{\eta}_j$ into a stochastic realisation of streamflow, $\boldsymbol{Q}_j$, with Eq. (9).

In a synthetic case study, we could successfully verify the consistency of the implemented likelihood and sampling functions (Appendix C).

## 2.3 Evaluation criteria

How can the performance of empirical error models, as the ones presented in this study, be quantified? We argue that the performance of an error model in joint inference with a hydrological model should be judged according to following criteria: (a) good reproduction of observed dynamic fluctuations, (b) good overall predictive distributions and (c) small absolute deviance between model output and observations. The Flashiness Index (Sect. 2.3.1) quantifies (a). The reliability and the precision of the predictive distribution (Sect. 2.3.2 and 2.3.3, respectively) are used as an indicator for (b). The Nash-Sutcliffe Efficiency (Sect. 2.3.4) and the relative error in cumulative streamflow (Sect. 2.3.5) cover (c).

### 2.3.1 Flashiness Index

The Flashiness Index (Baker et al., 2004), $I_F$, is given by:

$$I_F(\boldsymbol{Q}) = \frac{\sum_{i=2}^{N} Q(t_i) - Q(t_{i-1})}{\sum_{i=2}^{N} Q(t_i)} \tag{10}$$

where $\boldsymbol{Q} = (Q(t_1), Q(t_2), \ldots, Q(t_N))$. $I_F$ is calculated for the observations, $Q_{\text{obs}}$, the output of the deterministic hydrological model at the maximum posterior parameter values, $\widehat{Q}_{\text{det}}$, and the individual realisations of the full predictive distribution of streamflow, $Q_j$. The resulting metrics are denoted as $I_{F,\text{obs}}$, $\widehat{I}_{\text{F,det}}$ and $I_F$, respectively, where the latter is the median of the flashiness indices of the individual realisations $Q_j$. $I_F$ is sensitive to the amount of autocorrelation in a streamflow time series, as well as the height of the peaks of $Q_{\text{det}}$.



### 2.3.2 Reliability

Reliability is defined equivalently to McInerney et al. (2017), as:

$$\Xi_{\mathrm{reli}} = \frac{2}{N_t} \sum_{i=1}^{N_t} |F_{Q(t_i)}(Q_{\mathrm{obs}}(t_i)) - F_\Psi(F_{Q(t_i)}(Q_{\mathrm{obs}}(t_i)))| \tag{11}$$

where $\Psi = \{F_{Q(t_i)}(Q_{\mathrm{obs}}(t_i))|i \in \mathbb{N}, i \le N_t\}$, $F_\Psi$ is the empirical cumulative distribution function of $\Psi$ and $F_{Q(t_i)}$ is the empirical cumulative distribution function of the predicted streamflow at time $t_i$. $\Xi_{\mathrm{reli}}$ can take values in the interval $[0,1]$, where smaller values of $\Xi_{\mathrm{reli}}$ correspond to better, and zero to perfect, reliability. It summarises the deviance of the observations from the predictive distribution over all time points, and the distance is measured in the uniform space. Therefore, the influence of heavy outliers on $\Xi_{\mathrm{reli}}$ is limited.

### 2.3.3 Precision

The precision metric is an indicator for the width of the predictive distributions over all time points, and was proposed by McInerney et al. (2017) as:

$$\Omega_{\mathrm{prec}} = \frac{\sum_{i=1}^{N_t} \sigma_{Q(t_i)}}{\sum_{i=1}^{N_t} Q_{\mathrm{obs}}(t_i)} \tag{12}$$

where $\sigma_{Q(t_i)}$ is the standard deviation of the predictive distribution at time point $t_i$ calculated from the ensemble of all stochastic predictions at that point in time. $\Omega_{\mathrm{prec}} \in \mathbb{R}^+$, and small values of $\Omega_{\mathrm{prec}}$ indicate high precision or small predictive uncertainty. The smaller the predictive uncertainty, the better the quality of the underlying model, given that the predictions are not overconfident.

### 2.3.4 Nash-Sutcliffe Efficiency

The Nash-Sutcliffe Efficiency (Nash and Sutcliffe, 1970), $E_N$, is defined as:

$$E_N(\boldsymbol{Q}, \boldsymbol{Q}_{\mathrm{obs}}) = 1 - \frac{\sum_{i=1}^{N}(Q(t_i) - Q_{\mathrm{obs}}(t_i))^2}{\sum_{i=1}^{N}(Q(t_i) - \overline{Q}_{\mathrm{obs}})^2} \tag{13}$$

where $\boldsymbol{Q} = (Q(t_1), Q(t_2), \dots, Q(t_N))$. It is used in this study to quantify the agreement between $\widehat{Q}_{\mathrm{det}}$ and $Q_{\mathrm{obs}}$, as well as between the $j$-th stochastic realisation $Q_j$ and $Q_{\mathrm{obs}}$. The two cases are denoted as $\widehat{E}_{N,\mathrm{det}}$ and $E_N$, respectively, where $E_N$ is the median of the efficiencies of the individual realisations $Q_j$. It is used as a rough measure of how well two hydrographs correspond to each other, primarily with the goal of identifying very poorly fitting hydrographs. It is known to be sensitive to errors in high flows (Legates and McCabe, 1999), which can be of particular practical interest. Therefore it complements the other measures, which are less informative with respect to errors in high flows.



### 2.3.5 Relative error in total cumulative streamflow

As a measure of systematic over- or under-prediction of streamflow, we calculate the relative error in total cumulative streamflow:

$$\Delta(Q, Q_{\mathrm{obs}}) = \frac{\int Q_{\mathrm{obs}}(t) - Q(t)dt}{\int Q_{\mathrm{obs}}(t)dt} \tag{14}$$

It is calculated w.r.t. the maximum posterior output of the deterministic model; $\widehat{\Delta}_{\mathrm{Q,det}} = \Delta_Q(\widehat{Q}_{\mathrm{det}}, Q_{\mathrm{obs}})$, as well as for the individual stochastic simulations: $\Delta_Q = \mathrm{median}(\Delta(Q_j, Q_{\mathrm{obs}}))$. Note that, contrary to McInerney et al. (2017), $\Delta_Q$ is the median error of the individual hydrograph realisations, not the error of the averaged hydrographs.

## 3   Case study setup

### 3.1   Catchments and data

The likelihood framework developed in Sect. 2.1 was tested in two case study sites, the Murg and the Maimai catchmets, which are described in this section. The Murg river flows through a hilly headwater catchment in temperate climate with a size of 80 km$^2$ in northeastern Switzerland. Some key hydrological summary statistics are listed in Table 1. Land use is predominantly agricultural (50 %), with forested headwaters (30 %) and a considerable part of urban areas (10 %). The mean elevation is 652 m a.s.l., spanning from 466 to 1035 m a.s.l. Streamflow peaks can be quite sharp, especially for small events, in which baseflow

conditions are reached again within just a few hours. This is potentially due to impervious areas being drained directly into the river. The data consists of hourly averages of streamflow, precipitation and potential evapotranspiration from January 1995 to December 2002. Calibration was performed in the first 5 years (Jan 1995-Dec 1999) and validation in the consecutive 3 years (Jan 2000-Dec 2002). Streamflow data is a courtesy of the Swiss Federal Office for the Environment (FOEN). Precipitation and potential evapotranspiration are based on meteorological data (Meteoschweiz, 2018) and were processed by the Swiss Fed-

eral Institute for Forest, Snow and Landscape Research (WSL), with the preprocessing tools of PREVAH (Viviroli et al., 2009).

The Maimai experimental catchments are a set of small headwater catchments with a long history of hydrological research. They are located on a deeply incised hillslope on the South Island of New Zealand. The area is forested and the climate is considerably more humid than in the Murg catchment (Table 1). The site was chosen for this study due to its homogeneous

characteristics and relatively simple hydrological response, which make it very suited for model evaluation and testing (e.g. Seibert and McDonnell (2002)). We use hourly data recorded in 1985-1987 in the M8 experimental catchment, the most intensely studied of the Maimai catchments. It has an area of ca. 7 ha with steep (34°) slopes. The reader is referred to Brammer and McDonnell (1996) for a more detailed description of the characteristics of the M8 and the other experimental catchments. This study does not attempt to make a significant contribution to the understanding of the hillslope processes in

the Maimai catchment (see McGlynn et al. (2002) for an extensive overview). Calibration was performed based on data from



**Table 1.** Properties of the two case study catchments. P is the precipitation and $R_C$ the runoff coefficient (calculated from cumulative streamflow and precipitation). $Q_{\mathrm{obs,max}}$, $Q_{\mathrm{obs,min}}$ and $\overline{Q}_{\mathrm{obs}}$ are the minimum, the maximum and the average streamflow, respectively. $I_{\mathrm{F,obs}}$ is the Flashiness Index (Baker et al., 2004).

| Catchment | Area | $P$ | $R_C$ | $Q_{\mathrm{obs,max}}$ | $Q_{\mathrm{obs,min}}$ | $\overline{Q}_{\mathrm{obs}}$ | $I_{\mathrm{F,obs}}$ |
|---|---|---|---|---|---|---|---|
| | [km$^2$] | [mm a$^{-1}$] | [-] | [mm h$^{-1}$] | [mm h$^{-1}$] | [mm h$^{-1}$] | [-] |
| Murg | 80 | 1369 | 0.57 | 2.7 | 1e-2 | 0.089 | 0.053 |
| Maimai | 0.07 | 2349 | 0.62 | 8.5 | 1e-4 | 0.17 | 0.13 |

Jan 1985-Dec 1986, and validation during Jan-Dec 1987. The data was kindly provided by Jeffrey McDonnell.

While the resolution of the original data was hourly, we produced data sets with 6-hourly and daily resolution by aggregation for both catchments. This setup allows us to systematically investigate the effect of the temporal resolution of the data on the joint inference of hydrological and error model parameters. This could contribute to the identification of the cause of previously encountered problems in joint inference (Goal 2b specified in Sect. 1). Furthermore, the two selected catchments are different in size, signatures (Table 1), and complexity of their hydrological response, so that the influence of the catchment or data properties can be assessed to some degree. To limit the scope of the study, we constrained the analysis to two catchments.

### 3.2 Deterministic Hydrological Model

The hydrological model used throughout this study is a simple, lumped bucket model with two elements (Figure 1), which are meant to represent the unsaturated soil zone and the subsurface flow being fed by it. A slower flow component is included though a linear outflow from the unsaturated zone reservoir directly. Due to its simplicity, and due to the fact that it is not clear whether the chosen model structure is suited for the studied catchment a priori, we expect systemic difficulties in reproducing the observed streamflow dynamics. This is a very common situation in hydrological modelling and it will lead to correlated and potentially heteroscedastic and non-normal errors. This allows us, in principle, to test the error models (Sect. 3.3) under realistic conditions. The streamflow simulated by this deterministic model is denoted as $Q_{\mathrm{det}}(t, \boldsymbol{\theta}) = Q_s + Q_f$, where $\boldsymbol{\theta} =$





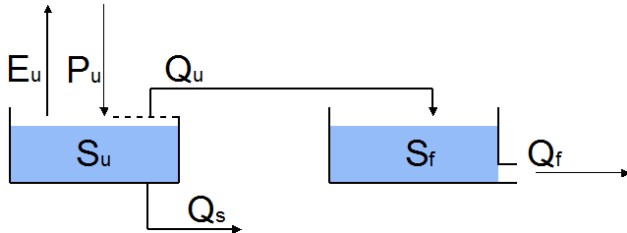

**Figure 1.** Structure of the deterministic hydrological model used in this study. $P_u$ is the precipitation and $E_u$ the evapotranspiration. $S_u$ represents the active water content of the unsaturated zone, while $S_f$ is a non-linear reservoir representing the fast flow component.

$(C_e, S_{\max}, k_u, k_f)$ are the hydrological parameters. The fluxes $(E_u, P_u, Q_u, Q_s, Q_f)$ and states $(S_u, S_f)$ of the model are given by:

$$
\frac{dS_u}{dt} = P_u - E_u - Q_u - Q_s
$$

$$
E_u = C_e E_p \frac{\frac{S_u}{S_{\max}}(1+m)}{\frac{S_u}{S_{\max}}+m}
$$

$$
Q_u = P_u \left(\frac{S_u}{S_{\max}}\right)^{\beta} \tag{15}
$$

$$
Q_s = k_u S_u
$$

$$
\frac{dS_f}{dt} = Q_u - Q_f
$$

$$
Q_f = k_f S_f{}^{\alpha} \tag{16}
$$

5  where $E_p$ is the potential evapotranspiration. The following parameters were kept fixed: $m = 0.01$, $\beta = 3$, and $\alpha = 2$.

### 3.3 Error Models

For $D_Q$, we use the skewed Student's $t$-distribution (Fig. 2) as the most general case, which is obtained by transforming the conventional Student's $t$-distribution according to Fernandez and Steel (1998). Thus, we introduce two error model parameters: $\gamma$, defining the degree of skewness, and $df$, the degrees of freedom as a measure for the kurtosis. The skewed Student's $t$-

10  distribution reduces to the normal distribution for $\gamma = 1$ and $df \to \infty$. Two assumptions are tested to centre $D_Q$ at $Q_{\det}$:

$$
E[D_Q] = Q_{\det}(t) \tag{17a}
$$

$$
\mathrm{mode}(D_Q) = Q_{\det}(t) \tag{17b}
$$




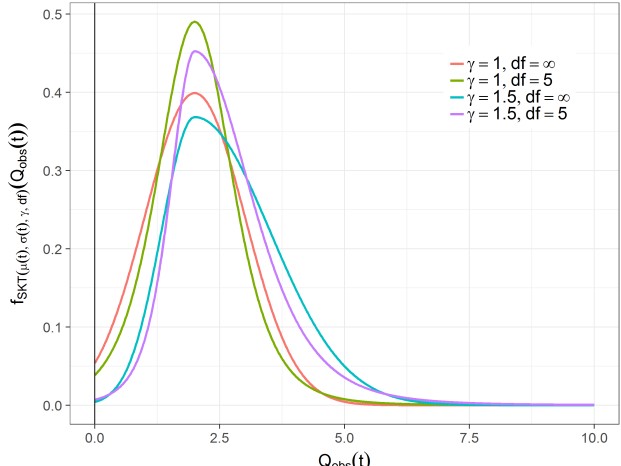

**Figure 2.** Shape of the skewed Student's *t*-distribution for different values of skewness, $\gamma$, and kurtosis, $df$.

i.e. we either assign the expected value or the highest probability of $D_Q$ to $Q_{\text{det}}$. A third alternative would be to set the median of $D_Q$ equal to $Q_{\text{det}}$. It is a priori unclear which of those options is most suitable. By testing the two options in Eq. (17), we include the lowest and the highest value, the third option would be a compromise between the two and was not included in the study. If not indicated otherwise, the assumption in Eq. (17a) was used. The results obtained with Eq. (17b) can be found in Appendix B.

The standard deviation of $D_Q$ is parameterised as follows:

$$\sigma_{D_Q}(t) = aQ_0 \left( \frac{Q_{\text{det}}}{Q_0} \right)^c + bQ_0 \tag{18}$$

Note that skewing a distribution with the approach developed by Fernandez and Steel (1998) changes its standard deviation; $\sigma_{D_Q}(t)$ is the standard deviation of $D_Q$ after skewing. Other parameterisations of $\sigma_{D_Q}$ are in principle possible; see McInerney et al. (2017) for a theoretical correspondence with transformation approaches. Like previous studies (Thyer et al., 2009; Schoups and Vrugt, 2010; Evin et al., 2013), we set $c = 1$ in Eq. (18). McInerney et al. (2017) have shown that transformation approaches with a first order correspondence to $c = 0.8$ or $c = 0.5$ can lead to more reliable and precise predictions than those corresponding to $c = 1$. To limit the scope of the analysis, which focuses on the autocorrelation of the errors, we set $c = 1$. This also leads to better comparability to the aforementioned studies. Note that the parameters $a$ and $b$ become dimensionless (and therefore more universal) by including a reference streamflow, $Q_0 = E[Q_{\text{obs}}]$. Thus, $a$ can be seen as the standard deviation of the error relative to the modelled streamflow, while $b$ represents a characteristic fraction of the reference streamflow $Q_0$, below which the magnitude of the error starts to become less dependent on $Q_{\text{det}}$.

Table 2 lists the error models applied in this study, together with their underlying assumptions. E1 is included as a reference case; it is based on the assumption of uncorrelated heteroscedastic errors with a normal distribution. These assumptions,





**Table 2.** Overview of the error models applied in this study and their corresponding parameters ($\times$: fitted). If $^\star$ is appended to the name of the error model, a smoothed version of $P_{\mathrm{err}}(t)$ (moving average of window size 5 h) was used in Eq. (19).

| Error Model | $\tau_{\min}$ | $\tau_{\max}$ | $a$ | $b$ | $\gamma$ | $df$ |
|---|---|---|---|---|---|---|
| E1 | 0 | 0 | $\times$ | $\times$ | 1 | $\infty$ |
| E2 | $= \tau_{\max}$ | $\times$ | $\times$ | $\times$ | 1 | $\infty$ |
| E3$^{(\star)}$ | 0 | $\times$ | $\times$ | $\times$ | 1 | $\infty$ |
| E3a$^{(\star)}$ | $\times$ | $\times$ | $\times$ | $\times$ | 1 | $\infty$ |
| E4$^{(\star)}$ | 0 | $\times$ | $\times$ | $\times$ | $\times$ | $\times$ |
| E4a$^{(\star)}$ | $\times$ | $\times$ | $\times$ | $\times$ | $\times$ | $\times$ |

with the exception of heteroscedasticity, are identical to the ones made when e.g. maximising the Nash-Sutcliffe Efficiency, or, equivalently, minimising the squared residuals. Error Model E2 represents a conventional approach of considering auto-correlation. In the case of equally spaced time-steps, it is similar to the error model applied e.g. by Evin et al. (2013), who assume that the rescaled errors follow an AR(1) process with a standard normal marginal distribution. One difference between the two approaches is that we truncate $D_Q$ at zero. In error model E3, we additionally account for the fact that $\tau$ might be time-dependent. The following formula for $\tau$ is used in those cases:

$$
\tau(t) \quad = \quad
\begin{cases}
\tau_{\min} & \text{for } P_{\mathrm{err}}(t) > 0 \\
\\
\tau_{\max} & \text{else}
\end{cases}
\tag{19}
$$

where $P_{\mathrm{err}}$ is the precipitation used as an input for the error model. In E3, $\tau_{\min}$ is fixed at 0, while in E3a, it is fitted. $P_{\mathrm{err}}$ was either equal to the recorded precipitation, $P$, or, in case of hourly resolution in the Maimai catchment, smoothed with a moving average of window size 5 h. This was done to prevent frequent jumps between $\tau_{\min}$ and $\tau_{\max}$ during precipitation events, and to be more robust w.r.t. potential time lags between observed precipitation and streamflow. Since in the Murg catchment, smoothing did not change the results substantially, $P_{\mathrm{err}} = P$ applies there. Thus, Error Model E3a (or E3) can be seen as a mixture of E1 and E2, in the sense that $\tau$ alternates between periods of high and low (or no) correlation. Finally, E4 relaxes the assumption of normality for $D_Q$; we use a skewed Student's $t$-Distribution, inferring the degrees of freedom and the skewness. Again, E4a denotes the version where $\tau_{\min}$ is inferred.

The prior distributions of all the parameters, listed in Table 3, were assumed to be independent normal or log-normal distributions with relatively large standard deviations. A unimodal distribution is the more accurate representation of our prior believe than e.g. a uniform distribution, since we do assume that values in the middle of the suspected range are more probable





**Table 3.** Prior distributions of the hydrological and error model parameters applied in all the cases where the respective parameter was used. N = Gaussian Normal; LN = log-normal. Where lower and upper boundaries are listed, the distribution is truncated at those values.

| Parameter | Distribution | Unit | $\mu$ | $\sigma$ | low. bound. | up. bound. |
|---|---|---|---|---|---|---|
| $C_E$ | $N$ | - | 1 | 0.2 | 0.2 | 3 |
| $S_{\mathrm{max}}$ | $LN$ | mm | 148 | 1086 | 2.7 | 1086 |
| $k_{\mathrm{u}}$ | $LN$ | $\mathrm{h}^{-1}$ | 1.8e-2 | 0.13 | 2.3e-6 | 5e-2 |
| $k_{\mathrm{f}}$ | $LN$ | $\mathrm{h}^{-1}$ | 0.37 | 2.7 | 2.3e-6 | 0.37 |
| $a$ | $LN$ | - | 0.2 | 0.2 | - | - |
| $b$ | $LN$ | - | 0.1 | 0.1 | 1e-2 | 0.5 |
| $\tau_{\mathrm{max}}$ | $LN$ | h | 148 | 1086 | 0 | 2000 |
| $\gamma$ | $LN$ | - | 1 | 0.2 | 0.1 | 5 |
| $df$ | $LN$ | - | 14 | 17 | 3 | - |

than at its edge. Note that this is primarily a conceptual difference, as large standard deviations were chosen to minimise the influence of the priors on the results.

## 4 Results

After providing some general results, this section contains a more detailed summary of the results for each of the tested error
models. The complete analysis included additional error models and performance metrics, which are included in Appendix B.

Figure 3 gives an overview of the difference in Flashiness Index, the reliability and the precision in the calibration and the validation periods for both catchments, all temporal resolutions of the data and all tested error models. Figure 4 provides additional information about the relative error in cumulative streamflow, $\Delta_Q$, and about $\widehat{E}_{\mathrm{N,det}}$. The temporal resolution of the
10 data has a pronounced effect on all the analysed performance metrics. The spread over all the combinations of error models and catchments is larger for higher temporal resolutions (Fig. 3 and 4). Furthermore, the average of each metric indicates decreasing performance for increasing temporal resolution. This loss in performance is more pronounced in the Murg catchment and for Error Models E2 and E3a than in the Maimai catchment and for other error models. The difference between the two catchments is most clearly visible in $\widehat{E}_{\mathrm{N,det}}$ (Fig. 4): for 6-hourly and daily resolution of the data, the worst performing error
model in the Maimai catchment has a better $\widehat{E}_{\mathrm{N,det}}$ than the best performing error model in the Murg catchment.





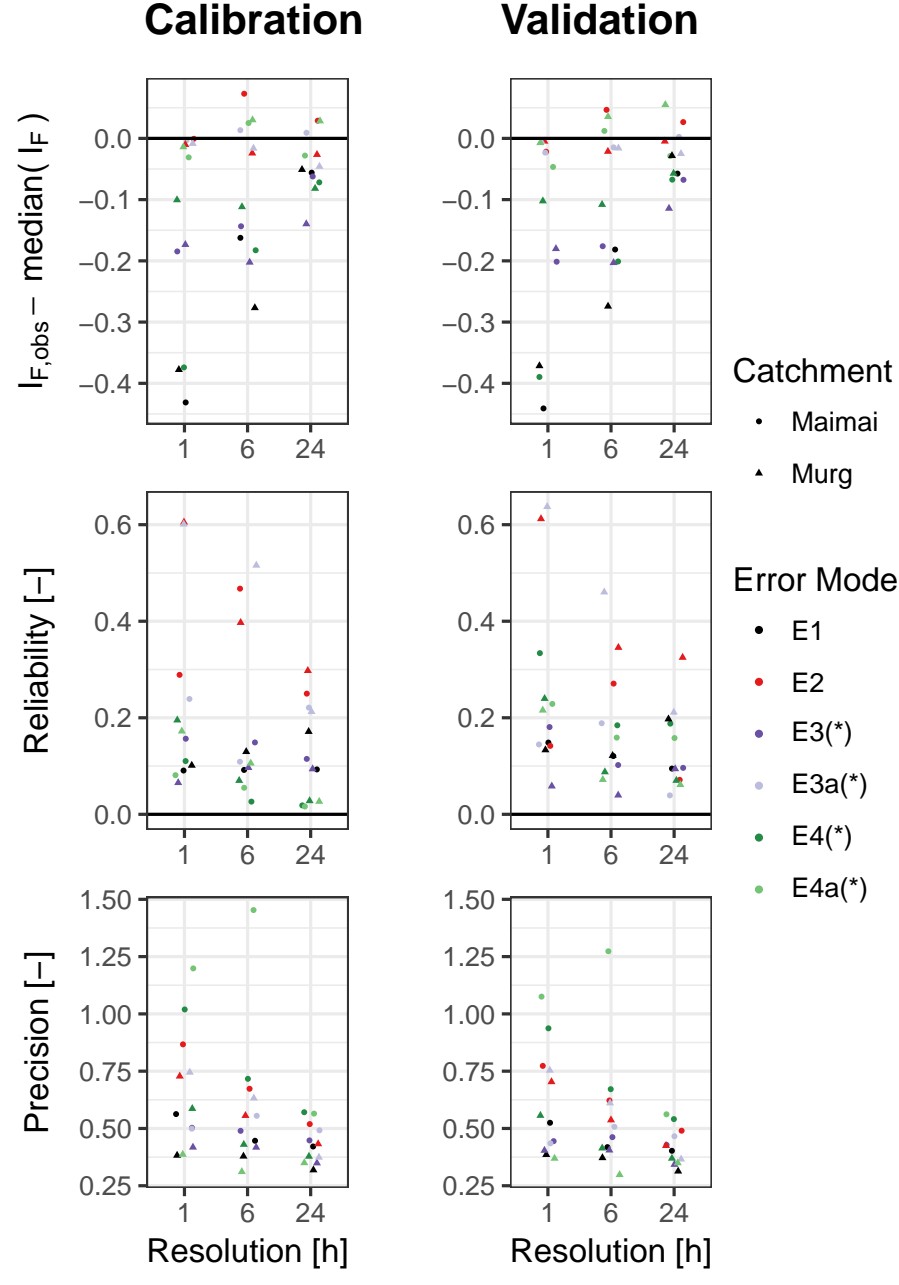

**Figure 3.** Performance of the error models w.r.t. flashiness index, reliability and precision for both catchments and all temporal resolutions. $P_{err}$ was smoothed ($\star$) exclusively for hourly data in the Maimai catchment.




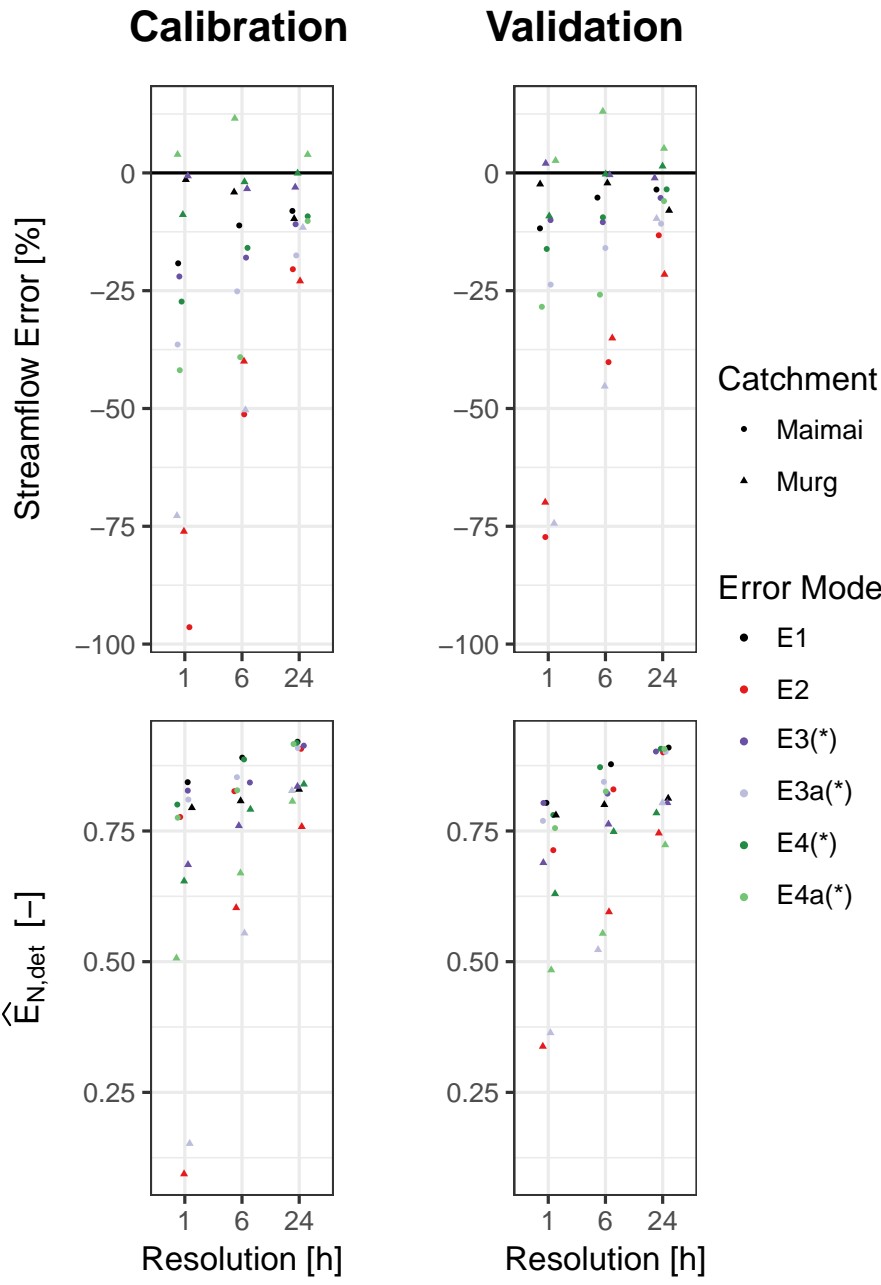

**Figure 4.** Performance of the error models in terms of the relative cumulative error in streamflow, $\Delta_Q$, and the Nash-Sutcliffe Efficiency, $\widehat{E}_{\mathrm{N,det}}$, for both catchments and all temporal resolutions. $P_{\mathrm{err}}$ was smoothed ($\star$) exclusively for hourly data in the Maimai catchment.





### 4.1 Individual error models

#### 4.1.1 Model E1

E1 tends to strongly overestimate the true flashiness in case of high temporal resolutions in both catchments (Fig. 3, the difference between the observed and the median of the predicted Flashiness Index is around -0.4 for both catchments). The

reliability of E1 is never the single best of the error models, but always among the best, and it is robust in light of varying temporal resolution ($\Xi_{\mathrm{reli}}$ is smaller or equal to 0.2 in all the cases, Fig. 3). E1 is also among the error models that provide the most precise predictions (average of 0.41 over all the cases), have the smallest $\Delta_Q$ (usually between 0 and -10 %) and the highest $\widehat{E}_{\mathrm{N,det}}$ overall (Fig. 4). Except for the Flashiness Index, its performance stays stable for high temporal resolutions of the data in both catchments. However, the high Flashiness Index of this model demonstrates the strong violation in the

description of the output behavior despite its good performance regarding the other performance metrics.

#### 4.1.2 Model E2

With the constant correlation assumption made in E2, $I_{\mathrm{F,obs}}$ is generally well reproduced by $I_F$ with deviances ranging from -0.03 to 0.07 (Fig. 3). $\widehat{I}_{\mathrm{F,det}}$ is often similar to $I_F$ for E2 (Tables B1 and B2), indicating that the large part of the flashiness of the model output is due to the hydrological model response and only a small part is due to the stochastic variability added

through the error model. Regarding all the other performance metrics, however, E2 is often among the worst performing error models. For example, in more than half of all the investigated combinations of catchments and temporal resolutions, E2 is the error model with the worst reliability (Fig. 3). E2 has an average precision of 0.61 over all the cases, compared to a precision of 0.41 of E1. It tends to produce large errors in cumulative streamflow, especially in case of hourly resolution ($\Delta_Q < -75\%$, Fig. 4). The degradation of the streamflow error and $\widehat{E}_{\mathrm{N,det}}$ with increasing temporal resolution is very pronounced for E2

compared to the other error models (Fig. 4).

#### 4.1.3 Model E3

E3 generally overestimates the true Flashiness, i.e. $I_F$ is often larger than $I_{\mathrm{F,obs}}$. The difference is around 0.2 for hourly and 6-hourly resolution and a bit less for daily resolution (Fig. 3). The overestimation of the flashiness by E3 is less severe than with E1. E3 results in stable reliability metrics for all temporal resolutions in both catchments: $\Xi_{\mathrm{reli}}$ is smaller than 0.2 in every

case and smaller than 0.1 in more than half of the cases (Fig. 3). In the validation period in the Murg catchment, it is the most reliable error model of all. The precision of E3 is in the range of [0.34,0.5] in all instances with an average value of 0.43, and it is unaffected by the temporal resolution (Fig. 3). The absolute value of $\Delta_Q$ is never larger than 25 % and usually smaller than 10 % (Fig. 4). In terms of $\widehat{E}_{\mathrm{N,det}}$, E3 reaches values larger than 0.75 in all cases except for hourly resolution in the Murg catchment, where it is 0.69. All the metrics show stable performance of E3 under increasing temporal resolution (Figs. 3 and

4).





### 4.1.4  Model E3a

When inferring $\tau_{\min}$ with Error Model E3a, we get close correspondence of $I_F$ and $I_{F,\mathrm{obs}}$ in all cases (Fig. 3, the deviation is never larger than 0.05). In the Maimai catchment, the reliability measure shows a stable performance in, with values between 0.04 and 0.19 in the validation period (Maimai, Fig. 3), showing no clear signs of worse performance for high temporal reso-

lutions. The inferred values of $\tau_{\min}$ were in the order of 1 d and therefore clearly smaller than $\tau_{\max}$ (Fig. 7). Furthermore, $\tau_{\min}$ was consistent among the different temporal resolutions.

In the Murg catchment, on the other hand, we see degenerating performance of E3a with increasing temporal resolution, with values of $\Xi_{\mathrm{reli}} > 0.5$ for 6-hourly and hourly data (Fig. 4), indicating poor performance. All the other metrics show a sim-

ilar pattern (Fig. 4). The inferred $\tau_{\min}$ were between 50 and 100 h, where values on the upper end of the spectrum coincided with bad reliabilities (Fig. 7).

### 4.1.5  Model E4

The stochastic model realisations with E4 tend to overestimate the true Flashiness Index; the difference between $I_{F,\mathrm{obs}}$ and $I_F$ is usually between -0.2 and -0.1 (Fig. 3). $I_F$ is often much larger than $\widehat{I}_{F,\mathrm{det}}$ in the Murg catchment (Table B1), indicating

that a relatively large part of the variability is accounted for by the error model and less by the hydrological model in that case. This manifests in smaller values of $\widehat{E}_{N,\mathrm{det}}$ with E4 compared to E1 (e.g. 0.65 for E4 with hourly resolution compared to 0.79 with E1, Fig. 4). In the Maimai catchment, the hydrological model captures a larger part of the variability than in the Murg catchment, and the difference between $I_F$ and $\widehat{I}_{F,\mathrm{det}}$ is smaller (Table B2). Concerning the reliability, $\Xi_{\mathrm{reli}}$ is largely smaller than 0.2, indicating well-conditioned predictive distributions, except in the validation period for hourly resolution (Fig.

3). In the Maimai catchment, reliability is better in the calibration period compared to the validation period, which is a sign of over-fitting. Especially for daily resolution, E4 provides very good reliabilities in the calibration period ($\Xi_{\mathrm{reli}} < 0.03$, Fig. 3). The average precision of E4 is 0.60. $\Delta_Q$ is not more extreme than -27 % in any case and usually less severe than 20 % (Fig. 4). A slight degradation of $\Delta_Q$ with increasing temporal resolution can be observed.

### 4.1.6  Model E4a

E4a results in $I_F$ that are very close to the observed flashiness in all cases: the difference is never more extreme than 0.05 (Fig. 3). $\widehat{I}_{F,\mathrm{det}}$ is often smaller than $I_{F,\mathrm{obs}}$ in the Murg catchment, which, similar as in E4, is an indication that most of the variability is explained by the error model and not the hydrological model. $\Xi_{\mathrm{reli}}$ is always smaller (better) than 0.2 except for the validation period with hourly resolution in both catchments. Like with E4, we can see a tendency for over-fitting with E4a in the Maimai catchment: in the calibration period, reliabilities of 0.02, 0.05 and 0.08 are reached, while the validation results in values of

0.16, 0.16 and 0.23 for daily, 6-hourly and hourly resolutions, respectively (Table B2). A look at the precision metric (Fig. 3) shows that E4a gives unrealistically large prediction uncertainty in the Maimai catchment for 6-hourly and hourly resolution but that it is among the most precise error models in the Murg catchment. Similarly, E4a produces relatively large errors in





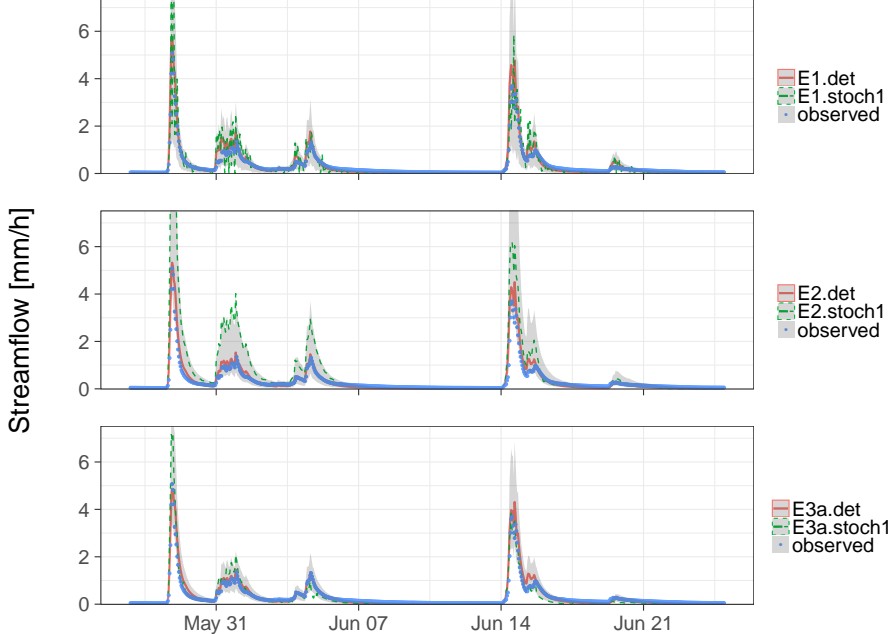

**Figure 5.** Streamflow predictions for the Maimai creek in a part of the validation period (1993). Deterministic predictions with the maximum posterior parameter values are shown together with the 90%-confidence bands and one single stochastic streamflow realisation for each of the error models.

cumulative streamflow in the Maimai catchment, but very small ones in the Murg catchment (Fig. 4). Opposed to that, $\widehat{E}_{\mathrm{N,det}}$ is larger than 0.75 in all cases in the Maimai catchment, while it reaches values as low as 0.5 for hourly resolution in the Murg catchment.

## 4.2 Relaxing the constant-correlation assumption

Error Model E3, which accounts for reduced correlation of errors during the precipitation events, leads to an overall improvement in the investigated performance metrics (except $I_F$) compared to E2, which assumes constant correlation (Fig. 3 and 4). For example, the reliability for hourly resolution in the Murg catchment is 0.06 and 0.61 for E3 and E2, respectively (Fig. 3). In contrast to E2, the performance of E3 does not show systematically worse performance for finer temporal resolution of the data. In fact, E3 and E1 show a similar stability in performance, but E3 provides more realistic estimates of the correlation during recessions and baseflow, leading to a better estimate of $I_F$. Figure 6 shows typical results of E2 and E3 w.r.t. streamflow bias, visible as a bias in $\eta$, and posterior correlation between heteroscedasticity and correlation parameters $a$ and $\tau_{\max}$. Note also the smaller standard deviation (parameter $a$) resulting from E3.



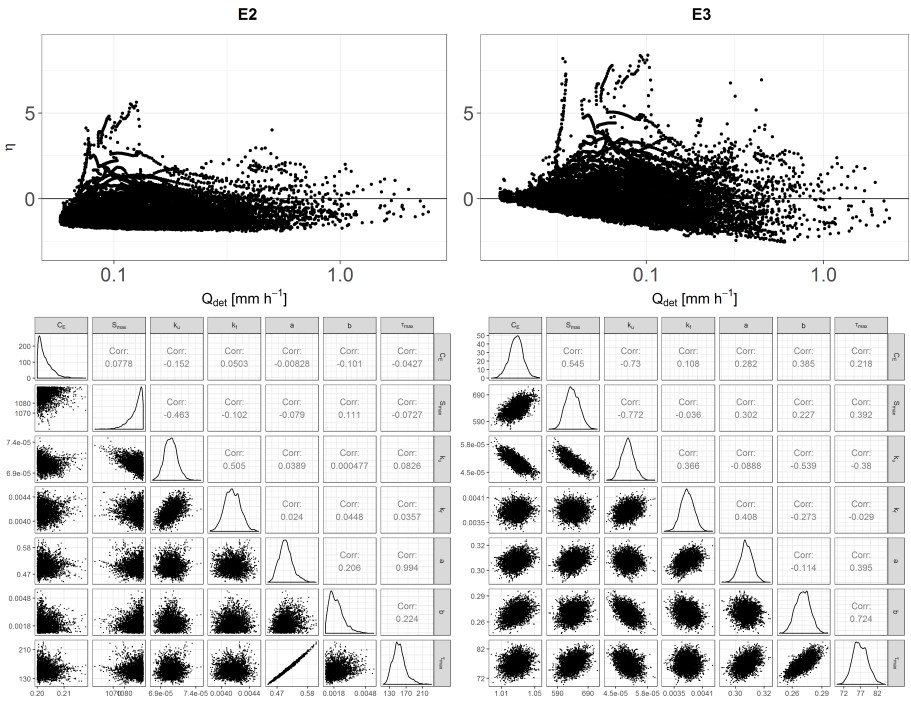

**Figure 6.** Transformed residuals, $\eta$, as a function of modelled streamflow (top) and correlation structure of the posterior parameter sample (bottom) resulting with Error Models E2 (left) and E3 (right) for data with hourly resolution in the Murg catchment.

Figure 5 compares the predicted hydrographs of E1, E2 and E3a. In this case, allowing for different characteristic correlation times during precipitation events and dry periods (E3a) prevents the problematic behaviour encountered when making the constant correlation assumption. Note that E3a results in better estimates of $I_F$ than E3, since it considers correlation during precipitation events ($\tau_{\min} > 0$). In the Murg Catchment, inferring $\tau_{\min}$ resulted in a degenerative performance for high temporal resolutions, which were also linked to higher values of $\tau_{\min}$ (Fig. 7). The posterior estimates of $\tau_{\max}$ depend on the resolution in both catchments. While large $\tau_{\min}$ coincide with the worst reliabilities, large $\tau_{\max}$ were also obtained together with good reliabilities (Fig. 7). The effect of $\tau_{\min}$ on the relative cumulative streamflow error is shown in Fig. 8 for 6-hourly data in the Murg catchment. The streamflow error starts to increase for $\tau_{\min} > 10\,\text{h}$ and at the same time $\widehat{E}_{N,\text{det}}$ decreases (not shown), approaching the one of E2.

## 4.3 Relaxing the assumption of normality

Relaxing the assumption of noramality by inferring $\gamma$ and $df$ (E4 and E4a) had a mixed effect on the numeric performance indices analyzed in this study. When $\tau_{\min} = 0$, including skewness and kurtosis (E4) often led to a better reliability in the calibration period, but a worse reliability in the validation period compared to the assumption of a normal distribution with





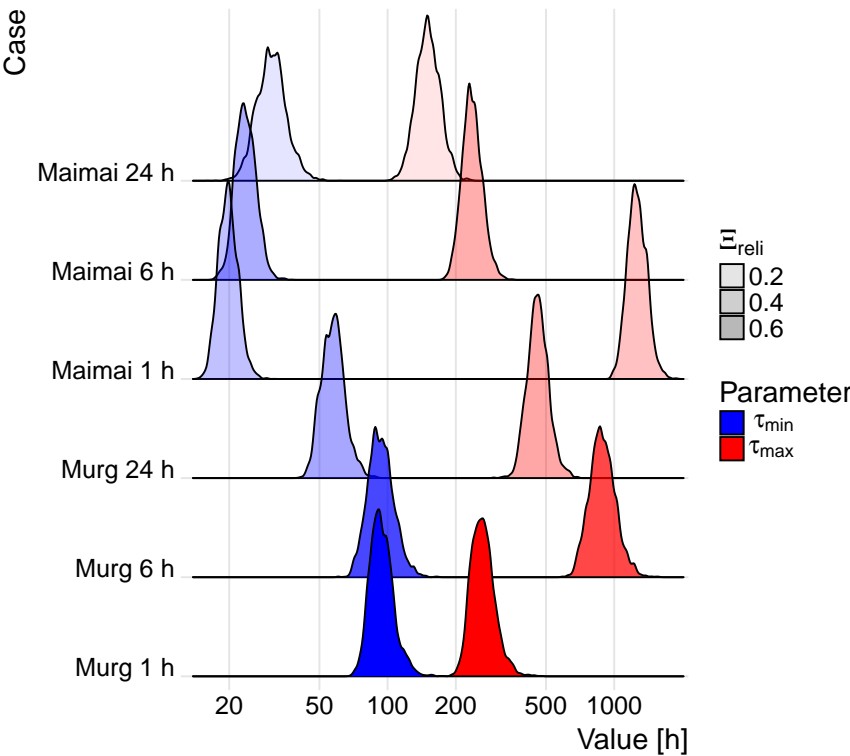

**Figure 7.** Marginal posterior densities of $\tau_{\min}$ and $\tau_{\max}$, and corresponding reliability measures $\Xi_{\mathrm{reli}}$ in the validation period resulting from Error Model E3a in all combinations of catchments and temporal resolutions.

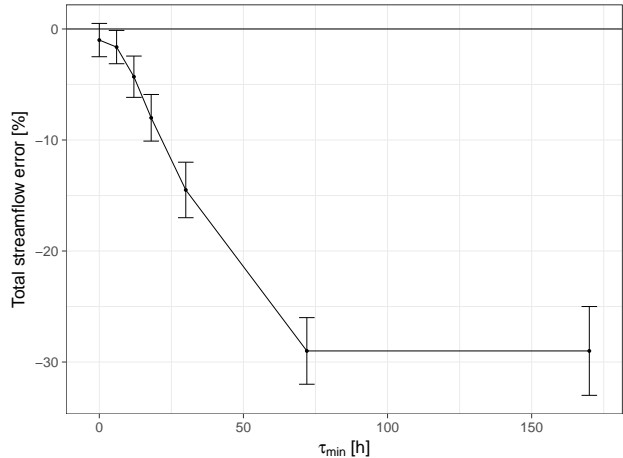

**Figure 8.** Relationship between the fixed correlation time during precipitation events, $\tau_{\min}$, and the total streamflow error, $\Delta_Q$, for 6-hourly data resolution in the Murg catchment. Each point corresponds to a full inference and prediction procedure. The error bars span two standard deviations of 500 stochastic predictions. E3 corresponds to $\tau_{\min} = 0$ and E2 to $\tau_{\min} = \tau_{\max} \approx 170$ h.





E3 (Fig. 3). Predictions with E4 were generally less precise than the ones with E3, e.g. $\Xi_{\mathrm{reli}}$ was around 0.5 with E3 and 1.0 with E4 for hourly resolution in the Maimai catchment (Fig. 3). When $\tau_{\mathrm{min}}$ was inferred additionally, the non-normal case (E4a) showed better performance metrics than the normal case (E3a) in the Murg catchment, but worse ones in the Maimai catchment. E4 and E4a in the Maimai catchment were the only cases that showed a pronounced difference between calibration

and validation, which is a sign of overfitting. A visual inspection of the QQ-plots of $\eta$ revealed that E4 and E4a successfully reduced some very heavy outliers that strongly violated the assumption of normality. In both catchments, the inferred $\gamma$ were in the range of $[1.5, 2.8]$ for E4 and E4a. The values at the upper end of this spectrum were reached for hourly resolutions, and they were associated with underestimation of the peak flows by the deterministic hydrological model, reflected in reduced $\widehat{E}_{\mathrm{N,det}}$. For example, E4a resulted in $\gamma \approx 2.5$, $\widehat{E}_{\mathrm{N,det}} = 0.5$ and an underestimation of preak flows by the hydrological model

for hourly data in the Murg catchment. Inferred $df$ were always at or close to the lower limit of 3, which is indicative of heavy outliers.

Regarding the location of $D_Q$ w.r.t. $Q_{\mathrm{det}}$, the assumption in Eq. (17a) led to better results than Eq. (17b) in the Murg catchment. For example, $\Xi_{\mathrm{reli}}$ with E4a is 0.22 or 0.87 when applying Eq. (17a) or (17b), respectively (Table B1). In the

Maimai catchment, the opposite is true: $\Xi_{\mathrm{reli}}$ is 0.32 or 0.23 with Eq. (17a) or (17b), respectively (Table B2). The difference between results obtained with Eqs. (17a) and (17b) is generally larger for higher temporal resolutions.

## 5  Discussion

### 5.1  Presence and absence of autocorrelation

Assumptions about the presence (E2) and absence (E1) of autocorrelation in $\eta$ were shown to have profound effects on the qual-

ity of the prediction in the cases investigated in this study. Neglecting autocorrelation leads to close correspondence between $\widehat{Q}_{\mathrm{det}}$ and $Q_{\mathrm{obs}}$ in terms of the Nash-Sutcliffe coefficient and to relatively well-fulfilled assumptions about the distribution of $\eta$ in the uniform space (i.e. small values of $\Xi_{\mathrm{reli}}$). However, major assumptions of the underlying statistical model are clearly violated. Most striking is the violation of the zero correlation assumption (Fig. 9), which translates into unrealistic oscillations of the stochastic streamflow predictions (E1 in Fig. 5). Note that E1 also comes with disadvantages related to operational fore-

casts, where one can make more accurate predictions for streamflow in the near future given an error in previous streamflows when accounting for correlated errors (Giudice et al., 2013). This effect was not analyzed in this study.

Accounting for the fact that $\eta$ is obviously autocorrelated, and therefore describing it by a Gaussian process with constant autocorrelation (E2), comes with additional difficulties. Those are: strong interactions between estimates of hydrological water balance parameters and heteroscedasticity and autocorrelation parameters of the error model (E2 in Fig. 6), smaller $E_N$,

$\widehat{E}_{\mathrm{N,det}}$, and worse $\Delta_Q$ compared to E1. Strong posterior correlations between $\tau$ and $a$ coincided with systematic overprediction of streamflow. Evin et al. (2013), who tested an error model similar to E2 on daily data, obtained very similar results in terms of interactions between water balance parameters, heteroscedasticity and correlation parameters. The reasons for those problems are still poorly understood. Failing to reproduce the problems under synthetic conditions, Evin et al. (2014) suggest





that the "nonrobustness of the joint approach" might be caused by "structural errors in the hydrological and / or error models". Based on case studies with daily data, they find that (i) the catchments where these problems are absent are all wet catchments with relatively high runoff coefficients and low ephemerality. To this, we can add that (ii) the performance of the corresponding error model in our study (E2) strongly degrades for finer temporal resolution of the data within two relatively wet catchments.

## 5.2 (Non-)Stationarity of autocorrelation

Figure 9 visualizes one potential reason for the degrading performance of E2 for high-frequency data: our assumptions about the stochastic process (OU-process with constant correlation time $\tau$) seem to be much better fulfilled for the daily than for the hourly data. In the latter case, a visual assessment of $\eta(t)$ obtained with E1, reveals strongly reduced auto-correlation during storms compared to inter-storm periods (Fig. 9). Yang et al. (2007) made similar observations. This raises the hypothesis that the neglection of non-stationarity of the autocorrelation is a major deficit of conventional error models, which leads to the previously encountered problems in the joint inference of autoregressive and hydrological model parameters mentioned in Sect. 5.1.

What is the physical explanation for non-stationary autocorrelation of the errors $\eta$? The autocorrelation of errors in streamflow is primarily caused by the memory effect of errors in storage (Kavetski et al., 2003). Since this memory effect is smaller during periods of rapid change, e.g. during precipitation events, the correlation of the errors in streamflow is expected to be smaller as well during those times. The degree of the reduction of correlation may depend on multiple factors, like the precipitation intensity or volume, the extent to which the precipitation signal is filtered by the catchment, time-lags between precipitation and runoff, and potentially others.

A very simple way of considering this reduced correlation (E3) provides strongly improved results compared to the assumption of stationary correlation (Sect. 4.2). This indicates that neglection of the non-stationarity of the autoregressive parameter is a substantial shortcoming of conventional error models, which causes, at least partly, the well-known problems related to joint inference.

To challenge this hypothesis, one could argue that the improved performance of E3 (compared to E2) might also be achieved when reducing $\tau$ during completely arbitrary time intervals instead of precipitation events. This would dismiss the hypotheses that the precipitation has a direct influence on $\tau$ and that considering this influence leads to a better inference behavior. To test this, we shifted $P_{\mathrm{err}}$ (Eq. 19) substantially in time, so that it would not correspond to the observed precipitation $P$ anymore, while still keeping the major properties (duration and intermittency) of the time intervals during which $\tau$ is reduced. Then, inference was performed with E3 again. The low Nash-Sutcliffe Efficiency and the high streamflow error of the stochastic predictions in that case (E3$^{\dagger}$ in Table B2) shows that it is indeed important to reduce $\tau$ during the precipitation events and not during arbitrary periods with the same intermittency and duration as the precipitation events. With the shifted $P_{\mathrm{err}}$, the resulting $\tau_{\max}$ ($\approx 145\,\mathrm{h}$) was much smaller than the original $\tau_{\max}$ ($\approx 1400\,\mathrm{h}$), confirming the hypothesis of reduced correlation time of errors in streamflow during precipitation events.

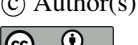



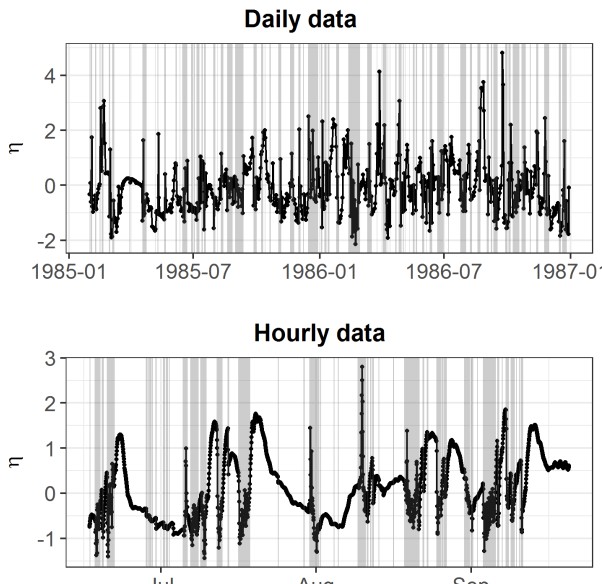

**Figure 9.** Time series of $\eta$ corresponding to the maximum posterior parameter set obtained with E1 in the Maimai catchment for daily resolution (left), and hourly resolution (right). Intervals where $P > 0$ are shaded in grey.

One could also argue that the improved performance of E3 compared to E2 is primarily due to assuming reduced autocorrelation during periods with strong outliers (i.e. storm events) and that those outliers (visible in Fig. (6)) should be accounted for by appropriate values of $\gamma$ and $df$, instead of reducing their influence by neglecting correlation in the periods they appear.

5    Or, similarly said, if the autoregressive process with constant correlation is applied to appropriately standardized residuals, which are marginally normally distributed, it should not cause any problems. To explore this possibility, we performed some experimental analysis for hourly resolution in the Murg catchment: we modified E1 by fixing $\gamma = 1.5$ and $df = 5$ (E1$^+$). This led to a well-conditioned $\eta$ and performance metrics that were comparable to or better than the ones of E1 (Table B1). Then, we inferred $\tau$ under the assumption of constant correlation, while skewness and kurtosis were kept fixed at the values given

10   above (E2$^+$). The resulting performance metrics and a visual assessment of the hydrographs revealed strong deficiencies of this approach compared to E3 and to E1$^+$ (Table B1). This indicates that it is not enough to ensure that the marginal distributions of errors is sufficiently well captured before applying an autoregressive process, but that it is also important to account for a potential non-stationarity of the correlation of the errors. Note that also the distributional parameters of $D_Q$ (e.g. $\gamma$ or $df$) could be non-stationary (Wani et al., In preparation).

It is still unclear what the optimal parametrization of a time-dependent correlation could be. Using the input to directly inform the correlation structure of the output requires knowledge of how the catchment transforms the signal. E.g., there could





be a significant lag time between precipitation and streamflow, which would have to be taken into account in Eq. (19). For the Maimai catchment, we found that using a smoothed version of $P_{\mathrm{err}}$ in Eq. (19) improved the performance of Error Models E3 and E4 in case of hourly resolved data (Table B2). For the coarser resolutions in the Maimai catchment, and for all the tested resolutions in the Murg river, transforming $P_{\mathrm{err}}$ in a similar way did not lead to a remarkable change in the results.

The influence of possible transformations of $P_{\mathrm{err}}$ to account for the filtering effect of the catchment was not systematically investigated in this study.

## 5.3   Inference of $\tau_{\mathrm{min}}$

The fact that $\tau_{\mathrm{min}}$ (Eq. 19) could only be inferred with partial success, shows that there are still problematic interactions among

parameters controlling the correlation of the errors and hydrological model parameters. Figure 7 indicates that those problems are more related to $\tau_{\mathrm{min}}$ than to $\tau_{\mathrm{max}}$, since higher values of $\tau_{\mathrm{min}}$ tend to coincide with bad performance. Or, in more general terms, the previously encountered problems in the joint inference of hydrological and correlation parameters (Evin et al., 2013) seem to originate from precipitation periods, not from dry periods. The fact that the inference of $\tau_{\mathrm{min}}$ is more successful in the Maimai catchment (Sect. 4.1.4), which has the simpler hydrological response, suggests that the realism of a hydrological

model facilitates the succesful inference of the correlation parameters.

These findings call for additional investigations into the issue of non-stationary correlation, potentially exploring other relationships between $\tau$ and $P$ or $Q_{\mathrm{det}}$. Making $\tau$ dependent on $Q_{\mathrm{det}}$ instead of $P$ would have the advantage that potential low-pass filtering or time-lag between precipitation and streamflow are taken care of by the hydrological model and need not

be considered anymore in the error model. We performed some exploratory analysis in that direction, so far with limited success.

## 5.4   Shape of the Distribution $D_Q$

Relaxing the assumption of marginal normality of $Q_{\mathrm{obs}}$ given $Q_{\mathrm{det}}$ successfully reduced some very heavy outliers that strongly

violated that assumption. However, this did not always translate to improved distributional assumptions in the uniform space, where $\Xi_{\mathrm{reli}}$ is calculated. We suspect that the presence of strong outliers (large $\eta$) under the normal assumption led to the strong right-skew of $D_Q$ when inferring $\gamma$ and $df$, which was less appropriate for the rest of the distribution of observed streamflows. In that case, a different distributional shape for $D_Q$ would be more appropriate, e.g. a mixture distribution, that allows for some heavy tails on the upper side without skewing the central body too much to the right. Testing other distributional shapes for

$D_Q$ was beyond the scope of this study, however. Note that heavy outliers (i.e. $\eta \gg 0$) do not necessarily correspond to high streamflow; in both catchments the largest $\eta$ were observed during medium to low flows (Fig. 6), namely during small peaks of observed streamflow that were not captured by the model.





The ranking in performance of the two options to either place the mean or the mode of $D_Q$ at $Q_{\text{det}}$ (Eq. 17), was different for the two analyzed catchments. The previous led to better results in the Murg catchment, while the latter seemed preferable in the Maimai catchment. Ideally, we would like to satisfy both conditions, but this is obviously not possible when $D_Q$ is skewed.

## 6  Conclusions

We presented and evaluated a flexible framework for probabilistic model formulations (i.e. likelihood functions) to describe the total uncertainty of the output of deterministic hydrological models. This framework allows us to consider heteroscedastic errors with non-stationary correlation, non-equidistant observations and zero probability for negative streamflow. It does so by allowing for arbitrary and explicit marginal distributions for the observed streamflow at each point in time. For experts, it is easier to parameterise these marginal streamflow distributions than the distribution characterizing the autoregressive model or some non-intuitive transformations like Box-Cox. The consistent implementation of this framework was successfully checked with a synthetic case study.

Using a simple deterministic hydrological bucket model and two case study catchments, the flexible likelihood framework was used to systematically test different error models on real world data. Those error models represented various assumptions about the statistical properties of the errors in terms of autocorrelation, skewness and kurtosis. The assumptions were found to have a profound effect on the quality of the predictions. The key findings are as follows:

1. We confirmed that, as shown in previous work by various authors, accounting for autocorrelation with conventional approaches (represented by model E2) can lead to worse predictions than omitting autocorrelation (model E1). For example, model E2 had errors in cumulative streamflow of 76 % in the Murg catchment and 96 % in the Maimai catchment for hourly resolution in the calibration period. With model E1, in comparison, those errors were 1 and 19 %, respectively.

2. We showed that the predictions of conventional approaches to deal with autocorrelation worsen significantly as the temporal resolution increases. For example, the performance of model E2 in terms of Nash-Sutcliffe Efficiency goes from 0.76 to 0.09 in the calibration period when moving from daily to hourly data resolution. In comparison, the performance of model E1 remains relatively stable (Nash-Sutcliffe Efficiency goes from 0.83 to 0.79).

3. Since rapid changes in a catchments storage reduce its memory, errors in streamflow are expected to be less correlated during precipitation events than during dry weather. Based on the hypothesis that this non-stationarity increases when going from daily to hourly resolution, neglecting non-stationarity of correlation is the likeli cause for finding 2.

4. Accounting for non-stationarity in autocorrelation significantly alleviated the observed problems of finding 2. In particular, allowing for the autocorrelation to be lower during wet than during dry periods (models E3 and E4) led to more stable behaviour across time resolutions. For example, volume errors for model E3 in the Murg catchment were not larger that





3 % for all three investigated temporal resolutions. However, inferring the characteristic correlation time during precipitation events (model E3a) provided good results in only one of the two investigated catchments. Keeping that correlation fixed (model E3) could be seen as a pragmatic option with stable performance.

5. Accounting for autocorrelation results in more realistic characteristics of model output than omitting autocorrelation, which is confirming previous work. In particular, signatures such as the Flashiness Index are much better represented when including autocorrelation. For example, for an observed value of the Flashiness Index of 0.13 in the Maimai catchment in the calibration period, model E3a provided a value of 0.13, whereas model E1 resulted in a much larger value of 0.56.

6. Inferring the skewness and kurtosis can lead to better fulfilled distributional assumptions about the errors in case of low temporal resolution of the data. For higher resolutions, however, more freedom w.r.t. the shape of the distribution can actually lead to less accurate representation of the observed distribution.

These results contribute to a better characterization of the residual errors of deterministic hydrological models. However, some questions remain. For example, it is still unclear how the non-stationary autocorrelation should ideally be parametrized. The chosen approach, where we alternate between two values of the autoregressive parameter based on whether there is precipitation or not, might lead to problems in catchments with strong lags between precipitation and streamflow. In those cases, defining the autoregressive parameter as a function of modelled streamflow might be more suitable. Furthermore, it could be investigated whether distributions other than the Gaussian and the skewed Student's *t* are more appropriate. Overall, this study confirms previously encountered difficulties in finding a parametrization of an additive error term that adequately describes the effects of intrinsic stochasticity.

## Appendix A: Derivation of the likelihood function

To derive the conditional distribution of $Q(t_{i+1}) \,|\, Q(t_i)$ (and construct the likelihood function by iteratively multiplying the conditional probability densities), we have to propagate the distribution $\eta(t_{i+1}) \,|\, \eta(t_i)$ given by Eq. (4) to the streamflow using the (inverse) transformation $\eta_{trans}$ given by Eq. (3).

In sloppy notation (which makes it easier to get the key idea without getting in notational details), we get:

$$f\big(Q(t_{i+1}) \,|\, Q(t_i)\big) = f\big(\eta(t_{i+1}) \,|\, \eta(t_i)\big) \frac{\mathrm{d}\eta(t_{i+1})}{\mathrm{d}Q(t_{i+1})} = f_{\mathrm{OU}}\big(\eta(t_{i+1}) \,|\, \eta(t_i)\big) \frac{f_{D_Q}\big(Q(t_{i+1})\big)}{f_{\mathrm{N}(0,1)}\big(\eta(t_{i+1})\big)} \tag{A1}$$

where, in the final equation, $f_{\mathrm{OU}}$ refers to the standard Ornstein-Uhlenbeck process defined by Eq. (4) and the ratio of the densities $f_{D_Q}$ and $f_{\mathrm{N}(0,1)}$ results from the derivative and inner derivative of the transformation given by Eq. (3) (the derivative of cumulative distribution functions are the corresponding probability densities).




With explicit notation of functions and arguments, we get

$$f\big(Q(t_{i+1}) \mid Q(t_i), \boldsymbol{\theta}, \boldsymbol{\psi}\big)$$

$$= f\Big(\eta_{\mathrm{trans}}\big(Q(t_{i+1}), Q_{\mathrm{det}}(t_{i+1}, \boldsymbol{\theta}), \boldsymbol{\psi}\big) \mid \eta_{\mathrm{trans}}\big(Q(t_i), Q_{\mathrm{det}}(t_i, \boldsymbol{\theta}), \boldsymbol{\psi}\big)\Big)\frac{\mathrm{d}\eta_{\mathrm{trans}}}{\mathrm{d}Q}\big(Q(t_{i+1}), Q_{\mathrm{det}}(t_{i+1}, \boldsymbol{\theta}), \boldsymbol{\psi}\big)$$

$$= f_{\mathrm{N}\left(\eta_{\mathrm{trans}}\big(Q(t_i), Q_{\mathrm{det}}(t_i, \boldsymbol{\theta}), \boldsymbol{\psi}\big)\exp\left(-\frac{t_{i+1}-t_i}{\tau}\right), \sqrt{1-\exp\left(-2\frac{t_{i+1}-t_i}{\tau}\right)}\right)}\Big(\eta_{\mathrm{trans}}\big(Q(t_{i+1}), Q_{\mathrm{det}}(t_{i+1}, \boldsymbol{\theta}), \boldsymbol{\psi}\big)\Big)$$

$$\cdot \frac{f_{D_Q\big(Q_{\mathrm{det}}(t_{i+1}, \boldsymbol{\theta}), \boldsymbol{\psi}\big)}\big(Q(t_{i+1})\big)}{f_{\mathrm{N}(0,1)}\Big(\eta_{\mathrm{trans}}\big(Q(t_{i+1}), Q_{\mathrm{det}}(t_{i+1}, \boldsymbol{\theta}), \boldsymbol{\psi}\big)\Big)} \quad \text{(A2)}$$

This corresponds to the first sub-equation of Eq. (7). The order of the factors was changed in Eq. (7) to emphasize the product of the marginal distribution $f_{D_Q}$ with a modification facor that tends to unity if $t_{i+1} - t_i$ becomes much larger than $\tau$. The other sub-equations in Eq. (7) consider truncating the streamflow distribution at zero and assigning a point mass corresponding to the integral of the tail below zero to a streamflow of zero.

## Appendix B: Complete results

## Appendix C: Synthetic case study: inferring known true parameters

To check if the implemented likelihood and sampling functions are inverses of each other, we produce a streamflow sample with known parameters according to the procedure outlined in Sect. 2.2. Table C1 shows the results when trying to re-infer those known parameters with the error models presented in this study. In all cases, the true value of the parameters are inside the posterior 95%-confidence intervals.

## Appendix D: Specific error models

### D1  Normal distribution

$$D_Q(\boldsymbol{\xi}) = \mathrm{N}(\mu, \sigma) \quad , \quad \boldsymbol{\xi} = (\mu, \sigma)$$

$$\mu(Q_{\mathrm{det}}) = Q_{\mathrm{det}} \quad , \quad \sigma(Q_{\mathrm{det}}, a, b, c) = aQ_0\left(\frac{Q_{\mathrm{det}}}{Q_0}\right)^c + bQ_0 \quad , \quad \boldsymbol{\psi} = (a, b, c) \tag{D1}$$

$Q_0$ is a chosen constant to make the fraction that is taken to the power of $c$ non-dimensional. A modification of the constant $Q_0$ leads to a re-definition of the parameter $a$. Therefore, introducing the constant $Q_0$ does not increase the number of parameters but it simplifies the units of the parameters $a$ and $b$ that become the same as those of streamflow, whereas $c$ is non-dimensional. Empirical evidence has shown that the normal distribution works astonishingly well. However, there is still as small number of outliers that violate the distributional assumptions relatively strongly. For this reason, a distribution with heavier tails seems appropriate.


**Table B1.** Murg: summary of the predictions in the calibration and the validation period made with error models E1-E4 for different temporal resolutions of the hydrological data. Values are medians (and standard deviations) of the quality indices of the deterministic model output for the maximum posterior parameters, as well as those of 500 streamflow realisations produced with the full posterior parameter distributions. ⋆: smoothing $P_{err}(t)$ with a moving average window of size 5 h before applying Eq. 19. ~denotes the option where mode$(D_Q) = Q_{det}$. $^+$ means that $\gamma = 1.5$ and $df = 5$ was fixed.

| Case | Calibration | | | | | | | | | Validation | | | | | | | | |
| --- | --- | --- | --- | --- | --- | --- | --- | --- | --- | --- | --- | --- | --- | --- | --- | --- | --- | --- |
| | $\Xi_{reli}$ | $\Omega_{prec}$ | $\hat{E}_{N,det}$ | $E_N$ | $\hat{\Delta}_{Q,det}$ [%] | $\Delta_Q$ [%] | $\hat{I}_{F,det}$ | $I_F$ | $I_{F,obs}$ | $\Xi_{reli}$ | $\Omega_{prec}$ | $\hat{E}_{N,det}$ | $E_N$ | $\hat{\Delta}_{Q,det}$ [%] | $\Delta_Q$ [%] | $\hat{I}_{F,det}$ | $I_F$ | $I_{F,obs}$ |
| 24h E1 | 0.17 | 0.32 | 0.83 | 0.68(0.04) | -10 | -10(2.3) | 0.29 | 0.36(0.01) | 0.31 | 0.2 | 0.31 | 0.81 | 0.63(0.04) | -8 | -8(2.4) | 0.29 | 0.36(0.01) | 0.33 |
| 24h E2 | 0.3 | 0.43 | 0.76 | 0.48(0.1) | -23 | -23(5) | 0.29 | 0.34(0.01) | 0.31 | 0.32 | 0.42 | 0.75 | 0.41(0.1) | -21 | -22(5.3) | 0.28 | 0.33(0.01) | 0.33 |
| 24h E3 | 0.09 | 0.35 | 0.84 | 0.65(0.04) | -3 | -3(1.6) | 0.3 | 0.45(0.01) | 0.31 | 0.09 | 0.34 | 0.8 | 0.59(0.04) | -1 | -1(1.7) | 0.29 | 0.44(0.02) | 0.33 |
| 24h E3a | 0.21 | 0.37 | 0.83 | 0.62(0.06) | -11 | -12(3.5) | 0.29 | 0.36(0.01) | 0.31 | 0.21 | 0.37 | 0.8 | 0.56(0.05) | -8 | -10(3.7) | 0.29 | 0.35(0.01) | 0.33 |
| 24h $\widetilde{E4}$ | 0.04 | 0.35 | 0.83 | 0.66(0.1) | 5 | -1(1.7) | 0.27 | 0.41(0.02) | 0.31 | 0.09 | 0.34 | 0.78 | 0.59(0.16) | 6 | 1(1.9) | 0.26 | 0.4(0.02) | 0.33 |
| 24h E4 | 0.03 | 0.38 | 0.84 | 0.65(0.19) | 0 | 0(1.6) | 0.24 | 0.39(0.02) | 0.31 | 0.07 | 0.37 | 0.78 | 0.57(0.1) | 2 | 1(1.8) | 0.23 | 0.38(0.02) | 0.33 |
| 24h $\widetilde{E4}$a | 0.04 | 0.38 | 0.76 | 0.6(0.16) | 16 | -3(3.7) | 0.22 | 0.29(0.01) | 0.31 | 0.15 | 0.38 | 0.64 | 0.48(0.25) | 18 | -2(4) | 0.2 | 0.28(0.01) | 0.33 |
| 24h E4a | 0.03 | 0.35 | 0.81 | 0.66(0.37) | 5 | 4(2.6) | 0.2 | 0.28(0.02) | 0.31 | 0.06 | 0.35 | 0.72 | 0.56(1.4) | 6 | 5(2.8) | 0.19 | 0.27(0.02) | 0.33 |
| 6h E1 | 0.13 | 0.38 | 0.81 | 0.59(0.03) | -4 | -4(0.8) | 0.12 | 0.44(0.01) | 0.16 | 0.12 | 0.37 | 0.8 | 0.57(0.02) | -2 | -2(0.8) | 0.12 | 0.43(0.01) | 0.16 |
| 6h E2 | 0.4 | 0.56 | 0.6 | 0.13(0.15) | -34 | -40(5.7) | 0.14 | 0.18(0) | 0.16 | 0.35 | 0.54 | 0.6 | 0.05(0.14) | -30 | -35(5.7) | 0.14 | 0.18(0) | 0.16 |
| 6h E3 | 0.1 | 0.42 | 0.76 | 0.5(0.04) | -3 | -3(1.4) | 0.15 | 0.36(0.01) | 0.16 | 0.04 | 0.41 | 0.76 | 0.48(0.03) | 0 | 0(1.5) | 0.14 | 0.36(0.01) | 0.16 |
| 6h $\widetilde{E3a}$ | 0.52 | 0.63 | 0.55 | -0.03(0.17) | -41 | -50(7.3) | 0.14 | 0.18(0) | 0.16 | 0.46 | 0.61 | 0.52 | -0.18(0.2) | -36 | -45(7.5) | 0.14 | 0.18(0) | 0.16 |
| 6h $\widetilde{E4}$ | 0.05 | 0.38 | 0.79 | 0.62(0.1) | 7 | -2(1.4) | 0.1 | 0.27(0.01) | 0.16 | 0.1 | 0.37 | 0.74 | 0.56(0.08) | 8 | 0(1.4) | 0.1 | 0.27(0.01) | 0.16 |
| 6h E4 | 0.07 | 0.43 | 0.79 | 0.59(0.1) | -2 | -2(1.4) | 0.08 | 0.27(0.01) | 0.16 | 0.09 | 0.41 | 0.75 | 0.52(0.13) | 0 | 0(1.5) | 0.08 | 0.27(0.01) | 0.16 |
| 6h $\widetilde{E4}$a | 0.05 | 0.4 | 0.63 | 0.51(0.64) | 25 | 2(3.1) | 0.07 | 0.14(0.01) | 0.16 | 0.08 | 0.38 | 0.45 | 0.35(0.19) | 27 | 5(3.2) | 0.06 | 0.13(0.01) | 0.16 |
| 6h E4a | 0.11 | 0.31 | 0.67 | 0.58(0.07) | 12 | 12(2.1) | 0.06 | 0.13(0) | 0.16 | 0.07 | 0.3 | 0.55 | 0.45(0.08) | 13 | 13(2.2) | 0.05 | 0.12(0) | 0.16 |
| 1h E1 | 0.1 | 0.38 | 0.79 | 0.54(0.01) | -1 | -1(0.4) | 0.03 | 0.43(0) | 0.05 | 0.13 | 0.39 | 0.78 | 0.58(0.02) | -2 | -2(0.5) | 0.03 | 0.43(0.01) | 0.06 |
| 1h E1$^+$ | 0.06 | 0.33 | 0.75 | 0.55(0.02) | 14 | -4(0.4) | 0.02 | 0.33(0) | 0.05 | 0.08 | 0.33 | 0.74 | 0.58(0.02) | 12 | -6(0.5) | 0.02 | 0.33(0) | 0.06 |
| 1h E2 | 0.61 | 0.73 | 0.09 | -0.9(0.33) | -61 | -76(7.8) | 0.04 | 0.06(0) | 0.05 | 0.61 | 0.7 | 0.34 | -0.28(0.24) | -56 | -70(8.8) | 0.04 | 0.06(0) | 0.06 |
| 1h E2$^+$ | 0.48 | 0.2 | 0.5 | 0.43(0.05) | 30 | 20(1.8) | 0.01 | 0.05(0) | 0.05 | 0.24 | 0.22 | 0.59 | 0.53(0.05) | 22 | 10(2.3) | 0.01 | 0.05(0) | 0.06 |
| 1h E3 | 0.07 | 0.42 | 0.69 | 0.45(0.02) | 5 | -1(1.3) | 0.04 | 0.22(0) | 0.05 | 0.06 | 0.4 | 0.69 | 0.52(0.02) | 7 | 2(1.6) | 0.04 | 0.24(0.01) | 0.06 |
| 1h E3a | 0.6 | 0.74 | 0.15 | -0.7(0.29) | -61 | -73(8.8) | 0.03 | 0.06(0) | 0.05 | 0.64 | 0.75 | 0.36 | -0.31(0.25) | -56 | -74(10.7) | 0.03 | 0.06(0) | 0.06 |
| 1h E3a* | 0.62 | 0.74 | 0.21 | -0.6(0.26) | -62 | -72(8.8) | 0.03 | 0.06(0) | 0.05 | 0.65 | 0.75 | 0.39 | -0.27(0.22) | -56 | -75(9.4) | 0.03 | 0.06(0) | 0.06 |
| 1h $\widetilde{E4}$ | 0.49 | 0.96 | 0.45 | -0.29(0.26) | 30 | -47(4.3) | 0.01 | 0.18(0) | 0.05 | 0.51 | 0.93 | 0.48 | -0.02(0.23) | 28 | -45(4.9) | 0.01 | 0.19(0) | 0.06 |
| 1h E4 | 0.2 | 0.59 | 0.65 | 0.41(0.26) | -8 | -9(2.5) | 0.01 | 0.15(0) | 0.05 | 0.24 | 0.56 | 0.63 | 0.47(0.07) | -8 | -9(2.5) | 0.01 | 0.16(0) | 0.06 |
| 1h $\widetilde{E4}$a | 0.85 | 1.85 | 0.49 | -5.58(0.97) | 15 | -205(15.1) | 0.01 | 0.07(0) | 0.05 | 0.87 | 1.78 | 0.49 | -3.8(0.83) | 13 | -200(17.1) | 0.01 | 0.08(0) | 0.06 |
| 1h E4a | 0.18 | 0.38 | 0.5 | 0.4(0.03) | 4 | 4(2.1) | 0.01 | 0.06(0) | 0.05 | 0.22 | 0.36 | 0.49 | 0.41(0.03) | 4 | 3(2.2) | 0.01 | 0.06(0) | 0.06 |





**Table B2.** Maimai: summary of the predictions in the calibration and the validation period made with error models E1-E4 for different temporal resolutions of the hydrological data. Values are medians (and standard deviation) of the quality indices of the deterministic model output for the maximum posterior parameters, as well as those of 500 streamflow realisations produced with the full posterior parameter distributions. $^\star$ : smoothing $P_{\mathrm{err}}(t)$ with a moving average window of size 5 h before applying Eq. 19. $\sim$ denotes the option where mode($D_Q$) = $Q_{\mathrm{det}}$ = $Q_{\mathrm{det}}$. $^\dagger$ : $P_{\mathrm{err}} \neq P$.

| Case | Calibration | | | | | | | | | Validation | | | | | | | | |
|---|---|---|---|---|---|---|---|---|---|---|---|---|---|---|---|---|---|---|
| | $\Xi_{reli}$ | $\Omega_{prec}$ | $\hat{E}_{N,det}$ | $E_N$ | $\hat{\Delta}_{Q,det}$ [%] | $\Delta_Q$ [%] | $\hat{I}_{F,det}$ | $I_F$ | $I_{F,obs}$ | $\Xi_{reli}$ | $\Omega_{prec}$ | $\hat{E}_{N,det}$ | $E_N$ | $\hat{\Delta}_{Q,det}$ [%] | $\Delta_Q$ [%] | $\hat{I}_{F,det}$ | $I_F$ | $I_{F,obs}$ |
| 24h E1 | 0.09 | 0.42 | 0.92 | 0.73(0.06) | -8 | -8(3.7) | 0.77 | 0.88(0.03) | 0.83 | 0.09 | 0.4 | 0.91 | 0.70(0.08) | -3 | -4(3.9) | 0.84 | 0.94(0.04) | 0.88 |
| 24h E2 | 0.25 | 0.52 | 0.91 | 0.62(0.1) | -17 | -20(7.4) | 0.74 | 0.8(0.03) | 0.83 | 0.07 | 0.49 | 0.9 | 0.59(0.13) | -11 | -13(7.9) | 0.81 | 0.85(0.03) | 0.88 |
| 24h E3 | 0.11 | 0.45 | 0.91 | 0.70(0.08) | -11 | -11(4.1) | 0.79 | 0.89(0.04) | 0.83 | 0.1 | 0.43 | 0.9 | 0.65(0.1) | -6 | -5(4.5) | 0.87 | 0.95(0.04) | 0.88 |
| 24h E3a | 0.22 | 0.49 | 0.91 | 0.64(0.09) | -16 | -18(6.4) | 0.75 | 0.82(0.03) | 0.83 | 0.04 | 0.47 | 0.9 | 0.62(0.11) | -10 | -11(6.3) | 0.82 | 0.88(0.04) | 0.88 |
| 24h $\widetilde{E4}$ | 0.05 | 0.56 | 0.92 | 0.53(0.24) | -6 | -16(6.2) | 0.81 | 0.93(0.05) | 0.83 | 0.14 | 0.54 | 0.91 | 0.48(0.29) | -1 | -12(6.8) | 0.88 | 0.99(0.05) | 0.88 |
| 24h E4 | 0.02 | 0.57 | 0.92 | 0.6(0.27) | -9 | -9(5.5) | 0.78 | 0.9(0.04) | 0.83 | 0.19 | 0.54 | 0.91 | 0.57(0.33) | -4 | -3(5.7) | 0.85 | 0.95(0.05) | 0.88 |
| 24h $\widetilde{E4}$a | 0.07 | 0.62 | 0.92 | 0.44(0.37) | -5 | -24(8.4) | 0.8 | 0.88(0.04) | 0.83 | 0.07 | 0.63 | 0.91 | 0.37(2.74) | -1 | -19(10.9) | 0.88 | 0.95(0.05) | 0.88 |
| 24h E4a | 0.02 | 0.56 | 0.92 | 0.6(0.25) | -10 | -10(6.6) | 0.77 | 0.85(0.04) | 0.83 | 0.16 | 0.56 | 0.91 | 0.55(0.94) | -5 | -6(7.5) | 0.85 | 0.91(0.05) | 0.88 |
| 6h E1 | 0.09 | 0.45 | 0.89 | 0.69(0.05) | -11 | -11(2.2) | 0.4 | 0.63(0.02) | 0.46 | 0.12 | 0.42 | 0.88 | 0.68(0.05) | -5 | -5(2.5) | 0.45 | 0.65(0.03) | 0.47 |
| 6h E2 | 0.47 | 0.67 | 0.83 | 0.34(0.19) | -37 | -51(9.3) | 0.36 | 0.39(0.01) | 0.46 | 0.27 | 0.62 | 0.83 | 0.37(0.2) | -27 | -40(11.2) | 0.39 | 0.43(0.01) | 0.47 |
| 6h E3 | 0.15 | 0.49 | 0.84 | 0.58(0.07) | -17 | -18(2.8) | 0.45 | 0.61(0.02) | 0.46 | 0.1 | 0.46 | 0.82 | 0.54(0.09) | -10 | -10(3.3) | 0.5 | 0.65(0.03) | 0.47 |
| 6h E3a | 0.11 | 0.56 | 0.85 | 0.5(0.12) | -14 | -25(6) | 0.38 | 0.45(0.02) | 0.46 | 0.19 | 0.51 | 0.84 | 0.49(0.14) | -6 | -16(7) | 0.42 | 0.49(0.02) | 0.47 |
| 6h $\widetilde{E4}$ | 0.08 | 0.59 | 0.89 | 0.52(0.14) | -13 | -14(3.6) | 0.42 | 0.65(0.03) | 0.46 | 0.2 | 0.56 | 0.87 | 0.48(0.21) | -7 | -7(4.2) | 0.46 | 0.67(0.03) | 0.47 |
| 6h E4 | 0.03 | 0.72 | 0.89 | 0.39(0.27) | -15 | -16(4.1) | 0.4 | 0.65(0.03) | 0.46 | 0.18 | 0.67 | 0.87 | 0.39(0.88) | -9 | -9(4.7) | 0.44 | 0.67(0.04) | 0.47 |
| 6h $\widetilde{E4}$a | 0.06 | 0.92 | 0.89 | -0.19(1.76) | -1 | -46(12.5) | 0.38 | 0.47(0.02) | 0.46 | 0.14 | 0.84 | 0.88 | -0.08(1.66) | 5 | -37(12.2) | 0.42 | 0.51(0.03) | 0.47 |
| 6h E4a | 0.05 | 1.45 | 0.83 | -0.45(3.16) | -29 | -39(19.3) | 0.33 | 0.44(0.02) | 0.46 | 0.16 | 1.27 | 0.83 | -0.34(5.2) | -19 | -26(20.6) | 0.36 | 0.46(0.03) | 0.47 |
| 1h E1 | 0.09 | 0.56 | 0.84 | 0.48(0.04) | -19 | -19(1.2) | 0.14 | 0.56(0.01) | 0.13 | 0.15 | 0.52 | 0.8 | 0.41(0.06) | -11 | -12(1.4) | 0.15 | 0.56(0.01) | 0.12 |
| 1h E2 | 0.29 | 0.87 | 0.78 | -1.19(0.73) | -26 | -96(17.2) | 0.12 | 0.13(0) | 0.13 | 0.14 | 0.77 | 0.71 | -1.38(0.92) | -15 | -77(17.1) | 0.13 | 0.14(0) | 0.12 |
| 1h E3 | 0.09 | 0.58 | 0.78 | 0.43(0.04) | -21 | -26(1.9) | 0.13 | 0.38(0.01) | 0.13 | 0.29 | 0.52 | 0.71 | 0.33(0.07) | -11 | -14(2.3) | 0.14 | 0.39(0.01) | 0.12 |
| 1h E3$^\star$ | 0.16 | 0.5 | 0.83 | 0.62(0.02) | -16 | -22(2) | 0.11 | 0.32(0.01) | 0.13 | 0.18 | 0.44 | 0.8 | 0.59(0.04) | -6 | -10(2.2) | 0.12 | 0.32(0.01) | 0.12 |
| 1h E3$^\star$$^\dagger$ | 0.14 | 0.72 | 0.78 | 0.02(0.24) | -30 | -51(6.7) | 0.11 | 0.26(0.01) | 0.13 | 0.14 | 0.65 | 0.73 | -0.14(0.29) | -19 | -41(7.3) | 0.13 | 0.24(0.01) | 0.12 |
| 1h E3a$^\star$ | 0.24 | 0.5 | 0.81 | 0.51(0.1) | -20 | -36(5.8) | 0.13 | 0.13(0) | 0.13 | 0.14 | 0.44 | 0.77 | 0.43(0.15) | -10 | -24(6.2) | 0.14 | 0.15(0) | 0.12 |
| 1h $\widetilde{E4}$ | 0.12 | 0.82 | 0.83 | -0.11(0.46) | -16 | -34(3) | 0.12 | 0.46(0.02) | 0.13 | 0.3 | 0.77 | 0.8 | -0.2(1.21) | -9 | -25(3.7) | 0.13 | 0.47(0.02) | 0.12 |
| 1h $\widetilde{E4}$$^\star$ | 0.07 | 0.63 | 0.86 | 0.31(0.43) | -8 | -24(2.2) | 0.1 | 0.42(0.02) | 0.13 | 0.27 | 0.59 | 0.86 | 0.3(0.56) | -1 | -16(2.6) | 0.11 | 0.43(0.02) | 0.12 |
| 1h E4$^\star$ | 0.11 | 1.02 | 0.8 | -0.17(0.61) | -27 | -27(3.9) | 0.08 | 0.5(0.02) | 0.13 | 0.33 | 0.94 | 0.78 | -0.22(0.93) | -16 | -16(4.8) | 0.08 | 0.51(0.03) | 0.12 |
| 1h $\widetilde{E4}$a$^\star$ | 0.2 | 0.4 | 0.72 | 0.58(0.33) | 18 | 14(3) | 0.1 | 0.12(0) | 0.13 | 0.32 | 0.4 | 0.71 | 0.58(0.27) | 19 | 2(4.8) | 0.11 | 0.12(0.01) | 0.12 |
| 1h E4a$^\star$ | 0.08 | 1.2 | 0.78 | -0.27(1.76) | -35 | -42(13.9) | 0.1 | 0.16(0.01) | 0.13 | 0.23 | 1.08 | 0.76 | -0.32(1.8) | -22 | -28(15.7) | 0.11 | 0.17(0.01) | 0.12 |





**Table C1.** Synthetic case study based on parameter values obtained for the Murg river with daily resolution. ML = maximum likelihood estimation, CI = confidence interval.

| Param. | Unit | True value | E1 ML | E1 95%-CI | E2 ML | E2 95%-CI | E3 ML | E3 95%-CI | E3a ML | E3a 95%-CI | E4 ML | E4 95%-CI | E4a ML | E4a 95%-CI |
|---|---|---|---|---|---|---|---|---|---|---|---|---|---|---|
| $C_E$ | - | 1.03 | 1.043 | 1.027 / 1.063 | 1.02 | 0.98 / 1.07 | 1.02 | 1.0 / 1.04 | 1.04 | 1.007 / 1.075 | 1.03 | 1.004 / 1.058 | 1.04 | 1.0 / 1.06 |
| $S_{max}$ | mm | 345 | 352 | 329 / 370 | 336 | 298 / 373 | 325 | 302 / 346 | 330 | 300 / 365 | 330 | 307 / 356 | 338 | 302 / 368 |
| $k_u$ | $h^{-1}$ | 8.5e-5 | 8.59e-5 | 7.7e-5 / 9.7e-5 | 9.76e-5 | 7.44e-5 / 1.28e-4 | 9.5e-5 | 8.0e-5 / 1.1e-4 | 7.95e-5 | 6.09e-5 / 9.29e-5 | 9.87e-5 | 8.14e-5 / 1.13e-4 | 8.48e-5 | 7.43e-5 / 1.08e-4 |
| $k_f$ | $h^{-1}$ | 7.59e-4 | 8.31e-4 | 7.06e-4 / 9.88e-4 | 8.82e-5 | 7.09e-5 / 1.19e-4 | 8.29e-4 | 6.6e-4 / 9.8e-4 | 6.77e-4 | 5.39e-4 / 7.96e-4 | 8.83e-4 | 6.81e-4 / 1.00e-3 | 7.72e-4 | 6.65e-4 / 1.03e-3 |
| $a$ | - | 2.04 | 2.03 | 1.91 / 2.15 | 1.82 | 1.62 / 2.16 | 1.93 | 1.81 / 2.07 | 2.12 | 1.92 / 2.27 | 1.96 | 1.79 / 2.14 | 1.95 | 1.77 / 2.11 |
| $b$ | - | 2.0e-2 | 2.0e-2 | 1.2e-2 / 3.4e-2 | 0.017 | 0.011 / 0.042 | 1.66e-2 | 4.6e-3 / 3.0e-2 | 2.45e-2 | 1.43e-2 / 4.23e-2 | 3.44e-2 | 1.77e-2 / 5.17e-2 | 1.63e-2 | 1.05e-2 / 2.99e-2 |
| $\tau_{max}$ | h | 393 | - | - | 297 | 255 / 424 | 385 | 335 / 445 | 414 | 347 / 497 | 374 | 326 / 432 | 357 | 300 / 425 |
| $\tau_{min}$ | h | 48 | - | - | - | - | - | - | 47.2 | 41.3 / 57.7 | - | - | 44.6 | 38.6 / 52.7 |
| $\gamma$ | - | 1.2 | - | - | - | - | - | - | - | - | 1.21 | 1.15 / 1.28 | 1.16 | 1.10 / 1.23 |
| $df$ | - | 7.39 | - | - | - | - | - | - | - | - | 6.09 | 3.97 / 10.2 | 9.48 | 5.63 / 18.3 |





## D2  Student t distribution

$$D_Q(\boldsymbol{\xi}) = T_{df,\sigma}(\mu,\sigma,df) \quad , \quad \boldsymbol{\xi} = (\mu,\sigma,df)$$

$$\mu(Q_{\text{det}}) = Q_{\text{det}} \quad , \quad \sigma_{T_{df}} = aQ_0\left(\frac{Q_{\text{det}}}{Q_0}\right)^c + bQ_0 \quad , \quad \boldsymbol{\psi} = (a,b,c)$$

(D2)

The student t distribution with degrees of freedom $df > 2$ is a straightforward candidate with heavier tails that reduces to the normal distribution for $df \to \infty$. Note that we need to rescale the original Student t-distribution, $T(df)$, to the standard deviation $\sigma$, i.e. $T(\sigma,df)$:

$$f_{\mathrm{T}_{df,\sigma}}(x) = \frac{1}{\sigma}\sqrt{\frac{df}{df-2}}\, f_{\mathrm{T}_{df}}\left(\frac{1}{\sigma}\sqrt{\frac{df}{df-2}}\, x\right)$$

(D3)

and

$$F_{\mathrm{T}_{df,\sigma}}(x) = F_{\mathrm{T}_{df}}\left(\frac{1}{\sigma}\sqrt{\frac{df}{df-2}}\, x\right) \qquad .$$

(D4)

Note that the degrees of freedom, $df$, have to be larger than 2 to make the standard deviation finite and allow for rescaling to a given standard deviation, $\sigma$.

## D3  Skewed Student t distribution

$$D_Q(\boldsymbol{\xi}) = \mathrm{sk}_\gamma[\mathrm{T}_{df,\sigma}](Q_{\text{det}},\sigma,df,\gamma) \quad , \quad \boldsymbol{\xi} = (Q_{\text{det}},\sigma,df,\gamma)$$

$$\sigma_{\mathrm{sk}_\gamma[\mathrm{T}_{df,\sigma}]} = aQ_0\left(\frac{Q_{\text{det}}}{Q_0}\right)^c + bQ_0 \quad , \quad \boldsymbol{\psi} = (a,b,c)$$

(D5)

To account for the often encountered case of skewed errors of deterministic hydrological models, we transform the Student t distribution with a generally applicable method of skewing distributions (Fernandez and Steel, 1998). For $\gamma = 1$, the skewed Student t distribution reduces to the conventional Student t distribution. Note that the skewing happens after we rescaled the original Student t-distribution to the standard deviation $\sigma$. The skewing changes the distributions' standard deviation again, thus $\sigma \neq \sigma_{\mathrm{sk}_\gamma[\mathrm{T}_{df,\sigma}]}$. The density and cumulative distribution functions of the skewed rescaled distribution, are:

$$f_{\mathrm{sk}_\gamma[\mathrm{T}_{df,\sigma}]}(x) = \begin{cases} \dfrac{2}{\gamma + \dfrac{1}{\gamma}} f_{\mathrm{T}_{df,\sigma}}(\gamma x) = \dfrac{2}{\gamma + \dfrac{1}{\gamma}}\dfrac{1}{\sigma}\sqrt{\dfrac{df}{df-2}}\, f_{\mathrm{T}_{df}}\left(\dfrac{1}{\sigma}\sqrt{\dfrac{df}{df-2}}\,\gamma x\right) & \text{for } x \leq 0 \\[3em] \dfrac{2}{\gamma + \dfrac{1}{\gamma}} f_{\mathrm{T}_{df,\sigma}}\left(\dfrac{x}{\gamma}\right) = \dfrac{2}{\gamma + \dfrac{1}{\gamma}}\dfrac{1}{\sigma}\sqrt{\dfrac{df}{df-2}}\, f_{\mathrm{T}_{df}}\left(\dfrac{1}{\sigma}\sqrt{\dfrac{df}{df-2}}\,\dfrac{x}{\gamma}\right) & \text{for } x \geq 0 \, . \end{cases}$$

(D6)



and

$$
F_{\mathrm{sk}_\gamma[\mathrm{T}_{df,\sigma}]}(x) = \begin{cases}
\dfrac{2}{1+\gamma^2} F_{\mathrm{T}_{df,\sigma}}(\gamma x) = \dfrac{2}{1+\gamma^2} F_{\mathrm{T}_{df}}\left(\dfrac{1}{\sigma}\sqrt{\dfrac{df}{df-2}}\,\gamma x\right) & \text{for } x \le 0 \\[2em]
\dfrac{1}{1+\gamma^2} + \dfrac{2}{1+\dfrac{1}{\gamma^2}}\left(F_{\mathrm{T}_{df,\sigma}}\left(\dfrac{x}{\gamma}\right) - \dfrac{1}{2}\right) & \\[2em]
\quad = \dfrac{1}{1+\gamma^2} + \dfrac{2}{1+\dfrac{1}{\gamma^2}}\left(F_{\mathrm{T}_{df}}\left(\dfrac{1}{\sigma}\sqrt{\dfrac{df}{df-2}}\,\dfrac{x}{\gamma}\right) - \dfrac{1}{2}\right) & \text{for } x \ge 0\,.
\end{cases}
\tag{D7}
$$

And the mean and the variance of the skewed rescaled distribution are:

$$
\mu_{\mathrm{sk}_\gamma[\mathrm{T}_{df,\sigma}]} = 2\sigma\,\frac{\gamma^2 - \dfrac{1}{\gamma^2}}{\gamma + \dfrac{1}{\gamma}}\,\frac{\sqrt{df(df-2)}}{df-1}\,\frac{\Gamma\left(\dfrac{df+1}{2}\right)}{\sqrt{\pi\,df}\,\Gamma\left(\dfrac{df}{2}\right)}
\tag{D8}
$$

and:

$$
\sigma^2_{\mathrm{sk}_\gamma[\mathrm{T}_{df,\sigma}]} = \frac{\gamma^3 + \dfrac{1}{\gamma^3}}{\gamma + \dfrac{1}{\gamma}}\,\sigma^2 - \mu^2_{\mathrm{sk}_\gamma[\mathrm{T}_{df,\sigma}]}
$$

$$
= \left(\frac{\gamma^3 + \dfrac{1}{\gamma^3}}{\gamma + \dfrac{1}{\gamma}} - 4\left(\frac{\gamma^2 - \dfrac{1}{\gamma^2}}{\gamma + \dfrac{1}{\gamma}}\right)^2\frac{df(df-2)}{(df-1)^2}\frac{\Gamma^2\left(\dfrac{df+1}{2}\right)}{\pi\,df\,\Gamma^2\left(\dfrac{df}{2}\right)}\right)\sigma^2\quad.
\tag{D9}
$$

To shift the distribution we can evaluate

$$
f_{\mathrm{sk}_\gamma[\mathrm{T}_{df,\sigma}]}(x - Q_{\det})
\tag{D10a}
$$

$$
f_{\mathrm{sk}_\gamma[\mathrm{T}_{df,\sigma}]}(x + \mathrm{med}_{\mathrm{sk}_\gamma[\mathrm{T}_{df,\sigma}]} - Q_{\det})
\tag{D10b}
$$

$$
f_{\mathrm{sk}_\gamma[\mathrm{T}_{df,\sigma}]}(x + \mu_{\mathrm{sk}_\gamma[\mathrm{T}_{df,\sigma}]} - Q_{\det})
\tag{D10c}
$$

In these cases, the mode, the median, and the mean are located at $x_0$, respectively.

**Appendix E: Notation**

$P$                 Precipitation used as an input to the hydrological model.





| $P_{\text{err}}$ | Precipitation used as an input to the error model where needed (not to the hydrological model). |
| --- | --- |
| $Q_{\text{det}}(t,\boldsymbol{\theta})$ | Deterministic hydrological model providing streamflow as a function of time, $t$, and hydrological model parameters $\boldsymbol{\theta}$. |
| $\widehat{Q}_{\text{det}}$ | Deterministic hydrological model output corresponding to the parameter vector $\widehat{\boldsymbol{\theta}}$ with maximum posterior probability. |
| $Q_{\text{obs}}(t)$ | Observed streamflow at time $t$. |
| $Q_{\text{trans}}(\eta)$ | Function transforming $\eta$ into streamflow (used to sample from the probabilistic model consisting of the hydrological model and the error model). |
| $D_Q$ | Distribution of observed streamflow at a certain point in time, given the output of the deterministic hydrological model at the same point in time. |
| $\boldsymbol{\theta}$ | Parameters of the deterministic hydrological model, $Q_{\text{det}}$. |
| $\boldsymbol{\psi}$ | Parameters of the error model, including heteroscedasticity and correlation parameters. |
| $\eta$ | Autocorrelated, stochastic process with standard normal asymptotic distribution that serves to describe the autocorrelation of the errors of the deterministic hydrological model. |
| $\tau$ | Characteristic correlation time of the process $\eta$. |
| $\tau_{\min}$ | Minimum value of $\tau$ in the cases where $\tau$ is a function of $P_{\text{err}}$ and therefore of time. |
| $\tau_{\max}$ | Maximum value of $\tau$ in the cases where $\tau$ is a function of $P_{\text{err}}$ and therefore of time. |
| $F_X$ | Cumulative distribution function of the distribution $X$. |
| $f_X$ | Probability density function of the distribution $X$. |
| $E[X]$ | Expected value of the random variable $X$. |
| $\text{N}(\mu,\sigma)$ | Normal distribution with mean $\mu$ and standard deviation $\sigma$. |
| $\text{T}(df,\sigma)$ | Rescaled Student-t distribution with $df$ degrees of freedom and standard deviation $\sigma$. |
| $\text{SKT}(\mu,\sigma,df)$ | Shifted and rescaled skewed Student-t distribution with mean $\mu$, standard deviation $\sigma$, and $df$ degrees of freedom. |
| $F_I$ | The median of the Flashiness Indices (Baker et al., 2004) of all the individual model realisations constituting a sample of model outputs. |
| $\widehat{F}_{\text{I,det}}$ | The Flashiness Index (Baker et al., 2004) of $\widehat{Q}_{\text{det}}$. |
| $F_{\text{I,obs}}$ | The Flashiness Index (Baker et al., 2004) of $Q_{\text{obs}}$. |
| $E_N$ | The median of the Nash-Sutcliffe Indices (Nash and Sutcliffe, 1970) of all the individual model realisations constituting a sample of model outputs. |
| $\widehat{E}_{\text{N,det}}$ | The Nash-Sutcliffe Index (Nash and Sutcliffe, 1970) of $\widehat{Q}_{\text{det}}$. |




$\Delta_Q$      The median of the relative errors in cumulative streamflow of all the individual model realisations constituting a sample of model outputs.

$\widehat{\Delta}_{\mathrm{Q,det}}$      The relative error in cumulative streamflow of $\widehat{Q}_{\mathrm{det}}$.

$\Xi_{\mathrm{reli}}$      Reliability metric (McInerney et al., 2017)

5    $\Omega_{\mathrm{prec}}$      Precision metric (McInerney et al., 2017)

OU-process    Ornstein-Uhlenbeck process (Uhlenbeck and Ornstein, 1930).

*Competing interests.* The authors declare that they have no conflict of interest.

*Acknowledgements.* This study was funded by the Swiss National Science Foundation (grant 200021_163322). The authors thank Me-teoSwiss (Federal Office of Meteorology and Climatology) for the meteorological data concerning the Murg catchment, Massimiliano Zappa

10   for the preprocessing of this data and Jeffrey McDonnell for the hydrological data of the Maimai catchment. Lorenz Ammann thanks Omar Wani for the inspiring discussions and exchange of ideas. Dmitri Kavetski provided valuable feedback on a draft of this paper.





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
