# Peer review of "A likelihood framework for deterministic hydrological models and the importance of non-stationary autocorrelation"

_Hydrology and Earth System Sciences, 2018_

## Short Comment (SC1) · 16 Aug 2018

[revised manuscript text omitted]

---

## Author Comment (AC1) · 24 Aug 2018

The authors would like to thank Mr. Montanari very much for his interest in this work, for reading the discussed paper, and for the very valuable and highly relevant comments that he provided. We will address all of those comments together with other future comments of the reviewers in this revision cycle. Nonetheless, we additionally reply to all of the comments by Mr. Montanari in the following paragraphs. We hope that those replies are satisfactory and we are happy to further discuss any potential misunderstandings of the comments.

Sincerely,

Lorenz Ammann, Peter Reichert and Fabrizio Fenicia

[Figure]

Page1, Line6: Yes, we mean the autocorrelation of the errors. We will include this in the next version.

Page2, Line3: If we understand correctly, there is general agreement on this point, the question is just whether residuals "cannot" be well represented by a normal distribution with constant mean and variance, or whether they "are rarely" well represented by such a distribution. We would prefer to stay with the latter formulation.

Page2, Line6: We agree that model structural errors can also lead to autocorrelation of errors. We will include this in the next version.

Page2, Line16: Yes, this is right, non-negativity is a characteristic of the streamflow, not of the errors. We will correct the wording. We will also adapt the wording in Page3, Line9 based on this.

Page3, Line22: Very valid comment, we did not formulate this clearly enough. In the next version, we will explain more clearly the difference between a parameterization of the innovation of the process describing the errors, and a parameterization of the distribution of streamflow given a model output. Generally speaking, the former approach is less intuitive and it is more difficult to formulate prior knowledge about what shape the distribution of the innovations of a stochastic process have. In particular, as the consequences of the distribution of the innovations on the marginal distributions of streamflow can often not be derived analytically. It is easier for hydrologists to formulate the marginal distribution of streamflow given model outputs, as this is a quantity hydrologists are much more familiar with.

Page4, Line5: Thanks. Yes, we believe that more complex error models in general have the potential to open new avenues for inference in hydrology, but they are still associated with some problems, the origin of which we do not fully understand yet. Therefore, it is important to investigate the potential causes of those problems to suggest improvements.

Page4, Line7: It was not our intention to say that those approaches are "loose" or "empirical". We wanted to say that some weaknesses of the simpler error models are not revealed in common approaches of uncertainty analysis, e.g. simple error models can often provide reasonable error bands. But when other characteristics are of interest, e.g. the flashiness index of the modelled streamflow, more complex error models might be needed. We will adapt the wording accordingly, and include a reference to the suggested paper.

Page5, Line9: We were not aware of that paper and agree that it would be appropriate to include a reference to it.

Page5, Line24: We agree that this paragraph is not formulated clearly enough. "i" is the index of the time points at which the streamflow was observed. We will rephrase the paragraph and try to make it more easily understandable.

Page6, Line11 and subsequent comment: This might be a misunderstanding. The presented method does account for heteroscedasticity and includes a "variance stabilization": the transformation of the observed streamflow is dependent on the modelled streamflow. This means that the streamflow at each time point is transformed by the same family of distributions, but with a different standard deviation accounting for larger errors in high-flows. Figure 6 shows the transformed observed streamflow (eta) as a function of the modelled streamflow. We did not include a formal test for homoscedasticity, but visual assessments indicate successful variance stabilization. The performance measure "reliability" can be seen as a test of normality, since it quantifies the deviance between a normal distribution and the set of transformed observed streamflow. If this was a misunderstanding we recognize that we were not clear enough and will include more explanations in the next version.

---

## Referee Comment (RC1) · J. Vrugt (Referee) · 28 Sep 2018

Review of HESS-submission "A framework for likelihood functions of deterministic hydrological models" by Ammann et al.

Summary: In this paper the authors introduce a parametric framework to residual analysis. This approach leads to formulation of a likelihood function which, with a suitable prior distribution, helps to evaluate the posterior density of nontraditional residual time series, e.g. truncated and subject to various degrees of skew, kurtosis and serial correlation. The framework allows for the use of transient nuisance variables (hyper parameters) to help accommodate so-called non-stationary residual patterns. The framework presented herein differs a bit from the standard likelihood paradigm in that the starting

point is some parametric family of distributions which describes the likelihood of observing the data, Q, given current model output, Qdet. Authors claim that the proposed likelihood function improves probabilistic inference of hydrologic models via MCMC machinery – with a more realistic description of parameter and predictive uncertainty. I enjoyed reading this paper as it combines theory development with practical application. The paper is well written and should be of interest to the readership of HESS. I hope the authors consider the following comments – I believe those will help to further improve the quality of this manuscript. Note, comments appear in order of my reading of the paper.

1. Page 5, Line 9-11. Authors state that most (many) modelers will have an intuitive idea about the probability distributions of the observations for a given model output. I disagree with this assertion. For the sake of my argument, lets follow the hydrologic example as presented in this work. Let's assume that the model simulates a discharge of 20 mm/day. What would be a reasonable expectation of the actual (observed) discharge at that time? 15? 30? I cannot confidently claim that I would know what probability distribution to assume for the observed discharge at that time. Of course, if 20 mm/day is among the largest simulated values, then I would generally expect the dispersion of this supposed distribution to be larger than for a simulated value of 5 mm/day. Yet, this is only the dispersion – I would not really have an idea about the underlying distribution – would I center this distribution on 20 mm/day? Or is my model systematically under or overestimating the data so that I should shift the distribution to higher or lower values, respectively. Of course, for low discharge values I know that the distribution is truncated at zero – and probably has a tail to the right. But then again do I center the distribution on the model simulated value? Or do we shift it up or downward? In other words, I do not agree with the assertion that many modelers will have an intuitive idea what the distribution of the observed discharge would be if the model output is known.

2. Page 5, Line 22-23. The authors refer to Eq. (3) before presenting Eq. (2). Do

not understand why this is done – would think that text can be presented so that Eq. (3) follows first – then followed by Eq. (2). Note, is Eq. (3) needed after all? The right-hand-side of Eq. (3) can be placed at end of Eq. (2) – then the index needs to be fixed.

3. Page 5, Line 27-29: I do not understand the statement that truncation at zero would lead to lighter tails on the lower end. Yes, truncation would move the probability of negative streamflow values to streamflow values larger than zero. In essence, one could then argue that the tail at the right-hand-side may become larger – as the pdf has to integrate to unity. Yet, because of truncation the left tail is essentially gone if simulated streamflow values are close to zero. The wording "lighter tails" may be a bit confusing as the tail is truncated. It is no longer there.

4. Page 5, Eq. (2) – (3) – thus, eta is the normally transformed counterpart of Q – with truncation accounted for?

5. Equation (4) – authors may consider for normal distribution, N, instead $\mathcal{N}(a,b)$, where "a" (mean) is the first term between brackets in Eq. (5) and "b" is the second term in Eq. (4). In text below Eq. (4) authors could then explain that "a" is the mean of the distribution and b is the variance.

6. Eq. (6) – reference should be given.

7. Page 6, Line 12-14. Maybe I am missing something here, but with any other likelihood function one can ignore missing data as well? One simply does not include this particular observation in the likelihood function. The authors may have a point if serial correlation is considered – then this removal is not straightforward as it breaks the AR-error model.

8. Eq. (7) – top line of curly brace may fit on one line if authors define rho = (ti+1 – ti)/tau, and then use rho in the equation – maybe etatrans written as etaT.

9. Then notation – not sure about the guidelines of HESS, but should theta (parameter vector) not be upright-bold instead of italic-bold? Same holds for the nuisance variables, psi.

10. Is notation DQ required or would fQ suffice instead? Then, the text would talk about a distribution of Q – instead of DQ.

11. A limitation of Eq. (4) is that serial correlation at higher-order lags cannot be modelled, right? Unless you specify different "rho's" in Eq. (6) – but this then leads to multiple likelihoods. This limitation should be stated in the text as residuals may exhibit/show residual correlation beyond lag-1.

12. In Eq. (8) how do we compute the first term on the right-hand-side – that is – the likelihood of the zeroth discharge observation (at t0)? Do we assume normality with dispersion of variance/(1-rhoˆ2)?

13. Page 7, Line 12-13: The statement "the likelihood function can be evaluated ana- lytically" is a bit confusing to me. What does the word "analytical" mean in this context? Most other commonly used likelihood functions in the applied (hydrologic) literature are simple to evaluate in practice, right? That means numerically. All that is needed are the model output and the data? What is different in the present context?

14. The authors use the affine invariant ensemble sampler of Foreman and Mackay et al. (2013) to sample the posterior parameter and nuisance variable distribution. The article would benefit from some more background information – that is – algorith- mic settings (number of walkers, the types of moves that are considered, etc.). Note, that this ensemble sampler has many elements in common with the DREAM family of MCMC algorithms – which uses parallel direction and snooker moves. For later work it may be interesting to compare both methods in terms of efficiency – and to evaluate the power and usefulness of the walk, stretch and replacement move. Note, that the ensemble sampler has two important shortcomings; 1) detailed balance requires the use of a relatively large number of walkers (chains) – this is a significant disadvan- tage for higher dimensional problems as each chain needs burn-in before reaching the
target distribution, and 2) the walkers require stepwise updating – this guarantees reversibility but does not make the sampler amenable to distributed computing, wherein each chain is evolved on a different core/node.

15. Equation (10) – the subscript "F" in the flashiness index, should this not be regular font – that is – upright? As "F" is an abbreviation for "flashiness" and not a variable. Same holds for some of the other summary metrics used in this paper, for example the Nash-Sutcliffe efficiency (subscript "N" should be regular = upright font). Note, that on Page, 8, Line 25 correct notation is used for the flashiness index of the deterministic model output.

16. Page 5, Line 24: "maximum posterior parameter values" – this is rather awkward wording as it literally means – the largest posterior parameter values. And it is not clear what this means either as each dimension of the target distribution will have a maximum posterior value – but all these maxima combined are unlikely to make up an actual posterior sample. Instead, what the authors should use is "maximum a-posteriori density (MAP) parameter values" – that is – the parameter values that maximize the product of the prior density and the likelihood.

17. Eq. (15) and (16) list the flux and water balance equations used by the hydrologic model – but equally important what numerical solution method is used to solve these equations? I assume that the authors have used an implicit solution with time-variable integration step? Solution maintains mass balance?

18. Page 12, Line 5: Why are these model parameters held constant? Why are they not part of the inference – this would be much stronger in my view. If held constant, then how does one know the assumed values are reasonable for the catchment of interest? Note, if I look at the equations then m, alpha and beta must have a large impact on the simulated model output. Hence, unless these parameters have a strong physical underpinning I do not see why one would keep them fixed in the present work. Certainly, the values of m, alpha and beta will affect the residual analysis.

19. The authors do not consider highly relevant work by Scharnagl et al. (2015) published in HESS: Inverse modeling of in situ soil water dynamics: accounting for heteroscedastic, autocorrelated, and non-Gaussian distributed residuals. This work also used a Student distribution for the conditional density of the residuals – and combined this with the template function of Fernandez and Steel (1998) to enable treatment of skewed residual distributions. Given the similarities with the work presented in this paper I think it is important for the authors to consider the listed work of Scharnagl et al.

20. Eq. (18) – does this function satisfy the laws of total expectation and total variance? This is a concern not typically addressed in the hydrologic literature – but the paper by Hernandez-Lopez in HESS (2017) makes some important points regarding preservation of expectation and variance of the error model.

21. I am wondering whether readability of the paper would improve if the section on error models is placed directly after the likelihood section. Indeed, the likelihood contains tau – which is then defined (among others) in the error model section.

22. Page 11, Line 16: What has happened to the index time in the formulation of Qdet? It appears on the left-hand side but does not appear on the right-hand side. Also, what are Qs and Qf? These entities are introduced but they are not discussed nor do they appear elsewhere in the paper?

23. At this point I am wondering why the authors are not using the more common terminology of P(.) for prior distribution and L(.|.) for likelihood function.

24. Figure 6 – the values of eta show a strong temporal correlation for error model E2 and E3. Would it be possible to plot, in some way, the decorrelated eta values (with serial correlation removed).

25. In general, it may be useful if the authors include a plot of the marginal posterior distributions of the model parameters and nuisance variables. As it stands it is difficult

to determine which parameters are well defined and which variables are not well defined by inference against the measured data (for one or more error models). In fact, the authors could compute the KL divergence of the prior and posterior distributions for each error model. In any case, it would be good to have insights on how well the parameters and nuisance variables are defined. Do their posterior distributions extend over the entire prior ranges, or are they limit to a small region inside the prior distribution? Note, Figure 6 goes a long way but is difficult to interpret as the matrix plot is rather small and the x-ranges are scaled according to the posterior uncertainty.

26. Figures 3 and 4: I find these results a bit difficult to interpret. The color/symbol coding is not necessarily clear – making it difficult to interpret the findings. I am sure the authors can find a way of plotting from which the main results are directly visible. Then, again, other readers may like to digest this plot.

27. Figure 5: Difficult to see the differences between the three panels. Would it be possible to enlarge the horizontal length of each of the subplots? Right now, the measured data interacts too much with the grey region, particularly when the posterior prediction/simulation uncertainty is small.

28. Note, the authors use the wording "prediction" – one could argue though that what is presented are simulations as the rainfall for the next is assumed known when simulating streamflow values.

29. Page 24, Line 9 – 12: Is this not due in large part because of ignoring the laws of total expectation and total variance? Per my previous comment on this topic.

30. I think a weakness of this paper is that the authors do not compare their findings against another likelihood function. In the introduction section, the authors discuss strength and limitations of previously used/developed likelihood functions – they use this as justification for their own approach. Yet, my own practical experience suggests that a simple AR-1 likelihood would already do quite a reasonable job. This likelihood is easy to include in the present paper. What is more, the authors should consider

the generalized likelihood function – it is argued that this likelihood has a limitation because of the treatment of serial correlation on non-standardized residuals – this is easy to remedy in practice. Then, the argument of analytic tractability I do not really follow (Page 3, Line 22).

31. Would the inference not lead to more realistic results if the authors augment their likelihood with an error model for the rainfall data? This would carry another set of nuisance variables / hyper parameters (depending in large part on the choice of rainfall prior) but make the inference more robust.

32. Just a thought – but is nonstationary the right wording in the present application of the likelihood function? If tau does vary between rainfall and dry periods – but these two values of tau repeat themselves in the future (e.g. are constant) – then one may argue that overall the residual time series is a stationary time series. Tau just differs between rainfall and non-rainfall days.

33. Overall, I think the author should better recognize the highly related work of Scharnagl (2015) published in the same journal (HESS). Indeed, this paper used the Student distribution with the Fernandez and Steel template function for skew.

I hope these comments are useful to further improve the paper, Jasper Vrugt jasper@uci.edu

---

## Referee Comment (RC2) · Anonymous Referee #2 · 6 Oct 2018

This is an interesting well-written paper that revisits some open problems with the statistical characterization of hydrological model residuals (differences between observed and simulated values) in the context of conceptual rainfall-runoff modeling. Specifically, it addresses the issue of accounting for autocorrelation of model residuals, which is known to be troublesome in e.g. semi-arid basins where performance of spatially lumped models often is sub-optimal. The paper shows that similar problems occur in humid basins when the temporal resolution increases from daily to hourly. A novel approach that uses different autocorrelation coefficients for dry and wet periods is shown to yield better probabilistic streamflow predictions compared to the common practice of using a constant autocorrelation coefficient.

Comments:

[Figure]

1. Title and contribution: the title is quite broadly formulated and doesn't really bring out the main novel contribution of the paper, i.e. improved autocorrelation modeling at sub-daily resolutions. In my opinion the proposed likelihood function framework is secondary to this: although it is different from previous approaches, its performance for constant autocorrelation is similar to previous approaches (at least qualitatively - a numerical comparison is not done in the paper), and the novel use of a variable autocorrelation coefficient could also readily be implemented with previous approaches. So it's not entirely clear what we gain from the new framework, even though I do find it quite elegant. If the main selling point is the new likelihood framework then more extensive comparisons (both theoretical and empirical) with existing approaches would be helpful. The proposed framework also has some (conceptual) issues, as discussed in the next point.

2. Section 2.1: the statistical model and corresponding likelihood is based on specifying the density of observed discharge Q conditioned on simulated discharge, Eq. 1. To avoid negative Q values, the density is truncated at zero by removing all probability mass for Q<0 and placing it at Q=0. This deviates from the usual truncation approach, which would scale the entire density by 1/(1-FQ(0)). In fact, the proposed approach results in strange bimodal looking densities with a peak at Q=0 and another at some Q>0; somehow I don't think this is an intuitive model that hydrologists would come up with based on prior knowledge (as suggested on page 5, line 11)! Another consequence of the chosen truncation is that the transformed variables eta in Eq. 2 are also truncated and not Gaussian. This is partially acknowledged on page 5 line 28, but I don't think it's correct that the lower tail of eta will be lighter: there simply will be no lower tail (truncation). Note that these issues could be remedied by adopting the usual truncation approach (scale the entire density) or by using a density with nonnegative support. It's not clear whether these truncation issues matter in practice, perhaps not for the humid basins studied here, but it may matter in drier basins with discharge close to zero.

3. Section 2.3, evaluation criteria: the reliability and precision metrics are counter-intuitive in that smaller values for these metrics indicate better performance. Unreliability and imprecision metrics? Another natural metric to consider is the maximum log likelihood value of each model (perhaps corrected with number of parameters, as in BIC).

4. Section 3.3, error models: the method of Fernandez and Steel (1998) to skew a symmetric density was also used by Schoups and Vrugt (2010), in their case to skew an exponential-power density. It may be appropriate to cite that paper here, especially if that's where you learned about the Fernandez and Steel method.

5. Table 2, page 14, line 1: the E1 model also truncates fQ at zero, which is another difference with maximizing NSE.

6. Figure 6: in the top-right plot for model E3, it's not clear that distributional assumptions for eta are satisfied; there are significant outliers in this plot, and the variance is not constant.

7. Conclusions: finding 5 (accounting for autocorrelation is good) seems to contradict finding 1 (accounting for autocorrelation can be bad); you may need to clarify/reformulate these a bit.

8. Conclusions: finding 3 states that errors in streamflow are expected to be less correlated during precipitation events than during dry weather. Is that always the case though? What about rainfall errors, these could lead to significant bias and correlated errors in simulated streamflow. Also, structural errors in the fast flow component of the model may be (much) larger than in the slow flow component. Perhaps a better, more general, justification for a nonstationary correlation model is to say that the error correlation structure can be expected to differ between wet and dry periods (for various reasons), and then let the data decide whether wet or dry has the larger autocorrelation coefficient.

Edits:

- page 4, line 5, "Understanding...remains poorly understood": remove "understanding".

- Eq. 13, Nash-Sutcliffe formula: change Q to Qobs in the denominator

- page 21, line 12: "normality" has a typo

- figure 9, caption: left/right should be top/bottom

- page 27, line 28: likeli -> likely

- page 28, line 17: "appropriate" has a typo

---

## Author Response (AR1)

**Author's response**

Note that all the references to the manuscript (made by referees and by authors) refer to the original version of the manuscript, not the revised markup version contained in this file.

**RC:** Referee comment

AC: Author comment

Changes made to the manuscript are highlighted in italic font.

**Reply to Referee #1, Jasper Vrugt**

J. Vrugt: Summary: In this paper the authors introduce a parametric framework to residual analysis. This approach leads to formulation of a likelihood function which, with a suitable prior distribution, helps to evaluate the posterior density of nontraditional residual time series, e.g. truncated and subject to various degrees of skew, kurtosis and serial correlation. The framework allows for the use of transient nuisance variables (hyper parameters) to help accommodate so-called non-stationary residual patterns. The framework presented herein differs a bit from the standard likelihood of observing the data, Q, given current model output, Qdet. Authors claim that the proposed likelihood function improves probabilistic inference of hydrologic models via MCMC machinery – with a more realistic description of parameter and predictive uncertainty. I enjoyed reading this paper as it combines theory development with practical application. The paper is well written and should be of interest to the readership of HESS. I hope the authors consider the following comments – I believe those will help to further improve the quality of this manuscript. Note, comments appear in order of my reading of the paper.

**AC:** Thanks for this general feedback. We have to clarify that the ability to deal with nonstationary correlation (or other parameters) is independent of the presented likelihood framework (as referee #2 correctly pointed out), and could (and should) also be achieved with other frameworks / methods.

We included a corresponding sentence in Sect. 5.2..

J. Vrugt (1): Page 5, Line 9-11. Authors state that most (many) modelers will have an intuitive idea about the probability distributions of the observations for a given model output. I disagree with this assertion. For the sake of my argument, lets follow the hydrologic example as presented in this work. Let's assume that the model simulates a discharge of 20 mm/day. What would be a reasonable expectation of the actual (observed) discharge at that time? 15? 30? I cannot confidently claim that I would know what probability distribution to assume for the observed discharge at that time. Of course, if 20 mm/day is among the largest simulated values, then I would generally expect the dispersion of this supposed distribution to be larger than for a simulated value of 5 mm/day. Yet, this is only the dispersion – I would not really have an idea about the underlying distribution – would I center this distribution on 20 mm/day? Or is my model systematically under or overestimating the data so that I should shift the distribution to higher or lower values, respectively. Of course, for low discharge values I know that the

distribution is truncated at zero – and probably has a tail to the right. But then again do I center the distribution on the model simulated value? Or do we shift it up or downward? In other words, I do not agree with the assertion that many modelers will have an intuitive idea what the distribution of the observed discharge would be if the model output is known.

**AC (1)**: This is an interesting point of discussion about one of the main motivations for the presented likelihood framework. Interestingly, there is a contrasting opinion of the referee J. Vrugt and the author of a short comment, Alberto Montanari, on exactly this point. We acknowledge that our wording "many modellers will have an intuitive idea about the probability distribution ..." is too strong. We agree that it can be difficult to formulate this distribution of the observed streamflow, as the example of J. Vrugt shows.

**We rephrased the sentence accordingly.**

However, in case we do have at least some idea about the shape of the distribution, the presented framework allows to incorporate this as prior knowledge. If we have no idea about the distribution, the presented framework is still useful, because we can communicate and discuss our assumptions in the space of streamflow (with the corresponding units), as the example in the comment of the referee shows. With previously used approaches to deal with skewness and kurtosis (e.g. Box-Cox transformation, generalized likelihood function) it is more difficult to discuss these assumptions, because they are made in transformed (Box-Cox) or innovation (generalized likelihood) spaces, which are less intuitive for us. Our point was that it is easier for hydrologists (although admittedly still not easy) to discuss the marginal distribution of streamflow (because they have been confronted with deviations of model results from observations for this quantity in the past) rather than Box- Cox parameters or the distributions. In summary, this discussion illustrates the major advantage of the presented framework: that distributional assumptions are transparent and easy to communicate, which means that they can be better discussed and questioned.

We mentioned this shortcoming of previous approaches more explicitly in the paragraph on page 3, line 20. Accordingly, we expanded the first paragraph of Sect. 2.1 to provide a clearer motivation for the presented approach of parameterizing the distribution of streamflow given model output, as compared to transformation approaches (Box-Cox) or probabilistic models formulated in the innovation space (generalized likelihood).

**J. Vrugt (2):** Page 5, Line 22-23. The authors refer to Eq. (3) before presenting Eq. (2). Do not understand why this is done – would think that text can be presented so that Eq. (3) follows first – then followed by Eq. (2). Note, is Eq. (3) needed after all? The right-hand-side of Eq. (3) can be placed at end of Eq. (2) – then the index needs to be fixed.

**AC(2):** We agree to reverse the order of the equations. Equation (3) is very important to introduce the transformation function before it is applied to the actual time series, as this transformation is the key of our concept of introducing autocorrelation for arbitrary marginal discharge distributions.

We reversed the order of Eq.(2) and Eq.(3) and we edited the adjacent paragraphs to provide better explanations of the equations and the idea of the transformation to combine autocorrelation with arbitrary marginal distributions of streamflow.

J. Vrugt (3): Page 5, Line 27-29: I do not understand the statement that truncation at zero would lead to lighter tails on the lower end. Yes, truncation would move the probability of negative streamflow values to streamflow values larger than zero. In essence, one could then argue that the tail at the right-hand-side may become larger – as the pdf has to integrate to unity. Yet, because of truncation the left tail is essentially gone if simulated streamflow values are close to zero. The wording "lighter tails" may be a bit confusing as the tail is truncated. It is no longer there.

**AC(3):** It is true that the negative part of the distribution DQ is truncated at each individual time step, so the negative tail at each time step is no longer there. However, here we refer to the marginal distribution of eta over all time steps, and usually there will be no sharp "cut" visible, since the truncation happens at different values at each time step.

We included a corresponding sentence. We also more clearly discussed that our framework allows for truncation with compensation by increased density for positive values (as described by the referee) or for assigning a finite probability for an observed discharge of zero (as actually done in this study).

**J. Vrugt(4):** Page 5, Eq. (2) - (3) - thus, eta is the normally transformed counterpart of Q – with truncation accounted for?

**AC(4):** Yes, this is exactly right. Together with the changes made regarding comment 3, we hope that this became clearer. Truncation will only be needed if the distributional shape of the discharge extends to negative values. This may not always have to be the case.

**J. Vrugt(5):** Equation (4) – authors may consider for normal distribution, N, instead \mathcal(N)(a,b), where "a" (mean) is the first term between brackets in Eq. (5) and "b" is the second term in Eq. (4). In text below Eq. (4) authors could then explain that "a" is the mean of the distribution and b is the variance.

**AC(5):** We agree that it must be made more explicit that the first term is the mean and the second is the standard deviation.

Rather than introducing two new variables, we stated in the text that the two elements are the mean and the standard deviation.

J. Vrugt(6): Eq. (6) – reference should be given.

AC(6): To clarify the derivation, we replaced the paragraph around Eqs. (5) and (6) by:

Note that for a constant time step  $\Delta t = t_{i+1}-t_i$ , Eq. (4) becomes

 $\eta(t_{i+1})|\eta(t_i) \sim \mathrm{N}\left(\phi\eta(t_i), \sqrt{1-\phi^2}\right)$

$$\phi = \exp\left(-\frac{\Delta t}{\tau}
ight)$$
 or  $\tau = -\frac{\Delta t}{\log(\phi)}$

This is an AR1 process with autoregression coefficient  $\phi$  and white noise variance  $(1 - \phi^2)$ .

J. Vrugt(7): Page 6, Line 12-14. Maybe I am missing something here, but with any other likelihood function one can ignore missing data as well? One simply does not include this particular observation in the likelihood function. The authors may have a point if serial correlation is considered – then this removal is not straightforward as it breaks the AR-error model.

**AC(7):** Yes, we agree. Any likelihood can deal with missing data when neglecting correlation, but it requires more effort with an AR error model. Since we think that considering correlation is important, we think it is necessary that future likelihoods can accommodate both, correlation and missing data (or varying time step sizes) naturally. Our point is that this is particularly simple in the suggested approach as it does not need any changes because there is no underlying assumption of equidistant points in time.

We mentioned this more explicitly in the manuscript.

**J. Vrugt(8):** Eq. (7) – top line of curly brace may fit on one line if authors define rho = (ti+1 - ti)/tau, and then use rho in the equation – maybe etatrans written as etaT.

**AC(8):** We agree that Eq. (7) is not ideally displayed. We prefer to implement the latter proposition of the referee.

We replaced eta\_trans by eta(ti), i.e. we substituted Eq. (2) into Eq. (7). Since the dependence of eta(t\_i) on Q(t\_i) is then not explicit anymore, we added a statement about that dependence and referred to Eq. (2). Making these changes, we realized that "theta" is not properly introduced in this section and the dependence of Qdet on "theta" is not consistently stated. We introduced "theta" and added the dependence at two points in the text. We also realized that Eq. (2) still contained a "xi"-function from a previous notation version and removed it from Eq. (2) and from the Appendix.

J. Vrugt(9): Then notation – not sure about the guidelines of HESS, but should theta (parameter vector) not be upright-bold instead of italic-bold? Same holds for the nuisance variables, psi.

**AC(9):** The current guidelines of HESS are italic bold for vectors, according to the information we have.

**J. Vrugt(10):** Is notation DQ required or would fQ suffice instead? Then, the text would talk about a distribution of Q – instead of DQ.

**AC(10):** This would be a possibility, and it would probably make the equations better readable. However, talking about the "distribution of Q" instead of DQ, would make the text quite a bit

with

longer, since the term appears often. We would prefer to stay with the name DQ, because with think it is overall simpler to read.

**J. Vrugt(11):** A limitation of Eq. (4) is that serial correlation at higher-order lags cannot be modelled, right? Unless you specify different "rho's" in Eq. (6) – but this then leads to multiple likelihoods. This limitation should be stated in the text as residuals may exhibit/show residual correlation beyond lag-1.

AC(11): Yes, we fully agree with this comment.

We included a corresponding statement about Eq. (4) in this version of the manuscript.

**J. Vrugt(12):** In Eq. (8) how do we compute the first term on the right-hand-side – that is – the likelihood of the zeroth discharge observation (at t0)? Do we assume normality with dispersion of variance/(1-rho2)?

**AC(12):** This term is calculated with Eq.(1). We recognize that it is confusing that the index "i" refers to the current time step for which we want to calculate the likelihood in Eq.(8), but that it refers to the time step before the current time in Eq. (7).

We referred explicitly to Eq. (1) and also modified the index "i" in Eq. (7), in the adjacent text, as well as in the Appendix A, so that it has the same meaning as in Eq. (8). We also changed the time index in section 2.3, so that it is consistent with the rest of the manuscript.

J. Vrugt(13): Page 7, Line 12-13: The statement "the likelihood function can be evaluated analytically" is a bit confusing to me. What does the word "analytical" mean in this context? Most other commonly used likelihood functions in the applied (hydrologic) literature are simple to evaluate in practice, right? That means numerically. All that is needed are the model output and the data? What is different in the present context?

**AC(13):** We agree that this is a property shared by most likelihoods formulated on top of a deterministic hydrological model. We wanted to express that our framework still belongs to that class and does not lead to additional numerical effort as e.g. stochastic hydrological models that may require PMCMC or ABC rather than standard MCMC. It was not our intention to state that our model is special in this respect.

We clarified this and replaced the expression "evaluated analytically" with "available in closed form" to make it clearer what we mean here.

J. Vrugt(14): The authors use the affine invariant ensemble sampler of Foreman and Mackay et al. (2013) to sample the posterior parameter and nuisance variable distribution. The article would benefit from some more background information – that is – algorithmic settings (number of walkers, the types of moves that are considered, etc.). Note, that this ensemble sampler has many elements in common with the DREAM family of MCMC algorithms – which uses parallel direction and snooker moves. For later work it may be interesting to compare both methods in terms of efficiency – and to evaluate the power and usefulness of the walk, stretch and replacement move. Note, that the ensemble sampler has two important shortcomings; 1) detailed balance requires the use of a relatively large number of walkers

(chains) – this is a significant disadvantage for higher dimensional problems as each chain needs burn-in before reaching the target distribution, and 2) the walkers require stepwise updating – this guarantees reversibility but does not make the sampler amenable to distributed computing, wherein each chain is evolved on a different core/node.

AC(14): We agree that more background information should be provided on this.

We included the specific settings used for sampling with this ensemble sampler.

We also agree that it would be interesting to compare the performance of the sampler applied in this study and the DREAM samplers in a future study.

J. Vrugt(15): Equation (10) – the subscript "F" in the flashiness index, should this not be regular font – that is – upright? As "F" is an abbreviation for "flashiness" and not a variable. Same holds for some of the other summary metrics used in this paper, for example the Nash-Sutcliffe efficiency (subscript "N" should be regular = upright font). Note, that on Page, 8, Line 25 correct notation is used for the flashiness index of the deterministic model output.

AC(15): This is right, thanks for the notice.

We checked and improved regular versus italics fonts in equations throughout the manuscript. Based on this we found that the vector notation (bold) for streamflow was not consistent in Sect. 2.3. Therefore we introduced the vector notation for time series of streamflow at the beginning of the section and improved the notation of the equations in that section.

J. Vrugt(16): Page 5, Line 24: "maximum posterior parameter values" – this is rather awkward wording as it literally means – the largest posterior parameter values. And it is not clear what this means either as each dimension of the target distribution will have a maximum posterior value – but all these maxima combined are unlikely to make up an actual posterior sample. Instead, what the authors should use is "maximum a-posteriori density (MAP) parameter values" – that is – the parameter values that maximize the product of the prior density and the likelihood.

**AC(16):** We assume that the referee means Page 8, Line 24 instead of Page 5, Line 24. What we mean by this is the single parameter vector that is associated with the largest posterior probability density of all the points in the parameter sample. As we are not referring to marginal posterior densities, this can hardly be misunderstood in the way the referee argues. However it certainly makes sense to add the word "density" to "maximum posterior".

We changed the wording "maximum posterior parameter values" to "parameter values at the maximum posterior density"

**J. Vrugt(17):** Eq. (15) and (16) list the flux and water balance equations used by the hydrologic model – but equally important what numerical solution method is used to solve these equations? I assume that the authors have used an implicit solution with time-variable integration step? Solution maintains mass balance?

**AC(17):** We very much agree with the referee. This information should be provided.

We included a reference to the software used to implement the hydrological model, which also contains information about the numerical integration schemes.

J. Vrugt(18): Page 12, Line 5: Why are these model parameters held constant? Why are they not part of the inference – this would be much stronger in my view. If held constant, then how does one know the assumed values are reasonable for the catchment of interest? Note, if I look at the equations then m, alpha and beta must have a large impact on the simulated model output. Hence, unless these parameters have a strong physical underpinning I do not see why one would keep them fixed in the present work. Certainly, the values of m, alpha and beta will affect the residual analysis.

AC(18): We agree that in principle, it is always desirable to infer more parameters. The mentioned parameters were kept fixed to keep the hydrological model parsimonious. Fixing some of the parameters is commonly done in hydrological bucket models, for example, the widely used GR4J model has 4 parameters that are inferred, which is equal to the number of hydrological parameters we infer in this study, and it has other parameters that are kept fixed, including the parameter that is equivalent to "beta" in this study. "m" can be seen as a smoothing parameter, and m=0.01 means that there is close to full evaporation as long as the reservoir Su is not empty. "alpha=2" was found to lead to reasonable results in both the investigated catchments and was fixed because of its potential interactions with kf. We do admit that we do not know if the fixed values of "beta" and "m" are ideal for the investigated catchments. Since we reached good fits with at least some error models in both catchments, we would argue that the values of "beta" and "m" are proven to be reasonable. Often when applying a hydrological model to a catchment, we do not really know whether the model is perfectly appropriate for that catchment and we cannot infer all the (potentially many) parameters of the model. Also, systematic errors are common in practice, so we do not want to avoid them here by overly complex models. One could argue that this limits the transferability of the results to other, more complex models. One could also argue that we should have tested different hydrological models, more catchments and more temporal resolutions to obtain more generalizable results. However, the focus of this paper is on the method development, which allows only for a limited amount of application case studies and comparisons.

We included the above mentioned explanations as to why those parameters were kept fixed, but we did not additionally include model runs where those parameters are fitted.

J. Vrugt(19): The authors do not consider highly relevant work by Scharnagl et al. (2015) published in HESS: Inverse modeling of in situ soil water dynamics: accounting for heteroscedastic, autocorrelated, and non-Gaussian distributed residuals. This work also used a Student distribution for the conditional density of the residuals – and combined this with the template function of Fernandez and Steel (1998) to enable treatment of skewed residual distributions. Given the similarities with the work presented in this paper I think it is important for the authors to consider the listed work of Scharnagl et al.

**AC(19):** We agree that the work of Scharnagl et al. is related to the topic of this study and we were not aware of it, since it was not published as a final paper in HESS. Their "Likelihood 2" uses a skewed Student t-Distribution, but they use it to describe the probability density of the

innovations, like Schoups and Vrugt (2010), not the probability density of the observed streamflow, as is done in this study. A difference to Schoups and Vrugt (2010) is that Scharnagl et al. (2015) apply the autocorrelated process to the standardized residuals, as the correction suggested by Evin et al. (2013). However, this approach does not give satisfying results in that case. Then, the relevance of "Likelihood 3" in Scharnagl et al. (2015) for predictive application was correctly questioned by one of the referees.

We included a reference to that discussion in the manuscript in the introduction and in Sect. 2.1.

**J. Vrugt(20):** Eq. (18) – does this function satisfy the laws of total expectation and total variance? This is a concern not typically addressed in the hydrological literature – but the paper by Hernandez-Lopez in HESS (2017) makes some important points regarding preservation of expectation and variance of the error model.

**AC(20):** The Law of Total Expectation and the Law of Total Variance are statistical theorems. There is no way of violating them for any correctly formulated probabilistic model. We are formulating a joint probability density of discharge at all observations points in equation (8) conditional on the output of the deterministic model. The choice and parameterization of the discharge distribution does not change the validity of fundamental statistical theorems. For this reason, the consideration of heteroscedasticity by Eq. (18) cannot lead to a violation of the Total Laws. Note that we carefully transform the distribution assumed for "eta" to the distribution of "Q" in equation (7); not doing this carefully could be a potential source of error and could lead to a violation of statistical theorems.

Why do Hernandez-Lopez (2015) state that the fulfillment of statistical theorems must be guaranteed by eliminating parameters from MCMC sampling and calculating them from the other components of the sample point (section 4.4 in their paper)? This argument is based on a fundamental misinterpretation of a statistical equation that is valid, if correctly interpreted. Their derivation of equation (22) resp. (B9) in appendix B demonstrates, that this equation links the parameters  $\alpha$  and  $\kappa$ , the error variance, the discharge variance and expectation for an error model with fixed parameters  $\alpha$  and  $\kappa$  (see equation B5 where this assumption is used). In Bayesian inference,  $\alpha$  and  $\kappa$  become random variables and equation (22) is no longer valid (it would contain a sum of random and non-random variables [the expectation and the variance of a random variable are not random]). Applying this invalid equation is the first problem of their approach. The second problem is that the Laws of total Expectation and Variance are integral equations over a multivariate distribution. They have no meaning for individual sampling points to which they apply them. The full sample will fulfill the statistical theorems as a result of the consistency of the approach and without explicit enforcement.

The more interesting question is whether the expectation of the probabilistic model for a given deterministic model output is equal to this deterministic model output. Our framework makes the formulation of such models possible (e.g. a lognormal distribution with mean equal to the deterministic model output). This seems at the first sight a desirable property of the model as it guarantees mass conservation (if the deterministic model conserves mass). Unfortunately, our

experience with such error model formulations were unsatisfactory. In cases in which the model output is very small, even small observations errors can lead to observations that are orders of magnitude larger than the output of the deterministic model and would thus require an extremely strongly skewed distribution. The consequence of such extremely skewed distributions would be that for each "large observation" a very large number of very small observations would be needed to keep the mean (as these observations cannot be much smaller than a small output of the hydrological model). In our experience, such distributions lead to unsatisfactory fits. Thus, the non-negativity of discharge observations (for non-tidal rivers) makes it practically nearly impossible to keep mass balances at very low discharge if there is a considerable observation error.

We added a paragraph in Sect. 5.4 to mention this problem which may also not have gained sufficient attention in the literature.

As for the Law of Total Expectation and Variance, we felt it unnecessary to state the fulfillment of any laws of probability in the paper as this is a property of any correctly formulated model.

**J. Vrugt(21):** I am wondering whether readability of the paper would improve if the section on error models is placed directly after the likelihood section. Indeed, the likelihood contains tau – which is then defined (among others) in the error model section.

AC(21): We agree with this suggestion.

We changed the order of the sections in the manuscript. In addition, we changed the first sentence of the section on error models slightly, so that it fits better to the new position in the manuscript. The paragraph on the priors, which used to be in the section about error models, was transformed into a separate section (3.3), and some wordings were changed in that paragraph.

**J. Vrugt(22):** Page 11, Line 16: What has happened to the index time in the formulation of Qdet? It appears on the left-hand side but does not appear on the right-hand side. Also, what are Qs and Qf? These entities are introduced but they are not discussed nor do they appear elsewhere in the paper?

**AC(22):** We agree that the arguments "t" and " $\theta$ " should also appear on the right hand side of the equation and that Qs and Qf should be mentioned in the text. They are the fast and the slow flow components of the model, respectively, and are given by Eq. (15) and illustrated in Fig. 1.

We added the arguments "t" and "tau" on the right hand side and included a statement about Qs and Qf in the adjacent text.

**J. Vrugt(23):** At this point I am wondering why the authors are not using the more common terminology of P(.) for prior distribution and L(.|.) for likelihood function.

**AC(23):** Only when the output of the probabilistic model is replaced by the observed data for inference, we obtain the likelihood as a function of the parameters given the observed data. The likelihood function is therefore crucial for inference. It is hardly possible to formulate this function directly. This is why scientists formulate probabilistic models as probability distributions

of outcomes given parameters and only afterwards get the likelihood function by substituting the observations for the outcomes. For this reason, it does not make sense to use L when formulating the probabilistic model. We then preferred to stay with the notation when substituting the observations to avoid unnecessary confusion. We recognize that this distinction was not entirely consistent throughout the manuscript.

We modified the text to more clearly distinguish the terms "probability distribution of observations conditional on parameters" and "likelihood function" (of the parameters) after substituting the observations. This means we changed the wording from "likelihood" to "probabilistic model" in all the places where it does not specifically refer to inference. However, there are some exceptions; we kept the term "likelihood function" in the title and the abstract (even though we refer to the probabilistic model there), to keep the keyword and the connection to the more common terminology that does not distinguish between the two.

**J. Vrugt(24):** Figure 6 – the values of eta show a strong temporal correlation for error model E2 and E3. Would it be possible to plot, in some way, the decorrelated eta values (with serial correlation removed).

**AC(24):** What we could plot is the deviation of eta from its expected value (given the previous eta) as a function of time, which could be interpreted as decorrelated eta values.

We calculated the standardized innovations of eta for the data underlying Figure 6 and included a time series plot of those innovations for both catchments in the supplementary information.

J. Vrugt(25): In general, it may be useful if the authors include a plot of the marginal posterior distributions of the model parameters and nuisance variables. As it stands it is difficult to determine which parameters are well defined and which variables are not well defined by inference against the measured data (for one or more error models). In fact, the authors could compute the KL divergence of the prior and posterior distributions for each error model. In any case, it would be good to have insights on how well the parameters and nuisance variables are defined. Do their posterior distributions extend over the entire prior ranges, or are they limit to a small region inside the prior distribution? Note, Figure 6 goes a long way but is difficult to interpret as the matrix plot is rather small and the x-ranges are scaled according to the posterior uncertainty.

AC(25): We agree that these would be useful plots.

We included the prior and the posterior marginal density plots for all the error models, parameters and temporal resolutions for one catchment in the supplementary information. We also computed the KL-divergence and included that information in the supplementary information, in the form of a figure for both catchments, all error models and all parameters.

**J. Vrugt(26):** Figures 3 and 4: I find these results a bit difficult to interpret. The color/symbol coding is not necessarily clear – making it difficult to interpret the findings. I am sure the authors can find a way of plotting from which the main results are directly visible. Then, again, other readers may like to digest this plot.

AC(26): We agree that the plots are a bit crowded and can be difficult to interpret.

We made these plots more easily interpretable by enlarging the size of the individual panels, by slightly changing the color coding, adding lines for visual reference and removing the jitter.

**J. Vrugt(27):** Figure 5: Difficult to see the differences between the three panels. Would it be possible to enlarge the horizontal length of each of the subplots? Right now, the measured data interacts too much with the grey region, particularly when the posterior prediction/simulation uncertainty is small.

**AC(27):** We enlarged the panels of Figure 5 horizontally. We also adapted the legend of Fig. 5 to include the more precise annotation of E3a\* instead of E3a. In the figure caption, we mention now that this figure is based on hourly resolution. Additionally, we also enlarged the panels of Figures 6 and 9.

J. Vrugt(28): Note, the authors use the wording "prediction" – one could argue though that what is presented are simulations as the rainfall for the next is assumed known when simulating streamflow values.

**AC(28):** We agree that what is input and what is predicted is a matter of systems boundaries. Thus, all predictions are conditional on some inputs. As we are dealing with hydrological and not (also) with climatological models, we still think that prediction should not lead to misunderstandings.

To clarify our system boundaries, we added a statement at the end of Section 2.2 to say that the hydrological model is evaluated for given precipitation and potential evapotranspiration data, also in the prediction period.

**J. Vrugt(29):** Page 24, Line 9 – 12: Is this not due in large part because of ignoring the laws of total expectation and total variance? Per my previous comment on this topic.

**AC(29):** As we are not ignoring the laws of total expectation and of total variance, this cannot be the reason (see our reply to comment 20). When looking at the time series of  $\eta$  in Fig. 9, using a constant autocorrelation time would obviously not be adequate as there are much shorter-term fluctuations during rainfall periods than during recessions. It is also clear from a hydrological point of view that (irregular) rainfall destroys the very strong autocorrelation structure we see during recession periods. The point of non-stationary autocorrelation was also raised by Th. Wöhling as referee comment 5 (Hydrol. Earth Syst. Sci. Discuss., 12, C831–C841, 2015) on the manuscript by Scharnagl et al. (2015) that was mentioned by the referee. This said, it is also clear that non-constant autocorrelation is not the only deficit of our deterministic and probabilistic models and further research is needed to further improve an adequate uncertainty description of hydrological models. However, the consideration of non-constant autocorrelation was a point that, in our view, has not been sufficiently discussed in the hydrological literature so far and we hope to contribute to stimulating this topic.

**J. Vrugt(30):** I think a weakness of this paper is that the authors do not compare their findings against another likelihood function. In the introduction section, the authors discuss strength and limitations of previously used/developed likelihood functions – they use this as justification for their own approach.

Yet, my own practical experience suggests that a simple AR-1 likelihood would already do quite a reasonable job. This likelihood is easy to include in the present paper. What is more, the authors should consider the generalized likelihood function – it is argued that this likelihood has a limitation because of the treatment of serial correlation on non-standardized residuals – this is easy to remedy in practice. Then, the argument of analytic tractability I do not really follow (Page 3, Line 22).

AC(30): The paper does systematically compare multiple likelihood functions. They were all implemented with the same framework, to ensure comparability, but they rest on fundamentally different assumptions. For example, likelihood E2 is a "simple AR-1 likelihood". It is clearly shown in the paper that its performance is very bad in the considered case studies. We see no necessity to test another, similar version of a simple AR1 model. As for the generalized likelihood function, we agree that a comparison with the presented framework would be interesting and useful. However, since both approaches are frameworks with considerable flexibility, a meaningful comparison would require to test a large number of probabilistic models covering a reasonable range of different assumptions with both frameworks. This would go clearly beyond the scope of this study. Since we do not attempt that comparison, we do not argue that the presented framework leads to better results than the generalized likelihood function, but only repeat the concerns that have been raised by Evin et al. (2013) about the generalized likelihood. Then, we do not completely understand what the referee means by "easy to remedy in practice". It is not obvious for us how the shortcomings documented in Evin et al. (2013) could be overcome since this would require a new approach that would have to be theoretically developed and tested with a practical application. As we understand it, what comes closest to the generalized likelihood function, including corrections of the mentioned shortcomings, is the "Likelihood 2" in the submitted manuscript of Scharnagl et al. (2015). There, a heavy-tailed distribution is assumed for the innovations of the stochastic process describing the residuals, as in the generalized likelihood, but the autocorrelated process is applied to the transformed residuals, as suggested by Evin et al. (2013). However, also Scharnagl et al. (2015) obtain heavily biased results when assuming constant autocorrelation in a case where it was not appropriate to assume so. Specifically, we would suspect that the generalized likelihood function, after addressing the concerns of Evin et al. (2013), might also benefit a lot from considering nonstationary correlation, which might lead to similar results as presented in this study. This would certainly be a very interesting potential future study.

We expanded page 3, line 22 and page 5, line 10 by including more explanations about the benefits of specifying the distributional assumptions in the intuitive space of streamflow as compared to the abstract space of transformed residuals or innovations of transformed residuals.

J. Vrugt(31): Would the inference not lead to more realistic results if the authors augment their likelihood with an error model for the rainfall data? This would carry another set of nuisance variables / hyper parameters (depending in large part on the choice of rainfall prior) but make the inference more robust.

**AC(31):** We agree that this is another important aspect for quantifying uncertainty of hydrological models. We consider such approaches, which try to distinguish between different

sources of uncertainty explicitly, as another class of approaches that come with their own benefits and shortcomings. This study intentionally focused on an approach to describe the total uncertainty in a lumped way, which minimizes the number of error model parameters and avoids the potential identifiability problems associated with estimating input errors.

We expanded the sentence on page 2, Line 23 accordingly, mentioning the benefits and shortcomings of explicitly accounting for input uncertainty in more detail.

J. Vrugt(32): Just a thought – but is nonstationary the right wording in the present application of the likelihood function? If tau does vary between rainfall and dry periods – but these two values of tau repeat themselves in the future (e.g. are constant) – then one may argue that overall the residual time series is a stationary time series. Tau just differs between rainfall and non-rainfall days.

**AC(32):** We acknowledge that we chose a very simplistic non-stationary pattern. We would still call it non-stationary because of the high potential we see in relaxing the assumption of stationary autocorrelation in general, preferably also with more complex patterns.

**J. Vrugt(33):** Overall, I think the author should better recognize the highly related work of Scharnagl (2015) published in the same journal (HESS). Indeed, this paper used the Student distribution with the Fernandez and Steel template function for skew.

AC(33): See comment 19.

**Reply to Referee #2**

**RC:** This is an interesting well-written paper that revisits some open problems with the statistical characterization of hydrological model residuals (differences between observed and simulated values) in the context of conceptual rainfall-runoff modeling. Specifically, it addresses the issue of accounting for autocorrelation of model residuals, which is known to be troublesome in e.g. semi-arid basins where performance of spatially lumped models often is sub-optimal. The paper shows that similar problems occur in humid basins when the temporal resolution increases from daily to hourly. A novel approach that uses different autocorrelation coefficients for dry and wet periods is shown to yield better probabilistic streamflow predictions compared to the common practice of using a constant autocorrelation coefficient.

AC: Thank you for this general feedback.

**RC (1):** Title and contribution: the title is quite broadly formulated and doesn't really bring out the main novel contribution of the paper, i.e. improved autocorrelation modeling at sub-daily resolutions. In my opinion the proposed likelihood function framework is secondary to this: although it is different from previous approaches, its performance for constant autocorrelation is similar to previous approaches (at least qualitatively – a numerical comparison is not done in the paper), and the novel use of a variable autocorrelation coefficient could also readily be implemented with previous approaches. So it's not entirely clear what we gain from the new framework, even though I do find it quite elegant. If the main selling point is the new likelihood framework then more extensive comparisons (both theoretical and empirical) with existing approaches would be helpful. The proposed framework also has some (conceptual) issues, as discussed in the next point.

**AC (1)**: The referee correctly points out that the two major elements of the manuscript, the likelihood framework and the variable autocorrelation coefficient are independent of each other. While we do want to stress that the latter can strongly improve the results of the inference procedure, we do not claim that the presented likelihood framework leads to better results than other approaches. For this it is too general, the results achieved with the framework will depend strongly on the assumptions made. It will indeed lead to similar results as previous likelihoods, if the assumptions made are very similar (e.g. constant correlation where it is not appropriate). The major novelty of the framework is the ability to transparently discuss the assumptions about the distribution of streamflow given the model output, as the next comment of the referee illustrates. With previously used approaches like Box-Cox transformations or the generalized likelihood, the assumed distribution of streamflow is often unknown and cannot be efficiently communicated and discussed. We do acknowledge that this benefit is of rather qualitative nature and cannot be illustrated by a quantitative comparison.

We included some more theoretical explanations about the potential benefits of the likelihood framework.

We also agree that the variable autocorrelation coefficient is among the most important novel contributions of the paper.

Therefore, we included it in the title of the paper, which was changed to: "A likelihood framework for deterministic hydrological models and the importance of non-stationary autocorrelation"

**RC(2):** Section 2.1: the statistical model and corresponding likelihood is based on specifying the density of observed discharge Q conditioned on simulated discharge, Eq. 1. To avoid negative Q values, the density is truncated at zero by removing all probability mass for Q<0 and placing it at Q=0. This deviates from the usual truncation approach, which would scale the entire density by 1/(1-FQ(0)). In fact, the proposed approach results in strange bimodal looking densities with a peak at Q=0 and another at some Q>0; somehow I don't think this is an intuitive model that hydrologists would come up with based on prior knowledge (as suggested on page 5, line 11)! Another consequence of the chosen truncation is that the transformed variables eta in Eq. 2 are also truncated and not Gaussian. This is partially acknowledged on page 5 line 28, but I don't think it's correct that the lower tail of eta will be lighter: there simply will be no lower tail (truncation). Note that these issues could be remedied by adopting the usual truncation approach (scale the entire density) or by using a density with nonnegative support. It's not clear whether these truncation issues matter in practice, perhaps not for the humid basins studied here, but it may matter in drier basins with discharge close to zero.

AC(2): This critique is partly based on a misunderstanding resulting from an insufficient discussion in our paper. The intention of our approach was to allow for a finite probability at Q=0 which is important for intermittent rivers and is often poorly reflected by the deterministic part of the model in which the discharge approaches zero only asymptotically. Such a finite probability for Q=0 can be desirable, see e.g. Smith et al. (2010), whose approach of a mixture distribution is in conceptual agreement with Eq.(1). However, truncation and assignment of the truncated mass to Q=0 is only needed if the distribution extends to negative values. As our framework allows for an arbitrary distribution of discharge, we can choose the distribution mentioned by the referee from the beginning (truncate at zero and rescale the density accordingly). In this case there will not be any additional truncation in our probabilistic model and the probability of Q=0 will be zero. We agree with the referee that this may often be the choice of the modeler and this option is fully covered by our framework (which we did not clearly write in the paper so far). In our study we chose the other distribution to illustrate the possibility of having a finite probability for Q=0, as suggested by Smith et al. (2010). This probability distribution can look a bit non-intuitive, but we still believe that e.g. hydrologists working in ephemeral catchments would appreciate having a finite probability for Q=0.

In Section 2.1, we improved the discussion of the flexibility the modeler has in choosing the distribution of discharge, in particular regarding distributing the probability for a negative outcome to all positive values of discharge. We also added a statement concerning the limitation of the truncation and rescaling approach on page 2, Line 9. To clearly distinguish between the two different "truncating" approaches, we reserved the word "truncating" for the approach that includes rescaling of the positive part of the distribution. Thus, the approach presented in the case studies, where the probability of a negative outcome is assigned to zero, is not called "truncation" anymore.

Concerning the truncation of the lower tail, the referee is correct in the statement that there is no lower tail at all anymore for each individual time step. However, when considering the marginal distribution of the etas at all the time steps, there is still a lower tail, since the individual distributions at each time step are each truncated to a different extent at the lower end, which still results in a continuous marginal distribution over all time steps. We admit that this is not mentioned clearly enough in the manuscript.

We complemented page 5, line 28 by mentioning that we mean the marginal distributions of the etas, which still has a lower tail.

**RC(3):** Section 2.3, evaluation criteria: the reliability and precision metrics are counterintuitive in that smaller values for these metrics indicate better performance. Unreliability and imprecision metrics? Another natural metric to consider is the maximum loglikelihood value of each model (perhaps corrected with number of parameters, as in BIC).

**AC(3):** In order to maintain consistency with McInerney et al. (2017), we would like to keep the names "reliability" and "precision". However, we agree that the names can be misleading in this case.

We added 2 arrows in Figure 3, clarifying that smaller reliability and precision values mean better results. We also added corresponding sentences in the captions of Table B1 and B2.

We agree that the maximum loglikelihood value would be another straightforward metric to consider, but is not clear what information we would gain from it that we do not already have in the other measures. It is not very meaningful for practical purposes and it says nothing about the quality of the predictive distribution since it only characterizes the single best model realization. Given also the broad range of measures already included in the study, we would prefer to not include the maximal loglikelihood as a measure.

**RC(4):** Section 3.3, error models: the method of Fernandez and Steel (1998) to skew a symmetric density was also used by Schoups and Vrugt (2010), in their case to skew an exponential-power density. It may be appropriate to cite that paper here, especially if that's where you learned about the Fernandez and Steel method.

**AC(4):** Thank you for pointing this out. We were not aware that Schoups and Vrugt (2010) have already used the approach of Fernandez and Steel (1998). We found it independently in the statistical literature.

We mentioned that already Schoups and Vrugt (2010) have used this skewing approach.

**RC(5):** Table 2, page 14, line 1: the E1 model also truncates fQ at zero, which is another difference with maximizing NSE.

**AC(5):** This is right, the two approaches are different in their assumption about the distribution of streamflow in the range of Q<=0. In any case where Q<0 is not observed (almost always), the negative part of the distribution will not affect inference, and therefore it will not lead to a

different result. However, when there is data of Q=0, the NSE maximization might lead to different results than E1.

We added this statement on page 14, line 1.

**RC(6):** Figure 6: in the top-right plot for model E3, it's not clear that distributional assumptions for eta are satisfied; there are significant outliers in this plot, and the variance is not constant.

**AC(6):** We agree with the statement of the referee. We do not claim that error model E3 results in perfectly fulfilled assumptions. Also the assumption of zero correlation during precipitation events is violated. We think that the major benefit of E3 is a pragmatic trade-off between fulfilling the assumptions to a satisfactory degree and still providing reasonable results in terms of the fit of the hydrological model and the predictive uncertainty.

**RC(7):** Conclusions: finding 5 (accounting for autocorrelation is good) seems to contradict finding 1 (accounting for autocorrelation can be bad); you may need to clarify/reformulate these a bit.

**AC(7)**: We agree that there is some contradiction in those two findings as they are formulated now.

We added a statement in finding 5 saying that the benefit of considering autocorrelation is only useful if the problems mentioned in finding 1 can be avoided.

**RC(8):** Conclusions: finding 3 states that errors in streamflow are expected to be less correlated during precipitation events than during dry weather. Is that always the case though? What about rainfall errors, these could lead to significant bias and correlated errors in simulated streamflow. Also, structural errors in the fast flow component of the model may be (much) larger than in the slow flow component. Perhaps a better, more general, justification for a nonstationary correlation model is to say that the error correlation structure can be expected to differ between wet and dry periods (for various reasons), and then let the data decide whether wet or dry has the larger autocorrelation coefficient.

**AC(8):** We agree with that statement. It is not completely certain a-priori that the rainfall events will have the less correlated errors, although we do believe that this will often be the case.

We added a statement on page 24 line 16 and adapted finding 3 of the conclusions to say that in principle also higher correlation of errors during precipitation events is possible, but we still mention that reduced correlation is more likely.

**RC(Edits):**

- page 4, line 5, "Understanding...remains poorly understood": remove "understanding".

- Eq. 13, Nash-Sutcliffe formula: change Q to Qobs in the denominator
- page 21, line 12: "normality" has a typo
- figure 9, caption: left/right should be top/bottom

- page 27, line 28: likeli -> likely

- page 28, line 17: "appropriate" has a typo

AC(Edits): Thank you for pointing these out. We agree that all of these are errors.

We corrected them in this version of the manuscript.

**Additional Changes**

Page 1, Line 23: Highlighted the special characteristic of observation errors: they only need to be accounted for when doing inference based on observations.

Page 2, Line 16: Based on a comment of Alberto Montanari, the non-negativity is not listed as a separate characteristic anymore, but added to the first point of the list about the non-normality of the residuals.

Page 2, Line 26: More precise wording regarding the treatment of heteroscedasticity in the weighted least squares approach.

Page 2, Line 29: Bibtex Entry was corrected: Del Giudice, D.

Page 4, Line 11: some clarifications w.r.t. the disadvantages of not accounting for autocorrelation in the residuals

Page 4, Line 21 and Line 30: some clarifications regarding the goals of this study

Page 5, Line 14: better wording regarding the application of copulas to access DQ by Wani et al.

Page 8, Line 16: more precise wording regarding the criteria of performance of an error model.

Page 8, Eq. (10): the absolute value operator was missing in the numerator.

Page 10, Eq. (14): replaced the integration with a sum

Page 13, Figure 2: We included the reference modelled streamflow, Qdet, and changed the centering of DQ so that Qdet is equal to the mean of DQ and not the mode. This is more representative of the method applied for generating the results in the paper.

Page 13, Line 15: Since Qobs is a deterministic value, the notation E[] does not make sense, we replaced it with the "bar" to denote the average.

Page 14, Line 12: Short sentence clarifying the need to account for temporal lags between precipitation and streamflow when using E3.

**Additional references**

Hernández-López, M. R. and Francés, F.: Bayesian joint inference of hydrological and generalized error models with the enforcement of Total Laws, Hydrol. Earth Syst. Sci. Discuss., https://doi.org/10.5194/hess-2017-9, 2017.

Scharnagl, B., Iden, S. C., Durner, W., Vereecken, H., and Herbst, M.: Inverse modelling of in situ soil water dynamics: accounting for heteroscedastic, autocorrelated, and non-Gaussian distributed residuals, Hydrol. Earth Syst. Sci. Discuss., 12, 2155-2199, https://doi.org/10.5194/hessd-12-2155-2015, 2015.

[revised manuscript text omitted]

case; it is based on the assumption of uncorrelated heteroscedastic errors with a normal distribution. These assumptions, with the exception of heteroscedasticity and the treatment of  $Q_{obs} = 0$ , are identical to the ones made when e.g. maximising the Nash-Sutcliffe Efficiency, or, equivalently, minimising the squared residuals. Error Model E2 represents a conventional approach of considering autocorrelation. In the case of equally spaced time-steps, it is similar to the error model applied e.g. by

5 Evin et al. (2013), who assume that the rescaled errors follow an AR(1) process with a standard normal marginal distribution. One difference between the two approaches is, again, the treatment of cases where  $Q_{obs} = 0$ . In error model E3, we additionally account for the fact that  $\tau$  might be time-dependent. The following formula for  $\tau$  is used in those cases:

$$\tau(t) = \begin{cases} \tau_{\min} & \text{for } P_{\text{err}}(t) > 0\\ \tau_{\max} & \text{else} \end{cases}$$
(11)

where  $P_{err}$  is the precipitation used as an input for the error model. In E3,  $\tau_{min}$  is fixed at 0, while in E3a, it is fitted.  $P_{err}$  was either equal to the recorded precipitation, P, or, in case of hourly resolution in the Maimai catchment, smoothed with a moving average of window size 5 h. This was done to prevent frequent jumps between  $\tau_{min}$  and  $\tau_{max}$  during precipitation events, and to be more robust w.r.t. potential time lags between observed precipitation and streamflow. Note that, if such time lags were excessively large, they would have to be considered in Eq. (11). Since in the Murg catchment, smoothing did not change the results substantially,  $P_{err} = P$  applies there. Thus, error Model E3a (or E3) can be seen as a mixture of E1 and E2, in the sense

15 that  $\tau$  alternates between periods of high and low (or no) correlation. Finally, E4 relaxes the assumption of normality for  $D_Q$ ; we use a skewed Student's *t*-distribution, inferring the degrees of freedom and the skewness. Again, E4a denotes the version where  $\tau_{min}$  is inferred.

**2.3 Inference and prediction**

Consider that for any practical case of inference or prediction, we will have a finite series of time points of interest  $(t_0, t_1, ..., t_n)$ and a corresponding time series of streamflow  $Q = (Q(t_0), Q(t_1), ..., Q(t_n))$  or, in analogy,  $Q_{det}$  and  $Q_{obs}$ . When performing inference, the parameters of the hydrological model,  $\theta$ , are estimated jointly with the parameters of the error model,  $\psi$ , by evaluating the likelihood function (Eq. 8) according to the following procedure:

- 1. Given a suggested parameter vector  $\theta$ , evaluate the deterministic hydrological model,  $Q_{det}Q_{det}$ , for all time points.
- 2. Using  $\psi$  and  $Q_{\text{det}}Q_{\text{det}}$ , calculate the likelihood in Eq. (8) by substituting the argument Q with the observed streamflow,
- 10

25

 $Q_{\rm obs}$ .

As the likelihood (Eq. 8) can efficiently be evaluated analytically is available in closed form for a given output of the hydrological model, like in many common likelihood functions in hydrology, we do Bayesian inference based on standard MCMC sampling of the posterior. The affine-invariant ensemble sampler by Foreman-Mackey et al. (2013) is used for this purpose. It uses the so-called "stretch move" to propose a new value for a point in parameter space based on other members

15 of the ensemble. The ensemble size consists of 100 walkers in this study and convergence is assessed visually. A full posterior sample consists of 10'000 model evaluations after successful convergence.

For prediction, stochastic realisations of model output are obtained by inverting Eq. (2):

$$Q_{\rm trans}(\eta, Q_{\rm det}, \psi) = F_{D_Q(Q_{\rm det}, \psi)}^{-1} \left( F_{\rm N(0,1)}(\eta) \right) \tag{12}$$

and applying the following procedure to produce a single stochastic streamflow realisation  $Q_i$ :

- 20 1. Randomly draw a parameter vector  $(\boldsymbol{\theta}, \boldsymbol{\psi})_j$  from the posterior sample.
  - 2. Using  $\theta_j$ , evaluate the deterministic hydrological model to obtain  $-Q_{\det,j}$ , for all time points.
  - 3. Using  $\tau_j \in \psi_j$  and Eq. (4), produce a stochastic realisation of an OU-process,  $\eta_j$ , with a standard normal marginal distribution.
  - 4. Use  $Q_{\text{det},j}$  and  $\psi_j$  and  $Q_{\text{det},j}$ , determined in Steps 1 and 2, to transform  $\eta_j$  into a stochastic realisation of streamflow,  $Q_j$ , with Eq. (12).

Note that a simulation with the hydrological model requires some additional input like precipitation and potential evapotranspiration data (Sect. 3.1), which is assumed to be known also for the prediction period. In a synthetic case study, we could successfully verify the consistency of the implemented likelihood and sampling functions (Appendix ?? see supplementary material).

**2.4 Evaluation criteria**

How can the performance of empirical error models, as the ones presented in this study, be quantified? We argue that the performance of an error model in joint inference with a hydrological model should be judged according to following criteria: (a) good reproduction of observed dynamic fluctuations by individual model realizations, (b) good overall predictive <del>distributions and</del>

5 marginal distribution of streamflow (c) small absolute deviance between model output and observations. The Flashiness Index (Sect. 2.4.1) quantifies is an indicator for (a). The reliability and the precision of the predictive distribution (Sect. 2.4.2 and 2.4.3, respectively) are used as an indicator for (b). The Nash-Sutcliffe Efficiency (Sect. 2.4.4) and the relative error in cumulative streamflow (Sect. 2.4.5) cover (c). In addition to those performance metrics, we calculated the Kullback-Leibler divergence (Kullback and Leibler, 1951) of the marginal posterior parameter distributions from the prior according to the method proposed
10 by Boltz et al. (2007).

**2.4.1 Flashiness Index**

The function to calculate the Flashiness Index (Baker et al., 2004),  $I_F$ , is given by:

$$I(\mathbf{Q}) = \frac{\sum_{i=1}^{n} |Q(t_i) - Q(t_{i-1})|}{\sum_{i=1}^{n} Q(t_i)}$$
(13)

where Q = (Q(t1),Q(t2),...,Q(tN)). IF is Q = (Q(t0),Q(t1),...,Q(tD)). Let x̂ denote the quantity x that is related to
the hydrological parameter values at the maximum posterior density. The Flashiness Index is calculated for the observations,
Qobs IF,obs = I(Qobs), the output of the deterministic hydrological modelat the maximum posterior parameter values, Q̂det,
ÎF,det = I(Q̂det), and the individual stochastic realisations of the full predictive distribution of streamflow , Qj. The resulting metrics are denoted as IF,obs, ÎF,det and IF, respectively, where the latter is the median of the flashiness indices of the individual realisations Qj. IF is predictive streamflow sample, IF = median(I(Qj)). IF is sensitive to the amount of autocorrelation in

20 a streamflow time series, as well as the height of the peaks of  $Q_{det}Q_{det}$  (since  $Q_j$  depends on  $Q_{det}$ ).

**2.4.2 Reliability**

Reliability is defined equivalently to McInerney et al. (2017), as:

$$\Xi_{\rm reli} = \frac{2}{n+1} \sum_{i=0}^{n} |F_{Q(t_i)}(Q_{\rm obs}(t_i)) - F_{\Psi}(F_{Q(t_i)}(Q_{\rm obs}(t_i)))|$$
(14)

25 where  $\Psi = \{F_{Q(t_i)}(Q_{obs}(t_i))|i \in \mathbb{N}, i \leq N_t\} \Psi = \{F_{Q(t_i)}(Q_{obs}(t_i))|i \in \mathbb{N}, 0 \leq i \leq n\}$ ,  $F_{\Psi}$  is the empirical cumulative distribution function of  $\Psi$  and  $F_{Q(t_i)}$  is the empirical cumulative distribution function of the predicted streamflow at time  $t_i$ .  $\Xi_{reli}$  can take values in the interval [0,1], where smaller values of  $\Xi_{reli}$  correspond to better, and zero to perfect, reliability. It summarises the deviance of the observations from the predictive distribution over all time points, and the distance is measured in the uniform space. Therefore, the influence of heavy outliers on  $\Xi_{reli}$  is limited.

**2.4.3 Precision**

The precision metric is an indicator for the width of the predictive distributions over all time points, and was proposed by McInerney et al. (2017) as:

$$\Omega_{\text{prec}} = \frac{\sum_{i=0}^{n} \sigma_Q(t_i)}{\sum_{i=0}^{n} Q_{\text{obs}}(t_i)}$$
(15)

5 where  $\sigma_{Q(t_i)} \circ \sigma_Q(t_i)$  is the standard deviation of the predictive distribution at time point  $t_i$  calculated from the ensemble of all stochastic predictions at that point in time.  $\Omega_{\text{prec}} \in \mathbb{R}^+$ , and small values of  $\Omega_{\text{prec}}$  indicate high precision or small predictive uncertainty. The smaller the predictive uncertainty, the better the quality of the underlying model, given that the predictions are not overconfident.

**2.4.4 Nash-Sutcliffe Efficiency**

10 The Nash-Sutcliffe Efficiency (Nash and Sutcliffe, 1970),  $E_N E_{NA}$  (f for function), is defined as:

$$E_{\rm N,f}(\boldsymbol{Q}, \boldsymbol{Q}_{\rm obs}) = 1 - \frac{\sum_{i=0}^{n} (Q(t_i) - Q_{\rm obs}(t_i))^2}{\sum_{i=0}^{n} (Q_{\rm obs}(t_i) - \overline{Q}_{\rm obs})^2}$$
(16)

where  $Q = (Q(t_1), Q(t_2), \dots, Q(t_N))Q = (Q(t_0), Q(t_1), \dots, Q(t_n))$ . It is used in this study to quantify the agreement between  $\hat{Q}_{det}$  and  $Q_{obs}$ , assess the output of the hydrological at the maximum posterior parameter density,  $\hat{E}_{N,det} = E_{N,f}(\hat{Q}_{det}, Q_{obs})$ , as well as between the *j*-th stochastic realisation  $Q_j$  and  $Q_{obs}$ . The two cases are denoted as  $\hat{E}_{N,det}$  and  $E_N$ , respectively, where

15  $E_N$  is the median of the efficiencies of the individual realisations  $Q_J$  the stochastic simulations,  $E_N = \text{median}(E_{N,t}(Q_{j,t},Q_{obs}))$ . It is used as a rough measure of how well two hydrographs correspond to each other, primarily with the goal of identifying very poorly fitting hydrographs. It is known to be sensitive to errors in high flows (Legates and McCabe, 1999), which can be of particular practical interest. Therefore it complements the other measures, which are less informative with respect to errors in high flows.

**20 2.4.5 Relative error in total cumulative streamflow**

As a measure of systematic over- or under-prediction of streamflow, we calculate the relative error in total cumulative streamflow:

$$\Delta(\boldsymbol{Q}, \boldsymbol{Q}_{\rm obs}) = \frac{\sum_{i=0}^{n} Q_{\rm obs}(t_i) - Q(t_i)}{\sum_{i=0}^{n} Q_{\rm obs}(t_i)}$$
(17)

It is calculated w.r.t. the maximum posterior output of the deterministic model;  $\hat{\Delta}_{Q,det} = \Delta_Q(\hat{Q}_{det}, Q_{obs})$  model output based on the parameter values at the maximum posterior density;  $\hat{\Delta}_{Q,det} = \Delta(\hat{Q}_{det}, Q_{obs})$ , as well as for the ensemble of individual stochastic simulations:  $\Delta_Q = \text{median}(\Delta(Q_j, Q_{obs})) \Delta_Q = \text{median}(\Delta(Q_j, Q_{obs}))$ . Note that, contrary to McInerney et al. (2017),  $\Delta_Q$  is the median error of all the individual hydrograph realisations, not the error of the averaged hydrographs average hydrograph.

**3 Case study setup**

**3.1 Catchments and data**

The likelihood probabilistic framework developed in Sect. 2.1 was tested in two case study sites, the Murg and the Maimai eatchmets catchments, which are described in this section. The Murg river flows through a hilly headwater catchment in temperate climate with a size of  $80 \text{ km}^2$  in northeastern Switzerland. Some key hydrological summary statistics are listed in Table 5 2. Land use is predominantly agricultural (50 %), with forested headwaters (30 %) and a considerable part of urban areas (10 %). The mean elevation is 652 m a.s.l., spanning from 466 to 1035 m a.s.l. Streamflow peaks can be quite sharp, especially for small events, in which baseflow conditions are reached again within just a few hours. This is potentially due to impervious areas being drained directly into the river. The data consists of hourly averages of streamflow, precipitation and potential evap-

10 otranspiration from January 1995 to December 2002. Calibration was performed in the first 5 years (Jan 1995-Dec 1999) and validation in the consecutive 3 years (Jan 2000-Dec 2002). Streamflow data is a courtesy of the Swiss Federal Office for the Environment (FOEN). Precipitation and potential evapotranspiration are based on meteorological data (Meteoschweiz, 2018) and were processed by the Swiss Federal Institute for Forest, Snow and Landscape Research (WSL), with the preprocessing tools of PREVAH (Viviroli et al., 2009).

15

The Maimai experimental catchments are a set of small headwater catchments with a long history of hydrological research. They are located on a deeply incised hillslope on the South Island of New Zealand. The area is forested and the climate is considerably more humid than in the Murg catchment (Table 2). The site was chosen for this study due to its homogeneous characteristics and relatively simple hydrological response, which make it very suited for model evaluation and testing (e.g. Seibert and McDonnell (2002)). We use hourly data recorded in 1985-1987 in the M8 experimental catchment, the most 20 intensely studied of the Maimai catchments. It has an area of ca. 7 ha with steep  $(34^{\circ})$  slopes. The reader is referred to Brammer and McDonnell (1996) for a more detailed description of the characteristics of the M8 and the other experimental catchments. This study does not attempt to make a significant contribution to the understanding of the hillslope processes in the Maimai catchment (see McGlynn et al. (2002) for an extensive overview). Calibration was performed based on data from Jan 1985-Dec 1986, and validation during Jan-Dec 1987. The data was kindly provided by Jeffrey McDonnell.

25

While the resolution of the original data was hourly, we produced data sets with 6-hourly and daily resolution by aggregation for both catchments. This setup allows us to systematically investigate the effect of the temporal resolution of the data on the joint inference of hydrological and error model parameters. This could contribute to the identification of the cause of previously

encountered problems in joint inference (Goal 2b specified in Sect. 1). Furthermore, the two selected catchments are different 30 in size, signatures (Table 2), and complexity of their hydrological response, so that the influence of the catchment or data properties can be assessed to some degree. To limit the scope of the study, we constrained the analysis to two catchments.

**Table 2.** Properties of the two case study catchments. P is the precipitation and  $R_{C-R_{C}}$  the runoff coefficient (calculated from cumulative streamflow and precipitation).  $Q_{\text{obs,max}}$ ,  $Q_{\text{obs,min}}$  and  $\overline{Q}_{\text{obs}}$  are the minimum, the maximum and the average streamflow, respectively.  $I_{\text{F,obs}}$  is the Flashiness Index (Baker et al., 2004).

[revised manuscript text omitted]

---

## Referee Report (RR1)

**Review for "A likelihood framework for deterministic hydrological models and the importance of non-stationary autocorrelation"**

**Summary**

The study introduces a new approach for treating non-stationary auto-correlation in residuals of hydrological models. A likelihood framework is used to evaluate different complexities in auto-correlation, and this is applied for different temporal resolution of data in two catchments. The improved representation of auto-correlation is shown to solve previously identified problems with joint calibration of hydrological and residual error models, and the authors demonstrate how auto-correlation is important for hydrological signatures such as flashiness index.

I found this study to be well written and insightful. The framework for comparing different complexities of the error models, and the experiments implemented via this framework, worked well. I believe this paper will be of clear interest for the HESS audience, and recommend it be published following the following minor comments being addressed.

**Main comments**

1. Title and focus: I found the title to be somewhat misleading, in that it refers to deterministic hydrological predictions, whereas this paper is largely about improving probabilistic predictions. I think the authors are under-selling their contribution by not including the term "probabilistic" or "uncertainty" in the title.

2. While results of this study are promising, the findings are somewhat limited by considering only 2 catchments. I feel the authors should recognize this in the discussion or conclusions, and emphasize that further research required on large datasets to determine which error model to use for different catchments, temporal resolutions, etc.

3. The authors state on page 3, lines 23-27 that "An explicit marginal distribution of streamflow (Krzysztofowicz, 2002) facilitates scientific communication and discussion, since hydrologists are generally more familiar with streamflow than with Box-Cox transformation parameters or distributions of the innovations of residuals." I see where the authors are coming from here, but predictive performance is likely as important as scientific communication. I would like to see comparison with existing error models listed as an area of further research in the discussion or conclusions.

4. In equation (7), the conditional probabilities when $Q(t_{i-1})$ =0 do not take into account information about $\mathbf{Q}$ from any previous time steps. In theory, these probabilities should take into account the last uncensored observation, and the fact that all observations in between are censored. See (Zeger and Brookmeyer, 1986) and (Hannachi, 2014) for details. This limitation should be recognised in equation (7).

**Minor comments**

**Pg 6, line 20:** "Note that, if the distributional assumptions about $D_Q$ hold at all points in time, $\eta(t_i)$ are a sample from a standard normal distribution, except for the lower tail, which can be lighter due to the truncation at zero at each individual time step. "

I agree there would be a lower proportion of $\eta(t_i)$ near the lower tail. But the pdf should have an area of 1, so I would have thought there would be an increase in samples of $\eta(t_i)$ close to the truncation point (which varies in time) to make up for this.

**Pg 10, line 6:** What does $\tau_{\min} = 0$ correspond to? Why was this chosen?

**Table 2:** It would be interesting to see the proportion of zero flows for these catchments, since zero flows are represented in the likelihood function.

**Figure 3,4:** It would also be useful to have arrows for "better" performance for all metrics (not just reliability and precision"

**Figures 3-6:** It would be useful to label all panels of figures to make it easier to refer to parts of figures.

**Pg 18, line 5-6:** " $\hat{I}_{F,\text{det}}$ is often similar to $I_F$ for E2 (Tables B1 and B2), indicating that the large part of the flashiness of the model output is due to the hydrological model response and only a small part is due to the stochastic variability added through the error model.

Isn't this true for all models?

**Pg 20, line 29-31:** "Figure 5 compares the predicted hydrographs of E1, E2 and E3a. In this case, allowing for different characteristic correlation times during precipitation events and dry periods (E3a) prevents the problematic behaviour encountered when making the constant correlation assumption."

It is not clear which problems the authors are referring to, and how E3a fixes them. Please provide more details.

**Figure 5:** It is difficult to see difference between models. Zooming in on a smaller time period may help emphasize differences.

**Minor edits (typos, etc)**

**Pg 5, line 17:** Move "e.g." to start of brackets

**Pg 10, line 28:** Change "10'000" to "10,000"

**Pg 11, line 15:** Change "empirical error models, as the ones" to "empirical error models, such as the ones "

**Pg 13, line 18:** Change "Streamflow data is a courtesy" to "Streamflow data is courtesy"

**Pg 15, line 12-13:** Change "prior believe" to "prior belief"

**Pg 19, line 16:** Change "the reliability measure shows a stable performance in," to "the reliability measure shows stable performance"

**Pg 29, line 11:** Change "catchments storage" to "catchment's storage"

**References**

HANNACHI, A. 2014. Intermittency, autoregression and censoring: a first-order AR model for daily precipitation. *Meteorological Applications,* 21**,** 384-397.
ZEGER, S. L. & BROOKMEYER, R. 1986. Regression Analsis with Censored Autocorrelated Data. *Journal of the American Statistical Association,* 81**,** 722-729.

---

## Author Response (AR3)

**Author's response**

Note that all the references to the manuscript (made by referees and by authors) refer to the manuscript resulting from the last revision cycle, not the revised markup version contained in this file.

**RC:** Referee comment

AC: Author comment

Changes made to the manuscript are highlighted in italic font.

**Reply to Referee #1**

**RC(1):** Table 1 provides an overview of the different error models and their assumptions. It would help readability to include an additional column that briefly summarizes in words (as opposed to just symbols) the main assumption or characteristic of each model. In fact, I did this for myself while reading the paper

AC(1): We agree that this is a very useful modification.

We included two additional columns in Table 1, specifying the assumptions regarding the correlation and the distribution of streamflow. In addition, we changed the order of the columns to be more consistent with the line of thinking of the reader.

**RC (2):** Evaluation criteria: I understand the authors are using evaluation criteria that have been defined elsewhere, and thus they simply adopt the same terminology, but unfortunately the terminology is counter-intuitive (e.g. low reliability = good, low precision = good). This doesn't affect the scientific merits of the paper, but it does affect communication and could be easily avoided by changing the terminology to something more intuitive

**AC (2):** We agree that the terminology is counter-intuitive and up to this point, we still preferred the consistency with previous literature over an intuitive name. However, given the repeated comments of the referees on this matter, we decided to make the performance metrics more intuitive and accept inconsistency with previous literature.

We redefined the reliability metric to its complement, i.e. new reliability = 1 – previous reliability. This was preferred over renaming it to something similar to "unreliability". The redefinition lead to changes in Equation 14, Figure 3 and 7, Table B1 and B2, and in several instances throughout the text. Additionally, we renamed the metric "precision" to "relative spread", which lead to changes in Figure 3, Table B1 and B2 and in several instances throughout the text.

RC (3): page 16, line 11: spell out the meaning of EN-det

AC(3): We agree that the term should be spelled out here.

We corrected this in the new version.

**RC (4):** page 28, lines 8-19: the discussion here points out that distributions with positive support lead to bad fits, but no details are given, for example which distributions were tested. As such the discussion in this paragraph is not entirely convincing. The last sentence suggests to use an inverse relation between skewness and discharge, but isn't that automatically fulfilled when using positive-support distributions? For example, the Gamma distribution is skewed near zero but becomes symmetric away from zero

**AC(4):** We agree that this part of the discussion is a bit vague, and that we should provide more detailed information on the tests that we run in this direction. In a limited exploration, we tested the lognormal distribution as a distribution with positive support, which is also more skewed when the mean is close to zero, like the Gamma distribution mentioned by the referee. However, for any distribution, the skewness would have to be extreme to allow for observed streamflow that is multiple orders of magnitude larger than the modelled streamflow. If that is not the case, then the inference will likely be affected by the strong outliers for small modelled streamflow. This was tentatively confirmed by our unsuccessful explorations with the lognormal distribution.

We clarified that we used the lognormal distribution and reformulated the reason why we believe that it is not worth keeping the mass balance at very small modelled streamflows. In addition, we removed the last sentence of the paragraph, which was correctly questioned by the referee.

**RC (5):** page 29, line 27: I would add the caveat that this conclusion is based on the chosen approach for handling non-normality (skewed student t distribution)

AC (5): We agree that this is an important caveat that should be added at that point.

We included a corresponding statement in the new version of the manuscript.

**Reply to Referee #2**

**RC:** The study introduces a new approach for treating non-stationary auto-correlation in residuals of hydrological models. A likelihood framework is used to evaluate different complexities in autocorrelation, and this is applied for different temporal resolution of data in two catchments. The improved representation of auto-correlation is shown to solve previously identified problems with joint calibration of hydrological and residual error models, and the authors demonstrate how autocorrelation is important for hydrological signatures such as flashiness index. I found this study to be well written and insightful. The framework for comparing different complexities of the error models, and the experiments implemented via this framework, worked well. I believe this paper will be of clear interest for the HESS audience, and recommend it be published following the following minor comments being addressed

AC: Thank you for this general feedback.

**RC (1):** Title and focus: I found the title to be somewhat misleading, in that it refers to deterministic hydrological predictions, whereas this paper is largely about improving probabilistic predictions. I think the authors are under-selling their contribution by not including the term "probabilistic" or "uncertainty" in the title.

**AC (1):** We acknowledge that the "probabilistic" or "uncertainty" are important keywords. The title does not really refer to deterministic predictions, however, but to deterministic hydrological models, which are made probabilistic by the presented likelihood framework. In our point of view, the term "likelihood" is sufficient to highlight the probabilistic nature of the chosen approach. We would prefer to keep the title as it is.

**RC (2):** While results of this study are promising, the findings are somewhat limited by considering only 2 catchments. I feel the authors should recognize this in the discussion or conclusions, and emphasize that further research required on large datasets to determine which error model to use for different catchments, temporal resolutions, etc.

**AC (2):** We agree that more extensive studies considering a larger number of catchments are needed to confirm the main findings of this paper. We supervised a Master thesis that extended this analysis to 3 additional catchments using different hydrological models, with very similar results (which are not published). Therefore, we are confident that the results are generalizable to a certain degree. This still has to be shown with an even larger number of catchments, however.

We included a corresponding statement in the conclusions.

**RC (3):** The authors state on page 3, lines 23-27 that "An explicit marginal distribution of streamflow (Krzysztofowicz, 2002) facilitates scientific communication and discussion, since hydrologists are generally more familiar with streamflow than with Box-Cox transformation parameters or distributions of the innovations of residuals." I see where the authors are coming from here, but predictive performance is likely as important as scientific communication. I would like to see comparison with existing error models listed as an area of further research in the discussion or conclusions.

**AC (3):** We agree that predictive performance is an important goal of any probabilistic model, which is why we thoroughly assessed the predictive performance based on multiple metrics in this study. We did so for multiple error models, some of which can be considered to be "existing" ones (E1 and E2) and demonstrated the superiority of our approach also in practical terms. We agree that there is a need for more comparative studies in this direction.

We mentioned the need for studies comparing multiple assumptions about the characteristics of the errors in the conclusions.

**RC (5):** In equation (7), the conditional probabilities when  $Q(t_{i-1}) = 0$  do not take into account information about Q from any previous time steps. In theory, these probabilities should take into account the last uncensored observation, and the fact that all observations in between are censored. See (Zeger and Brookmeyer, 1986) and (Hannachi, 2014) for details. This limitation should be recognised in equation (7).

AC (5): Thanks for highlighting these very relevant papers. Indeed, the approach described therein is very similar to ours, with the difference that it allows to keep the memory of the autocorrelated process during times of censored observations. This can indeed be transferred to streamflow measurements where zero flow measurements can occur. It requires numerically solving an integral with dimension proportional to the length of the zero streamflow periods, which can be very long in ephemeral catchments. Due to the nature of streamflow data, for which frequent changes between zero and non-zero data are unlikely, the expected benefit of keeping the correlation during zero flow time periods is small, especially when correlation is reduced anyway during precipitation events. However, the approach presented in the papers mentioned by the referee could be very relevant in some catchments or other hydrological applications in general.

We included a reference to those papers in a new paragraph at the end of Section 2.1, discussing the relevance of their approach and why we do not keep correlation during censored time periods.

**RC (6):** Pg 6, line 20: "Note that, if the distributional assumptions about DQ hold at all points in time,  $\eta(ti)$  are a sample from a standard normal distribution, except for the lower tail, which can be lighter due to the truncation at zero at each individual time step. "I agree there would be a lower proportion of  $\eta(ti)$  near the lower tail. But the pdf should have an area of 1, so I would have thought there would be an increase in samples of  $\eta(ti)$  close to the truncation point (which varies in time) to make up for this.

**AC (6):** If the type of distribution chosen for DQ extends below 0, the area below 0 is assigned as a probability mass to 0. Therefore, the area of the pdf in the first line of Equation (1) is not 1, but 1 - X, where X is the integral of DQ from minus infinity to 0. It is an alternative option to truncate the distribution at zero as suggested by the reviewer (which implies renormalization); this leads to a model with zero probability for a discharge of zero which may be adequate for some catchments. We mention this option in the paper as well. In the chosen approach, there is not a higher density of samples of  $\eta(ti)$  close to the truncation point, but a discrete probability of  $\eta(ti)$  to be exactly at the truncation point, which corresponds to zero flow and varies in time.

**RC (7):** Pg 10, line 6: What does  $\tau_{min} = 0$  correspond to? Why was this chosen?

**AC (7):**  $\tau_{min} = 0$  means that there is no autocorrelation during precipitation events. From visual inspection of some time series of  $\eta$ , it was clear that the correlation is strongly reduced during precipitation events in case of high-frequency data (Figure 9). Setting  $\tau_{min} = 0$  seemed like a pragmatic and simple choice, even though the true correlation is likely to be larger than 0 during precipitation events. This was accounted for in other error models (E3a, E4a) by inferring  $\tau_{min}$ . A third alternative would be to fix  $\tau_{min}$  at a value larger than 0.

**RC (8):** It would be interesting to see the proportion of zero flows for these catchments, since zero flows are represented in the likelihood function.

**AC (8):** Neither of the two investigated catchments contained any zero-flow data (see Table 2). The catchments were not selected to test the ability of the chosen distribution to deal with zero flows, although this would have been another interesting aspect that could be investigated in a future study. The presented framework allows to easily compare different distributional assumptions about streamflow, which could include distributions that differ in their treatment of zero flows. However, the focus of this study was on the description of correlation and the shape of the non-zero part of the distribution.

**RC (9):** Figures 3, 4: It would also be useful to have arrows for "better" performance for all metrics (not just reliability and precision"

**AC (9):** We agree that it makes sense to include such an arrow for the Nash-Sutcliffe efficiency in Figure 4, but not for the difference between the observed and modelled Flashiness Index and the streamflow error. The latter two have an optimal value of 0 and can be positive or negative, so a one directional arrow would be misleading.

We included such an arrow for the Nash-Sutcliffe Efficiency

**RC (10):** Figures 3-6: It would be useful to label all panels of figures to make it easier to refer to parts of figures.

AC (10): We agree that this makes the orientation easier.

We included labels for all subplots of Figures 3, 4, 5, 6 and 9. The references in the text were adapted to point to the specific labels of the subplots where appropriate.

**RC (11):** Pg 18, line 5-6: " $I_{F,det}$  is often similar to  $I_F$  for E2 (Tables B1 and B2), indicating that the large part of the flashiness of the model output is due to the hydrological model response and only a small part is due to the stochastic variability added through the error model." Isn't this true for all models?

**AC (11):** The referee is right when it comes to daily resolution, where the characteristic correlation time is usually only a few time steps (i.e. days). In that case, the difference between the error models (which differ in their treatment of correlation) w.r.t. the Flashiness Index is small. In case of high-resolution data (e.g. hourly), the inferred characteristic correlation time

with E2 can span many time steps, resulting in only minor increases of the Flashiness Index through the probabilistic model ( $I_F$  is only marginally larger than  $I_{F,det}$ ), as opposed to e.g. E1, where the missing correlation causes a large increase in the Flashiness Index of the stochastic model output compared to the output of the deterministic hydrological model. We acknowledge that the message would be clearer if the sentence above mentioned that for E2, this is the case for all temporal resolutions, which distinguishes it from many of the other error models in this respect.

We mentioned in the corresponding sentence that we refer to all the resolutions.

**RC (12):** Pg 20, line 29-31: "Figure 5 compares the predicted hydrographs of E1, E2 and E3a. In this case, allowing for different characteristic correlation times during precipitation events and dry periods (E3a) prevents the problematic behaviour encountered when making the constant correlation assumption." It is not clear which problems the authors are referring to, and how E3a fixes them. Please provide more details.

AC (12): We very much agree with the statement of the referee.

We expanded the sentence, mentioning that we mean the better behaviour of the error bands and the more realistic stochastic output of the model.

**RC (13):** Figure 5: It is difficult to see difference between models. Zooming in on a smaller time period may help emphasize differences.

AC (13): We fully agree with the referee.

We zoomed in to the first half of the time period and adapted the limits of the y-axis of each panel to better highlight the different results of the error models.

**RC(Edits):**

Pg 5, line 17: Move "e.g." to start of brackets

Pg 10, line 28: Change "10'000" to "10,000"

Pg 11, line 15: Change "empirical error models, as the ones" to "empirical error models, such as the ones"

Pg 13, line 18: Change "Streamflow data is a courtesy" to "Streamflow data is courtesy"

Pg 15, line 12-13: Change "prior believe" to "prior belief"

Pg 19, line 16: Change "the reliability measure shows a stable performance in," to "the reliability measure shows stable performance"

Pg 29, line 11: Change "catchments storage" to "catchment's storage"

AC(Edits): Thank you for pointing these out. We agree that all of these are errors.

We corrected them in this version of the manuscript.

**Additional Changes**

Throughout document:

- changed to more consistent indentation using the \par command.
- removed repeated reference to Baker, 2004 for Flashiness Index
- changed notation of "df" to "df" (also in supplementary material) to be consistent with journal requirements

Eq. (11): changed wording from "for", "else" to "if", "otherwise"

Pg. 4, Line 24: changed wording form "in wet and dry periods" to "regarding wet and dry periods"

Pg. 5, Line 27: added that the mean of the distribution DQ varies with time

Pg. 6, Line 20: deleted unclear and unnecessary sentence

Pg. 7, Eq. (7): Replaced tau with tau(ti)

Pg. 8, Line 1: mentioned more general case of vanishing correlation, which is relative to the measurement interval

Pg. 8, Line 18: "probability density" instead of probability

Pg. 9, Figure 1: Corrected "kurtosis" to "degrees of freedom" and added units of the standard deviation

Pg. 17, Figure 3: changed text direction of "better" on right hand side to agree with the text direction of the y-axis label

Pg. 26, Figure 9: changed legend

Pg. 33, Line 22: "method of skewing" changed to "method for skewing"

Pg. 34, Eq. (6) and (7) changed "for" to "if"

Pg. 36, Line 1-4: corrected variable name of Flashiness Index.

**Additional references**

[revised manuscript text omitted]

where  $Q = (Q(t_0), Q(t_1), \dots, Q(t_n))$ . Let  $\hat{x}$  denote the quantity x that is related to the hydrological parameter values at the maximum posterior density. The Flashiness Index is calculated for the observations,  $I_{\text{F,obs}} = I(Q_{\text{obs}})$ , the output of the

25 deterministic hydrological model,  $\hat{I}_{F,det} = I(\hat{Q}_{det})$ , and the individual stochastic realisations of the predictive streamflow sample,  $I_F = \text{median}(I(Q_j))$ .  $I_F$  is sensitive to the amount of autocorrelation in a streamflow time series, as well as the height of the peaks of  $Q_{det}$  (since  $Q_j$  depends on  $Q_{det}$ ).

**2.4.2 Reliability**

Reliability is defined equivalently similarly to McInerney et al. (2017), as:

$$\Xi_{\rm reli} = 1 - \frac{2}{n+1} \sum_{i=0}^{n} |F_{Q(t_i)}(Q_{\rm obs}(t_i)) - F_{\Psi}(F_{Q(t_i)}(Q_{\rm obs}(t_i)))|$$
(14)

where  $\Psi = \{F_{Q(t_i)}(Q_{obs}(t_i)) | i \in \mathbb{N}, 0 \le i \le n\}$ ,  $F_{\Psi}$  is the empirical cumulative distribution function of  $\Psi$  and  $F_{Q(t_i)}$  is the empirical cumulative distribution function of the predicted streamflow at time  $t_i$ .  $\Xi_{reli}$  can take values in the interval [0, 1], where smaller-larger values of  $\Xi_{reli}$  correspond to better, and zero to perfect, reliability and unity means perfect reliability. It summarises the deviance of the observations from measures the degree to which the observations are consistent with being a sample of the predictive distribution over all time points, and the distance is measured. Since comparison happens in the uniform space. Therefore, the influence of heavy outliers on  $\Xi_{reli}$  is limited. Note that we use the complement of the reliability

10 measure proposed by McInerney et al. (2017), in order to allow for a more intuitive interpretation (larger values mean larger reliability).

**2.4.3 Precision Relative Spread**

The precision metric The relative spread is an indicator for the width of the predictive distributions over all time points, and was proposed by McInerney et al. (2017) as:

15
$$\Omega_{\text{spread}} = \frac{\sum_{i=0}^{n} \sigma_Q(t_i)}{\sum_{i=0}^{n} Q_{\text{obs}}(t_i)}$$
(15)

where  $\sigma_Q(t_i)$  is the standard deviation of the predictive distribution at time point  $t_i$  calculated from the ensemble of all stochastic predictions at that point in time.  $\Omega_{\text{prec}} \in \mathbb{R}^+ \Omega_{\text{spread}} \in \mathbb{R}^+$ , and small values of  $\Omega_{\text{prec}}$  indicate high precision  $\Omega_{\text{spread}}$  indicate precise predictions or small predictive uncertainty. The smaller the predictive uncertainty, the better the quality of the underlying model, given that the predictions are not overconfident. While McInerney et al. (2017) use the name "precision" for  $\Omega_{\text{prec}}$  we believe that "relative spread" is a more appropriate term considering its actual meaning.

20  $\Omega_{\text{spread}}$ , we believe that "relative spread" is a more appropriate term considering its actual meaning.

**2.4.4 Nash-Sutcliffe Efficiency**

The Nash-Sutcliffe Efficiency (Nash and Sutcliffe, 1970),  $E_{N,f}$  (f for function), is defined as:

$$E_{\rm N,f}(\boldsymbol{Q}, \boldsymbol{Q}_{\rm obs}) = 1 - \frac{\sum_{i=0}^{n} (Q(t_i) - Q_{\rm obs}(t_i))^2}{\sum_{i=0}^{n} (Q_{\rm obs}(t_i) - \overline{Q}_{\rm obs})^2}$$
(16)

where  $Q = (Q(t_0), Q(t_1), \dots, Q(t_n))$ . It is used in this study to assess the output of the hydrological at the maximum posterior parameter density,  $\hat{E}_{N,det} = E_{N,f}(\hat{Q}_{det}, Q_{obs})$ , as well as the stochastic simulations,  $E_N = \text{median}(E_{N,f}(Q_j, Q_{obs}))$ . It is used as a rough measure of how well two hydrographs correspond to each other, primarily with the goal of identifying very poorly fitting hydrographs. It is known to be sensitive to errors in high flows (Legates and McCabe, 1999), which can be of particular practical interest. Therefore it complements the other measures, which are less informative with respect to errors in high flows.

**2.4.5 Relative error in total cumulative streamflow**

As a measure of systematic over- or under-prediction of streamflow, we calculate the relative error in total cumulative streamflow:

$$\Delta(\boldsymbol{Q}, \boldsymbol{Q}_{\rm obs}) = \frac{\sum_{i=0}^{n} Q_{\rm obs}(t_i) - Q(t_i)}{\sum_{i=0}^{n} Q_{\rm obs}(t_i)}$$
(17)

5 It is calculated w.r.t. the model output based on the parameter values at the maximum posterior density;  $\hat{\Delta}_{Q,\text{det}} = \Delta(\hat{Q}_{\text{det}}, Q_{\text{obs}})$ , as well as for the ensemble of individual stochastic simulations:  $\Delta_Q = \text{median}(\Delta(Q_j, Q_{\text{obs}}))$ . Note that, contrary to McInerney et al. (2017),  $\Delta_Q$  is the median error of all the individual hydrograph realisations, not the error of the average hydrograph.

**3 Case study setup**

**3.1 Catchments and data**

- 10 The probabilistic framework developed in Sect. 2.1 was tested in two case study sites, the Murg and the Maimai catchments, which are described in this section. The Murg river flows through a hilly headwater catchment in temperate climate with a size of 80 km2 in northeastern Switzerland. Some key hydrological summary statistics are listed in Table 2. Land use is predominantly agricultural (50 %), with forested headwaters (30 %) and a considerable part of urban areas (10 %). The mean elevation is 652 m a.s.l., spanning from 466 to 1035 m a.s.l. Streamflow peaks can be quite sharp, especially for small events, in
- 15 which baseflow conditions are reached again within just a few hours. This is potentially due to impervious areas being drained directly into the river. The data consists of hourly averages of streamflow, precipitation and potential evapotranspiration from January 1995 to December 2002. Calibration was performed in the first 5 years (Jan 1995-Dec 1999) and validation in the consecutive 3 years (Jan 2000-Dec 2002). Streamflow data is a courtesy of the Swiss Federal Office for the Environment (FOEN). Precipitation and potential evapotranspiration are based on meteorological data (Meteoschweiz, 2018) and were
- 20 processed by the Swiss Federal Institute for Forest, Snow and Landscape Research (WSL), with the preprocessing tools of PREVAH (Viviroli et al., 2009).

The Maimai experimental catchments are a set of small headwater catchments with a long history of hydrological research. They are located on a deeply incised hillslope on the South Island of New Zealand. The area is forested and the climate is considerably more humid than in the Murg catchment (Table 2). The site was chosen for this study due to its homogeneous

- 30 the Maimai catchment (see McGlynn et al. (2002) for an extensive overview). Calibration was performed based on data from Jan 1985-Dec 1986, and validation during Jan-Dec 1987. The data was kindly provided by Jeffrey McDonnell.

**Table 2.** Properties of the two case study catchments. P is the precipitation and  $R_{\rm C}$  the runoff coefficient (calculated from cumulative streamflow and precipitation).  $Q_{\rm obs,max}$ ,  $Q_{\rm obs,min}$  and  $\overline{Q}_{\rm obs}$  are the minimum, the maximum and the average streamflow, respectively.  $I_{\rm F,obs}$  is the Flashiness Index(Baker et al., 2004).

| Catchment | Area
[km 2 ] | $P$ $[mm a^{-1}]$ | RC [-] | $Q_{ m obs,max}$ $[ m mmh^{-1}]$ | $Q_{ m obs,min}$ $[{ m mm}{ m h}^{-1}]$ | $\overline{Q}_{ m obs}$ $[{ m mm}{ m h}^{-1}]$ | $I_{ m F,obs}$ [-] |
|-----------|----------------------------|-------------------|--------------------------|----------------------------------|-----------------------------------------|------------------------------------------------|--------------------|
| Murg      | 80                         | 1369              | 0.57                     | 2.7                              | 1e-2                                    | 0.089                                          | 0.053              |
| Maimai    | 0.07                       | 2349              | 0.62                     | 8.5                              | 1e-4                                    | 0.17                                           | 0.13               |

Figure 2. Structure of the deterministic hydrological model used in this study.  $P_u$  is the precipitation and  $E_u$  the evapotranspiration.  $S_u$  represents the active water content of the unsaturated zone, while  $S_f$  is a non-linear reservoir representing the fast flow component.

While the resolution of the original data was hourly, we produced data sets with 6-hourly and daily resolution by aggregation for both catchments. This setup allows us to systematically investigate the effect of the temporal resolution of the data on the joint inference of hydrological and error model parameters. This could contribute to the identification of the cause of previously encountered problems in joint inference (Goal 2b specified in Sect. 1). Furthermore, the two selected catchments are different in size, signatures (Table 2), and complexity of their hydrological response, so that the influence of the catchment or data

properties can be assessed to some degree. To limit the scope of the study, we constrained the analysis to two catchments.

**3.2 Deterministic Hydrological Model**

5

The hydrological model used throughout this study is a simple, lumped bucket model with two reservoirs (Figure 2), which are meant to represent the unsaturated soil zone and the subsurface flow being fed by it. A slower flow component is included though a linear outflow from the unsaturated zone reservoir directly. Due to its simplicity, and due to the fact that it is not clear whether the chosen model structure is suited for the studied catchment a priori, we expect systemic difficulties in reproducing the observed streamflow dynamics. This is a very common situation in hydrological modelling and it will lead to correlated and potentially heteroscedastic and non-normal errors. This allows us, in principle, to test the error models (Sect. 2.2) under realistic conditions. The streamflow simulated by this deterministic model is denoted as  $Q_{det}(t, \theta) = Q_s(t, \theta) + Q_f(t, \theta)$ , where  $Q_s$  is the slow response of the model,  $Q_f$  is the fast response and  $\theta = (C_e, S_{max}, k_u, k_f)$  are the calibrated hydrological parameters. The fluxes  $(E_u, P_u, Q_u, Q_s, Q_f)$  and states  $(S_u, S_f)$  of the model are given by:

$$\frac{dS_{u}}{dt} = P_{u} - E_{u} - Q_{u} - Q_{s}$$

[revised manuscript text omitted]